# Angular and linear speed cells in the parahippocampal circuits

Davide Spalla [1], Alessandro Treves [1] & Charlotte N. Boccara [2✉]

An essential role of the hippocampal region is to integrate information to compute and update representations. How this transpires is highly debated. Many theories hinge on the integration of self-motion signals and the existence of continuous attractor networks (CAN). CAN models hypothesise that neurons coding for navigational correlates – such as position and direction – receive inputs from cells conjunctively coding for position, direction, and self-motion. As yet, very little data exist on such conjunctive coding in the hippocampal region. Here, we report neurons coding for angular and linear velocity, uniformly distributed across the medial entorhinal cortex (MEC), the presubiculum and the parasubiculum, except for MEC layer II. Self-motion neurons often conjunctively encoded position and/or direction, yet lacked a structured organisation. These results offer insights as to how linear/angular speed – derivative in time of position/direction – may allow the updating of spatial representations, possibly uncovering a generalised algorithm to update any representation.

[1] Sissa, Via Bonomea 265, 34136 Trieste, Italy. [2] University of Oslo, Faculty of Medicine, IMB, Sognsvannsveien 9 Domus Medica, 0372 Oslo, Norway.
✉email: charlotte.boccara@medisin.uio.no

The brain is constantly bombarded with all types of information that need to be related to each other in order to build a coherent picture of one's surroundings (or of the situation one is experiencing at a given time). A broad range of studies have indicated that one of the main roles of the hippocampal region is to do that: integrate multimodal information – coming from a variety of sensory and associative cortices, as well as deeper structures – to build a dynamic representation of an environment or an event[1–4]. The accurate updating of these representations – and their comparison with previously stored representations as well as projections of likely near futures – could allow one to evaluate the outcome of a range of decisions in order to react adequately. In the context of spatial cognition, information integration is implemented in interconnected subareas of the hippocampal region[5,6] through neurons coding for specific instantaneous navigational features such as position (place cells)[7], direction (head direction cells)[8], local metrics (grid cells)[9] and boundaries (border cells)[10,11]. Successful navigation is also thought to depend on the accurate updating of spatial representations, which themselves would hinge on self-motion signals and their integration with both positional and directional information[12–15]. Despite their crucial role, where and how self-motion signals are integrated remains largely elusive.

Linear speed modulation has so far mainly been reported in the CA1 region of the hippocampus and the medial entorhinal cortex (MEC) in conjunction with positional information or as a non-conjunctive code (speed cells)[16,17]. In addition, speed has been reported to influence oscillatory activity recorded in the hippocampal field potential where the theta power seems correlated to locomotory activity[18–20]. In contrast, angular velocity coding has not been yet established in principal neurons of the hippocampal region as a non-conjunctive code. A recent report has shed light on a few cells (7%, 12 units) in the superficial layers of the MEC, modulated by angular head velocity in conjunction with other spatial correlates, hinting at the presence of angular coding in this region[21]. Yet, these results need to be confirmed and the existence of angular velocity in the hippocampal region independently of spatial or directional coding remains to be established. Reports of angular velocity modulation have – so far – mostly come from recordings of subcortical structures (e.g., lateral mammillary nuclei, dorsal tegmental nucleus), linked to the processing of vestibular information, the retrosplenial cortex or as a modulating factor of head direction coding[22–27].

Most crucially, it has remained unclear whether angular velocity coding requires precisely tuned connectivity structures. In fact, some theoretical models within the class of continuous attractor networks (CANs)[13,15,28–32] assume a precise wiring diagram, in which neurons coding for instantaneous navigational correlates – such as position and/or direction – typically receive inputs from cells conjunctively coding for position, direction and self-motion[13,15,28–33]. These conjunctive cells – sometimes referred to as the 'hidden layer' – are hypothesised to mediate the shift of activity from position (or direction) at time $t$ to the next position (or direction) at time $t_{+1}$ (Figs. 1c and 2b). Exciting evidence compatible with such CAN models was recently provided by investigations of the *Drosophila melanogaster* central complex, where head direction cells whose activity was modulated by angular velocity were shown to be organised in a ring according to their preferred direction[34,35]. It is, however, unclear whether such precisely organised connections could be found in more complex species, or for functions other than directional coding.

To understand the circuit mechanism by which spatial representations can be updated in mammals, we carefully mapped the activity of individual neurons recorded in all the layers of three main areas of the rat parahippocampal region: the MEC, the parasubiculum and the presubiculum. We specifically investigated whether these neurons could respond to both linear and angular self-motions signals. Our study reveals the existence of parahippocampal neurons coding conjunctively for direction, position and self-motion, possible evidence for the elusive 'hidden layer', pillar of many CAN models[13]. Such a hidden layer would for example consist of a population of cells simultaneously sensitive to head direction and angular head velocity. However, the thorough examination of self-motion neurons did not indicate any logical scheme that could be realised through a specific wiring diagram. It appears that direction, position and speed selectivity are randomly admixed with each other, lacking any obvious structure in the coding properties of individual cells. In addition, we observed a continuous and almost homogenous distribution of properties across brain regions, without evidence of topographical or physiological clustering.

## Results

To understand whether and how both linear and angular speed modulation are integrated in the hippocampal circuits, we analysed the activity of 1436 principal neurons recorded in all layers of three interconnected subareas of the parahippocampal region of rats freely exploring open environments of various sizes (MEC: 396 cells; presubiculum: 605 cells; parasubiculum: 435 cells, Fig. 1a)[36].

**Angular velocity coding in the parahippocampal region.** First, we sought to determine whether parahippocampal neurons responded to the angular head velocity (AHV) signal, which is the derivative of head direction in time. To that end, we computed, for each cell, its angular velocity score as the Pearson product moment correlation between the instantaneous value of angular velocity and the firing rate of the cell across the recording session (see Methods and Supplementary Fig. 3a). We defined cells as AHV modulated when their score was greater than the 99th percentile of the shuffled distribution. This method led us to classify a total of 246 cells as angular head velocity cells, amounting to about one sixth of all parahippocampal cells (MEC: 16.9%; Prs: 17.0%; PaS: 17.2%; Fig. 1b–e and Supplementary Fig. 1a–d). We chose a region-wise shuffle, in which shuffles from all cells in the same region are pooled together (see Methods), based on the uniformity of behaviour and firing rate across sessions and neurons, as well as to offer a point of comparison with previously published studies[17,36]. However, to exclude the possibility of session- or cell rate-based artifacts, we verified that we found similar results with a within-cell shuffle analyses (see Methods and Supplementary Fig. 2d–f). AHV modulation was uniformly distributed across all layers of each region, except in MEC LII, which showed no such modulation, regardless of the shuffling method used (Kolmogorov–Smirnov test, $p < 0.001$, Supplementary Fig. 2).

As per our definition, AHV cells are neurons whose firing rate is positively modulated by angular velocity, meaning that these cells are more active when the animal is turning its head. About half of the AHV cells had their activity modulated solely when the animal had its head turning only in one direction, either clockwise (CW) or counterclockwise (CCW). The other half of the AHV cells were bidirectional (BiDir) and modulated by angular motions in both directions (MEC: 26.8% CW, 31.0% CCW, 58.2% BiDir; Prs: 25.9% CW, 23.1% CCW, 64.4% BiDir; PaS: 32.0% CW, 30.0% CCW, 52.0% BiDir; Fig. 1d and Supplementary Fig. 1a–d, note that the percentages do not sum to 100%, since unidirectional and bidirectional scores are not mutually exclusive, overlaps are represented in darker shades in the figures, see Methods). All layers of each region presented similar proportions of CW, CCW and

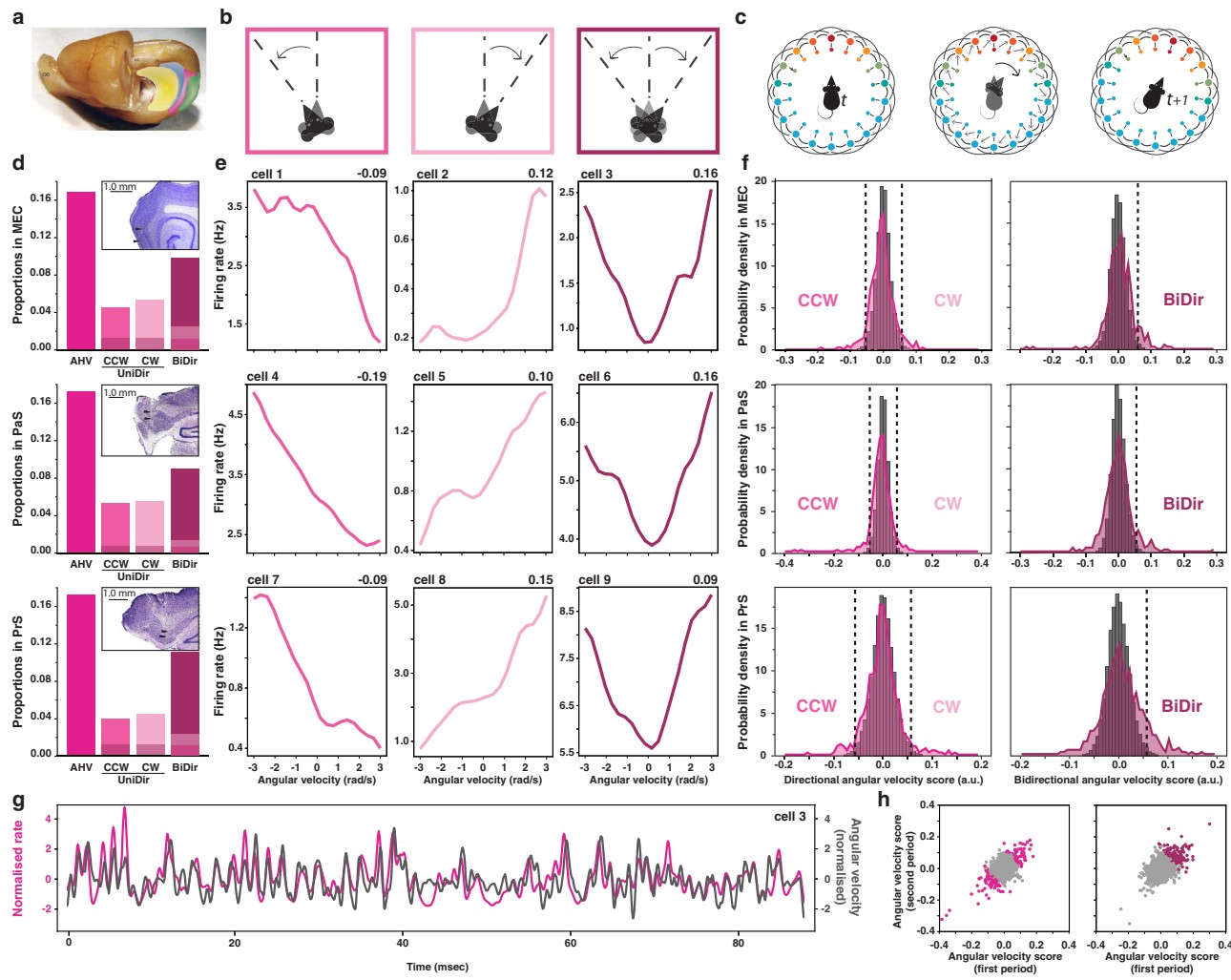

**Fig. 1 Angular velocity cell in the parahippocampal cortex. a** Whole rat brain, with partially removed left hemisphere to enable a midsagittal view of the right hemisphere and outlines of hippocampal formation (yellow), presubiculum (blue), parasubiculum (pink) and MEC (green). Adapted from[36] (**b**) Schematic representation of three type of angular head velocity (AHV) movement, from left to right: counterclockwise (CCW, dark pink), clockwise (CW, light pink) and bidirectional (BiDir, purple). **c** Schematics ring attractor depicting theoretical updating of head direction code from position at time $_t$ (left) to position at a later time $_{t+1}$ (right) following angular movement (middle). The outer layer of head direction (HD) cells is connected to a 'hidden' inner layer of conjunctive HD-by-AHV cells. The colour represents neural activation from maximum (red) to minimum (blue). **d** Proportions of AHV cells within MEC (top), PaS (middle) and PrS (bottom). From left to right: all AHV cells (fuchsia, MEC:17%, $n = 67$; PaS:17%, $n = 75$; PrS:17%, $n = 104$), CCW-AHV cells (dark pink, MEC:5%; PaS: 5%; PrS:4%), CW-AHV cells (light pink, MEC:5%; PaS: 6%; PrS: 5%), and BiDir-AHV cells (purple, MEC:10%; PaS:9%; PrS:11%). The shaded areas represent the intersection between AHV-CCW & AHV-BiDir (darker shade) or between AHV-CW & AHV-BiDir (lighter shade). Upper right corner boxes: representative Nissl-stained sagittal section showing example recorded track for each area. **e** Example AHV cells in MEC (top), PaS (middle) and PrS (bottom) showing firing rate as a distribution of angular velocity (in rad/s), score in upper right corner. From left to right: CW-AHV (dark pink), CCW-AHV (light pink) and BiDir-AHV (purple). **f** Distribution of unidirectional (left) and bidirectional (right) AHV scores across MEC (top), PaS (middle) and PrS (bottom) cell population comparing observed (coloured curve) and shuffled data (grey bars). Dashed lines represent 99 percentile thresholds for CCW- and CW-AHV (left) and Bidir-AHV (right). **g** Snapshot comparison between z-scored firing rate of an example AHV cell (pink curve) and instantaneous angular head velocity (black curve). **h** Scatter plots showing the correlation between the AHV scores calculated in the first and second half of each recording sessions (Left: unidirectional, CW- and CCW-AHV cells in pink; Right: bidirectional, BiDir-AHV cells in purple; rho stability = 0.52). Credits to Silvia Girardi for schematics and drawings.

bidirectional AHV cells (Kolmogorov–Smirnov test, $p < 0.001$, Supplementary Fig. 2a–c) with the exception of MEC layer II who did not present any AHV modulation. The use of a within-cell shuffle threshold did not significantly impact CW, CCW and BiDir percentages (Binomial test, $p < 0.001$, Supplementary Fig. 2). AHV modulation appeared stable in time across all regions, and we observed no change in modulation intensity (AHV score) while comparing successive half-sessions (Pearson correlation $\rho = 0.52$, $p < 0.001$, Fig. 1h). When calculating the Pearson product moment correlation, we found a small percentage of negatively modulated –

or anticorrelated – AHV cells, distributed across all regions and all layers (MEC: 2.5%; Prs: 6.1%; PaS: 6.4%; see methods for definition of anticorrelated cells and Supplementary Fig. 3 for examples). This cell population was analysed separately, taking into account that anticorrelated AHV cell may reflect a modulation not related to AHV, but rather a process that takes places only when the head of the animal is still.

To account for nonlinear coding and compare our results with recent reports[21], we further analysed all recorded cells with a generalised linear model (GLM) approach[16] for which neuron firing

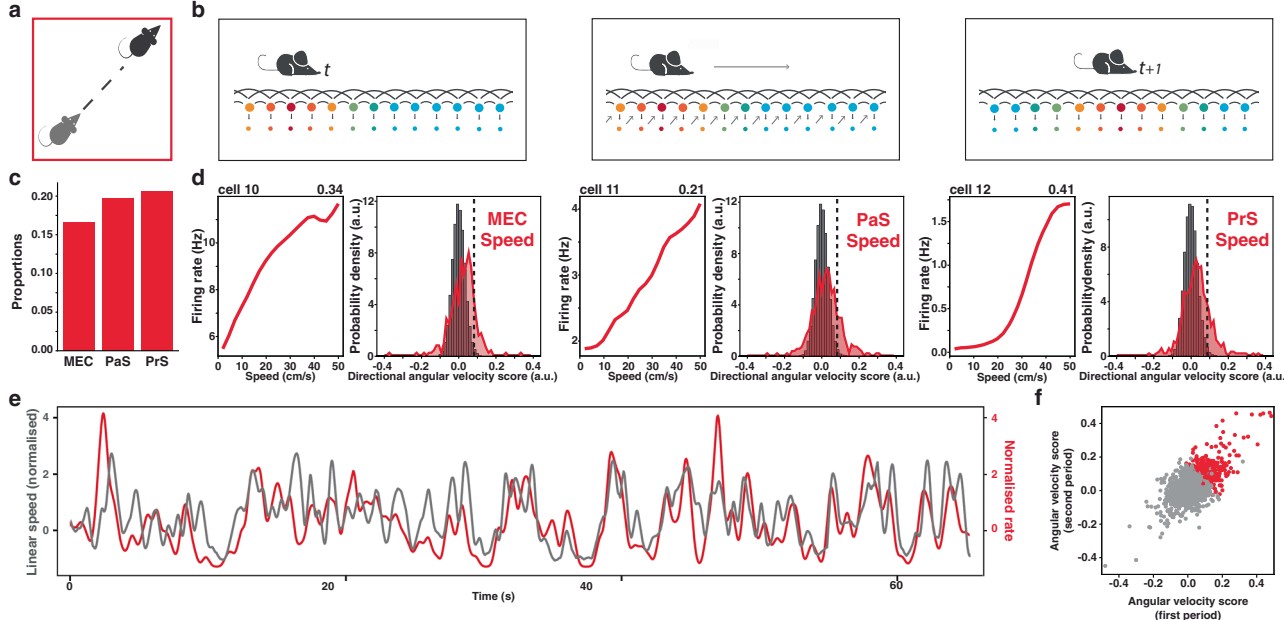

**Fig. 2 Speed cells are distributed across the parahippocampal cortex. a** Schematic representation of linear velocity movement (**b**) Schematics partial ring attractor depicting theoretical updating of positional code from position at time $_t$ (left) to position at a later time $_{t+1}$ (right) following linear movement (middle). The outer layer of conjunctive grid-by-HD cells is connected to a 'hidden' inner layer of conjunctive grid-by-HD-by-speed cells. The colour represents neural activation from maximum (red) to minimum (blue). **c** Proportions of speed cells in MEC (left, 16.7%, $n = 66$), PaS (middle, 19.8%, $n = 86$) and PrS (right, 20.1%, $n = 125$). These proportions are not significantly different. **d** Example speed cells (left) and general distribution of speed scores (right) in MEC (left panel), PaS (middle panel) and PrS (right panel). The example tuning curve on the left plot shows the mean firing rate (red) as a function of speed, between 2 cm/s and 50 cm/s. Scores are in the upper right corner. The histograms on the right compare the observed (coloured curve) and shuffle data (grey bars), dashed lines represent the 99-percentile threshold. **e** Snapshot comparison between z-scored firing rate of an example speed cell (red curve) and instantaneous linear speed (black curve). **f** Scatterplot showing the correlation between the speed score calculated in the first and second half of the recording session (speed cells in red, rho stability = 0.61). Credits to Silvia Girardi for schematics and drawings.

profiles are calculated as a function of velocity. For this approach, velocity values were treated as categorical variables and would allow, for example, to select cells responding to a specific speed band (see Methods and Supplementary Fig. 3b). While the cell populations yielded by the two methods were significantly overlapping (Binomial test, $p < 0.001$, Supplementary Fig. 3c), a substantial fraction of AHV modulated cells were only captured by one or the other method. Out of 182 AHV cells solely picked up by the GLM method, about half of them (54%) were anticorrelated, significantly overlapping with the cell population showing a negative Pearson score (Binomial test, $p$ value < 0.001; see Supplementary Fig. 3d). Only a small number of AHV cells solely picked up by the GLM method (11%) were responsive to a particular value of angular head velocity (measured with a gaussian fit), while the rest either did not show a modulation profile with a simple shape or presented a weak linear correlation (i.e., just below the Pearson threshold; see Supplementary Fig. 3d). The choice of method had little impact on the uniform distribution across regions, except for MEC layer II (see Supplementary Fig. 3d). Given the possible ambiguity linked to negative modulation, the dominance of linear (or pseudo-linear) coding in positively modulated AHV (see Supplementary Fig. 4) and to facilitate the comparison with linear speed analyses[17], we decided to present in the main figures the results obtained with the more conservative Pearson scoring methods. Yet, we systematically reported in the text or in the supplementary figures the results for the GLM-only population – i.e., cell classified as AHV by the GLM method and non-AHV by the Pearson scoring method.

**Parahippocampal neurons upstream of the entorhinal cortex code for linear speed.** Once established that angular velocity

coding was widespread across several parahippocampal areas, we tested whether linear speed coding could also extend beyond the medial entorhinal cortex (MEC). To that end, we determined the speed score of each cell as a Pearson product moment correlation between the instantaneous value of rectilinear speed and the firing rate of the cells across the recording session (see Methods and Supplementary Fig. 3a). We classified a total of 277 speed cells with a region-wise threshold and 295 with a within-cell threshold. Our results confirmed the existence of speed cells uniformly in all layers of the MEC in similar proportions to what was previously reported[17] (MEC all: 16.7%, LII: 23.9%, LIII: 18.7%, LV: 12.7% and LVI: 13.7%; Fig. 2a–e and Supplementary Figs. 1a, b and 2a). In addition, we observed that rectilinear speed signals could be found upstream of the MEC, in about one fifth of both PrS and PaS cells (Prs: 20.6%; PaS: 19,8%, Fig. 2a–e and Supplementary Fig. 1c–d). Speed cells were uniformly distributed across all layers in each area regardless of the shuffling method (Kolmogorov–Smirnov test, $p$ value < 0.001; Supplementary Fig. 2a–c). As for AHV cells, speed cells were stable across time (Fig. 2f). We also observed a significant proportion of cells negatively modulated by speed – cells whose firing was maximal at low speed and minimal a high speed. Such anticorrelated cells were distributed across all regions and all layers (MEC: 4.5%; Prs: 4.1%; PaS: 5.5%; see Supplementary Fig. 3e). For similar reasons to those given when reporting anticorrelated AHV cell, this cell population was treated separately in subsequent analyses.

Like for AHV cells, we further analysed speed modulation with a GLM approach, detecting a population of speed cells significantly overlapping with this obtained with the Pearson scoring method (binomial test, $p$ value < 0.001; Supplementary Fig. 3c). Contrary to AHV scoring, the GLM method yielded a

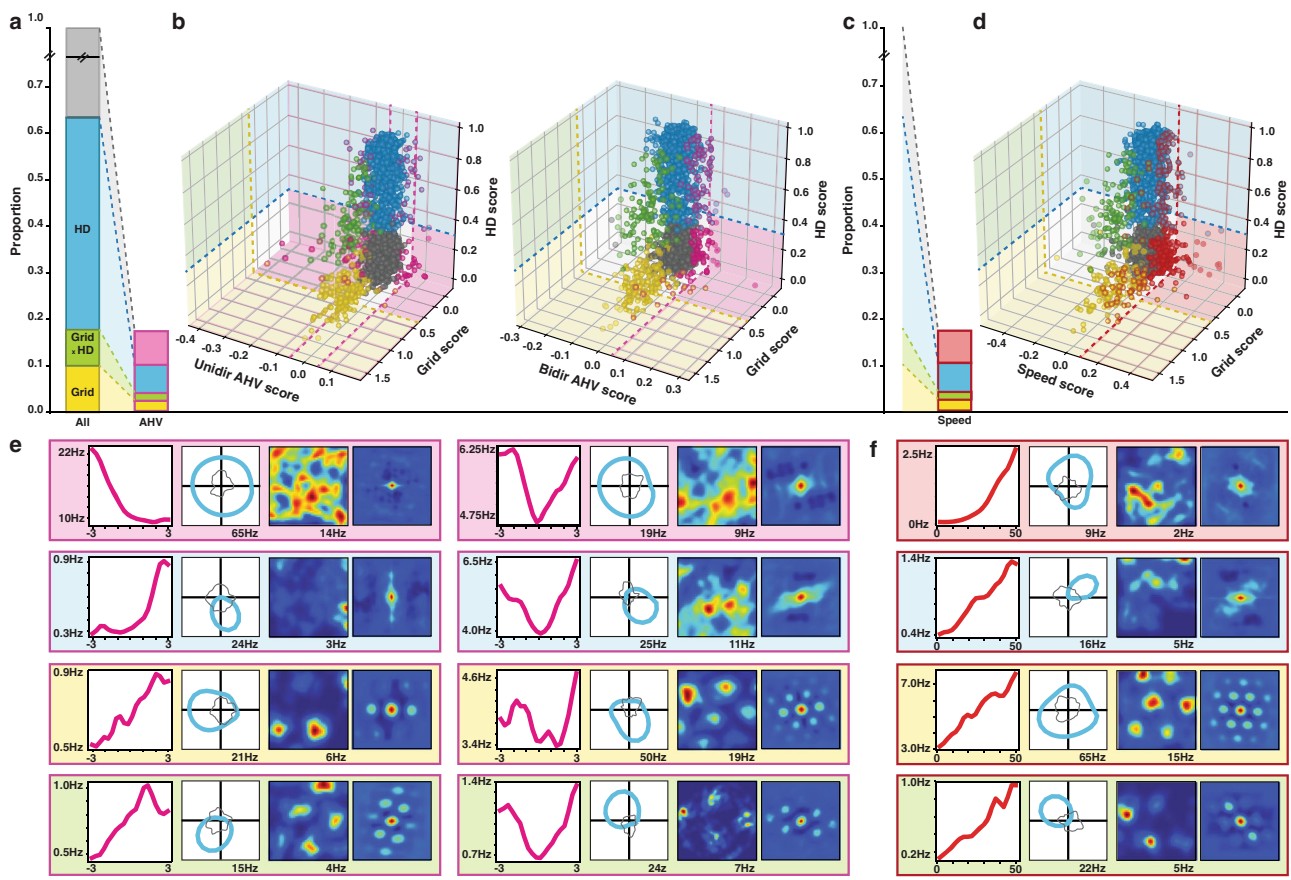

**Fig. 3 The majority of speed and angular velocity cells presents conjunctive coding. a** Intersection between angular head velocity (AHV), grid and head direction (HD) cells. The bar on the left shows the proportions of cells qualifying as grid (yellow, 17.8%), HD (blue, 53.3%) and grid x HD (green, 7.7%) cells in the total population. In grey are cells neither coding for HD nor grid. The bar on the right (pink contour) shows the same distribution among cells qualifying as AHV cells (grid: 13.4%; HD: 55.3%; grid x HD: 5.7%). **b** Scatterplots showing the intersection between grid, HD and AHV scores. Plot on the left: unidirectional – UniDir – AHV: i.e., CCW-AHV and CW-AHV. Plot on the right: bidirectional – BiDir– AHV. The colour code is the same as in (**a**). A pink contour denotes a modulation by AHV. Dotted lines represent region-averaged classification thresholds, computed to guide the visualisation. **c** Same as in (**a**) for speed cells. The red contour bar represents percentages of principal cells within the speed cell population (grid: 18.8%; HD: 35.4%; grid x HD: 3.3%). **d** Same as in (**b**) for speed cells. A red contour denotes a modulation by speed. **e** Examples of the four different kinds of AHV modulated cells, colour coded as in (**a**). From left to right: AHV tuning curve, HD polar plot, spatial firing rate map, spatial autocorrelogram. **f** Examples of the four different kinds of speed modulated cells, colour coded as in (**c**). Same plot as in (**e**).

much higher number of speed-modulated neurons (551 selected neurons). Out of 296 speed-modulated cells solely picked up by the GLM method, about one fifth were anticorrelated with the absolute value of speed, significantly overlapping with the cells selected on a negative Pearson score (Binomial test, *p* value < 0.001; see Supplementary Fig. 3e). Only a very small number of them (8%) were responsive to a particular speed band (measured with a gaussian fit), while the rest either did not show a simple-shape modulation profile or presented a weak linear correlation (i.e., just below the Pearson threshold; see Supplementary Fig. 3e). None of the speed cells solely picked up by the GLM method presented a steep sigmoid shape. Given that the Pearson methods allow some deviation from a strict linear coding and in order to allow for comparison with seminal reports of speed cells in the MEC[17], we decided to use the more conservative Pearson scoring methods for further analyses reported in the main figures. Yet, like for AHV modulated cells, any divergent results with cell classified as speed cells by the GLM method are reported in the text.

**Conjunctive coding of primary (place, direction) and derivative (velocity) signals**. Because angular head velocity (AHV) and

speed are the derivative in time of head direction and position, respectively, we defined positional and directional signals as primary and self-motion signals as derivative.

Given the key role of a 'hidden layer' of cells presenting conjunctive coding for continuous attractor network (CAN) theories, we next sought to determine to which degree self-motion signals are co-existing in conjunction with other types of coding at the unit level. To that end, we computed the grid and head direction (HD) scores of each recorded unit. We labelled as 'significantly modulated', cells whose score exceeded the 99th percentile of the score distribution calculated on shuffled data (region-wise shuffles, see Methods). According to these parameters, the majority (80.4%) of AHV cells coded for at least one other feature (HD: 55.2%; grid: 13.4%; grid-by-HD: 5.6%; rectilinear speed: 33.7%; Figs. 3a, b, e and 4a–c). These percentages were similar to the percentages observed in the general population (HD: 53.3%; grid: 17.8%; grid-by-HD: 7.7%; rectilinear speed: 19,2%; Fig. 3a). A similar distribution of conjunctive coding was observed among speed cells apart from a decrease in HD modulation (HD: 35.4%; grid: 18.7%; grid-by-HD: 3.2%; AHV: 29.9%; Figs. 3c, d, f and 4a–c). The GLM method yielded very similar results (AHV cell population: HD: 50.6%; grid: 17.6%; grid-by-HD: 6.4%, speed cell population: HD: 45.6%; grid: 17.9%; grid-by-HD: 4.4%). The use of within-cell shuffles resulted in

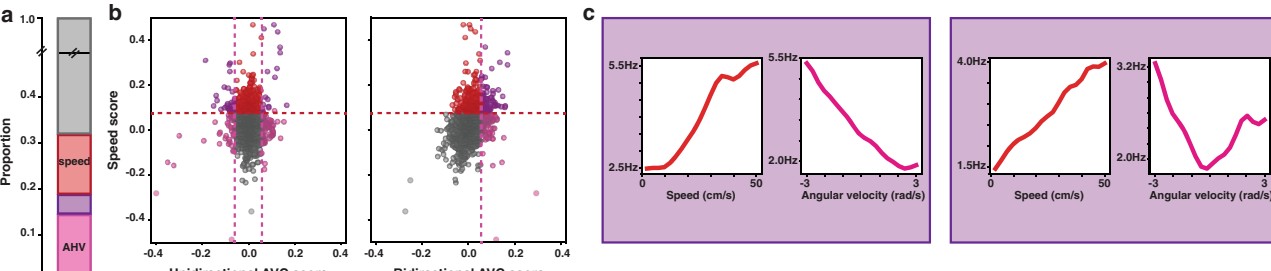

**Fig. 4 Intersection between angular head velocity and speed cells. a** The bar shows the proportions of cells in the whole population. Pink: cells coding only for AHV (11.3%); Red: cells coding only for speed (13.5%); Purple: cells coding for speed and AHV (5.8%); Grey: cells neither coding for AHV nor speed. **b** The scatterplots show the intersection between AHV and speed scores in the population. Plot on the left: unidirectional – UniDir – AHV: i.e., CCW-AHV and CW-AHV. Plot on the right: bidirectional – BiDir– AHV. Colour code as in (**a**). Dotted lines represent region-averaged classification thresholds. **c** Examples of two kind of conjunctive AHV X speed cells. Plots show the firing rate as a function of the angular velocity (pink) or speed (red). Left panel: speed x UniDir-AHV cell. Right panel: speed x BiDir-AHV cell.

higher percentages of HD and lower grid modulation in the general population (HD: 67%; grid: 9.1%; grid-by-HD: 4.1%). This tendency was also observed among the AHV and speed cell populations (AHV cell population: HD: 74.5%; grid: 4%; grid-by-HD: 3.5%, speed cell population: HD: 56.7%; grid: 10%; grid-by-HD: 2.7%). Regardless of the method used, we observed all possible types of conjunction of code including AHV-by-HD (the hidden layer of directional CAN models) and grid-by-HD-by-speed (the hidden layer of positional CAN models). These results contrasted with previously published studies of self-motion coding in the superficial layers of the MEC, that were either reporting a predominantly non-conjunctive code in speed cells when using a Pearson score[17] or a sparse and uniquely conjunctive AHV code when using a GLM approach[21]. Our analyses revealed a more balanced picture in which the majority of self-motion cells are conjunctive while 20% remains non-conjunctive.

To test whether the higher number of conjunctive speed modulated cells and higher proportions of AHV cells in our database could be explained by regional variations and/or an over-representation of MEC LII cells in previous reports, we compared the percentage of conjunctive cells across all layers of MEC, presubiculum and parasubiculum. We found similar percentage of conjunctive cells in all layers (Supplementary Fig. 5a–c), except for MEC LII, where the percentage of cells coding conjunctively for a primary and a derivative signal (3%) was significantly lower than the one in the general population (18%, proportion $z$-test, $p$ value < 0.05). The scores (grid, HD, AHV and linear speed score) were mostly independent from each other, and we did not observe any significant correlation between them, except for a relatively small correlation between speed and bidirectional AHV scores (Pearson $r = 0.29$, $p$ value < 0.001) and a small anticorrelation between speed and HD scores (Pearson $r = -0.14$, $p$ value < 0.001). Such distribution of mixed selectivity is compatible with a simple hypothesis of independent assignment of each of the coding properties in the general population: cells coding for different behavioural features neither segregate, nor cluster together. This was true for all conjunction combinations and all layers, except for PrS deep layers and MEC LIII, in which an under representation of speed x HD cells was found (binomial test, $p$ value < 0.05 for MEC LIII, $p$ value < 0.01 for PrS deep). Both within-cell shuffle thresholding and GLM methods resulted in similar homogenous distribution of conjunctive coding across all layers (KS tests yield $p$ values < 0.001). Furthermore, the within-cell shuffle thresholding confirmed the virtual absence of HD or AHV coding in MEC LII, while the GLM method detected a very small number of AHV modulated cells in MEC LII (5 cells), often in conjunction with grid coding.

That self-motion information is integrated at the unit level in all cell types and all tested layers (with the notable exception of MEC LII) is at odd with current CAN model and calls for their adjustment.

**Derivative signals seemed encoded differently than primary signals**. To grasp whether derivative signals (i.e., speed and AHV modulation) were encoded in a similar fashion as primary signals (i.e., position and direction modulation), we compared the firing properties of each class of neurons. We observed that cells coding for derivative signals exhibited higher average firing rates than cells coding for primary signals (Mann–Whitney $u$-test: $p$ value < 0.001; Fig. 5a). They also showed a shorter average inter-spike interval ($t$-test: $p$ value < 0.001; Fig. 5c) and a larger peak firing (defined as the fifth quintile of the rate distribution, Mann–Whitney $u$-test: $p$ value < 0.001; Fig. 5b). It is important to notice that, contrary to what would be expected from CAN models, unidirectional AHV cells were not silent when the rats were not turning their head. Instead, they decrease their rate in response to movement in one direction and increase it in the other. The differences in firing properties between primary and derivative neurons could be explained by the fact that the monotonic firing profiles, often encoding motion signals, are less sparse than the receptive-field coding of grid and HD cells which are largely silent outside their firing field. To test this hypothesis, we calculated the percentage of the correlate values at which each cell fired more than its average firing rate (see Methods and Fig. 5d). Derivative cells showed a significantly larger percentage (mean: 61.3%, std: 15%) than primary cells (mean: 29.8%, std: 9.4%; Mann–Whitney $u$-test: $p$ value < 0.001). All analyses to compare the firing properties of primary and derivative cells were done on so-called 'pure' cells (i.e., non-conjunctive) in order to have a meaningful comparison between non-overlapping populations. Yet, all the reported effects remained when we included conjunctive cells to our analyses. Furthermore, the use of within-cell shuffle thresholding or GLM methods did not impact any of these results.

To further characterise how derivative signals are encoded in the parahippocampal region, we fitted the rate-response tuning curve of both AHV- and speed modulated neurons with either a linear or a sigmoid function (see Methods). Regardless of the shuffling method, the majority of AHV cells (68 %) were better described by a linear fit, compared to a sigmoidal fit (Supplementary Fig. 4a–c). As the steepness parameter of the sigmoidal fits was generally low (mean: 0.47 $(rad/s)^{-1}$, std: 0.16 $(rad/s)^{-1}$), we concluded that most AHV cells followed a quasi-

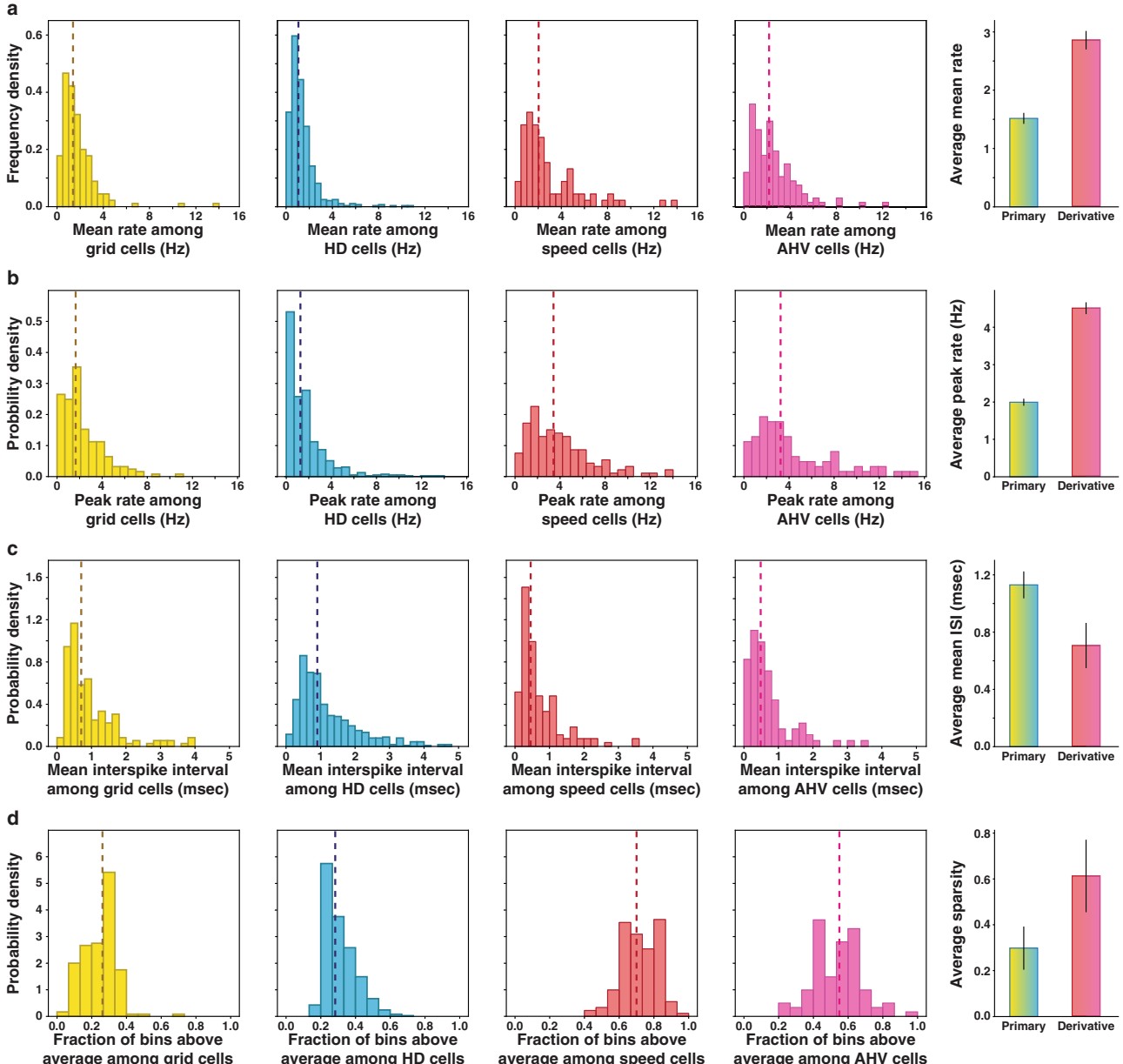

**Fig. 5 Primary and derivative correlates have different firing properties.** Comparison of firing properties between cells coding for primary correlates – grid cells (yellow) and HD cells (blue) – and derivative correlates – speed cells (red) and AHV cells (pink). Vertical dashed lines indicate median values. Histograms on the left depict primary and derivative averages for each firing property considered. **a** Distribution of the average firing rates (bin width: 0.5 Hz). Medians: grid cells 1.38, HD cells 1.06, speed cells 2.18 Hz, AHV cells 2.01 Hz. **b** Distribution of the peak firing rates calculated as the 5th percentile of the firing rate distribution of each cell (bin width: 1 Hz). Medians: grid cells 1.65 Hz, HD cells 1.25 Hz, speed cells 3.4 Hz, AHV cells 3.25 Hz. **c** Distribution of the average inter-spike interval (bin width: 0.1 s). Medians: grid cells 0.71 s, HD cells 0.9 s, speed cells 0.45 s, AHV cells 0.48 s. **d** Distribution of the proportion of tuning curve bin values above the average firing rate (bin width: 0.05, $N = 839$). Medians ($+/-$ SEM): grid cells 0.26, HD cells 0.28, speed cells 0.7, AHV cells 0.55. Note that motion cells show a larger average and peak firing rate, as well as a lower average inter-spike interval. This difference may be related to the nature of the coding: static cells use 'place like', sparse coding, while motion cells have a monotonic, dense response profile.

linear rate function. In contrast with the AHV population, the sigmoidal fit with low steepness was slightly more predominant among speed cells (56%). While this result fits with previous reports[37], we suspect that the rate saturation observed in our data at high speed may be due to the low sampling in that speed band (Supplementary Fig. 4a–c). Regardless, given the low steepness observed among sigmoid fits (mean: 0.05 cm/s$^{-1}$, std: 0.025 cm/s$^{-1}$), we concluded that the majority of so-called speed cells followed a quasi-linear rate function. The use of a within-cell shuffle thresholding method did not impact any of these results.

**Velocity coding is independent of theta modulation.** Because the theta rhythm of the local field potential has historically been strongly associated to running speed, we investigated whether both AHV and speed cells had their activity modulated by theta. Following previous work[36], we defined that a cell is theta modulated when its mean spectral power around the peak in the 5–11 Hz range was at least fivefold greater than the average spectral power in the 0–125 Hz range (see Methods). We observed that only 40% of the AHV and the speed cells passed these criteria for theta modulation (Fig. 6a), while many of the

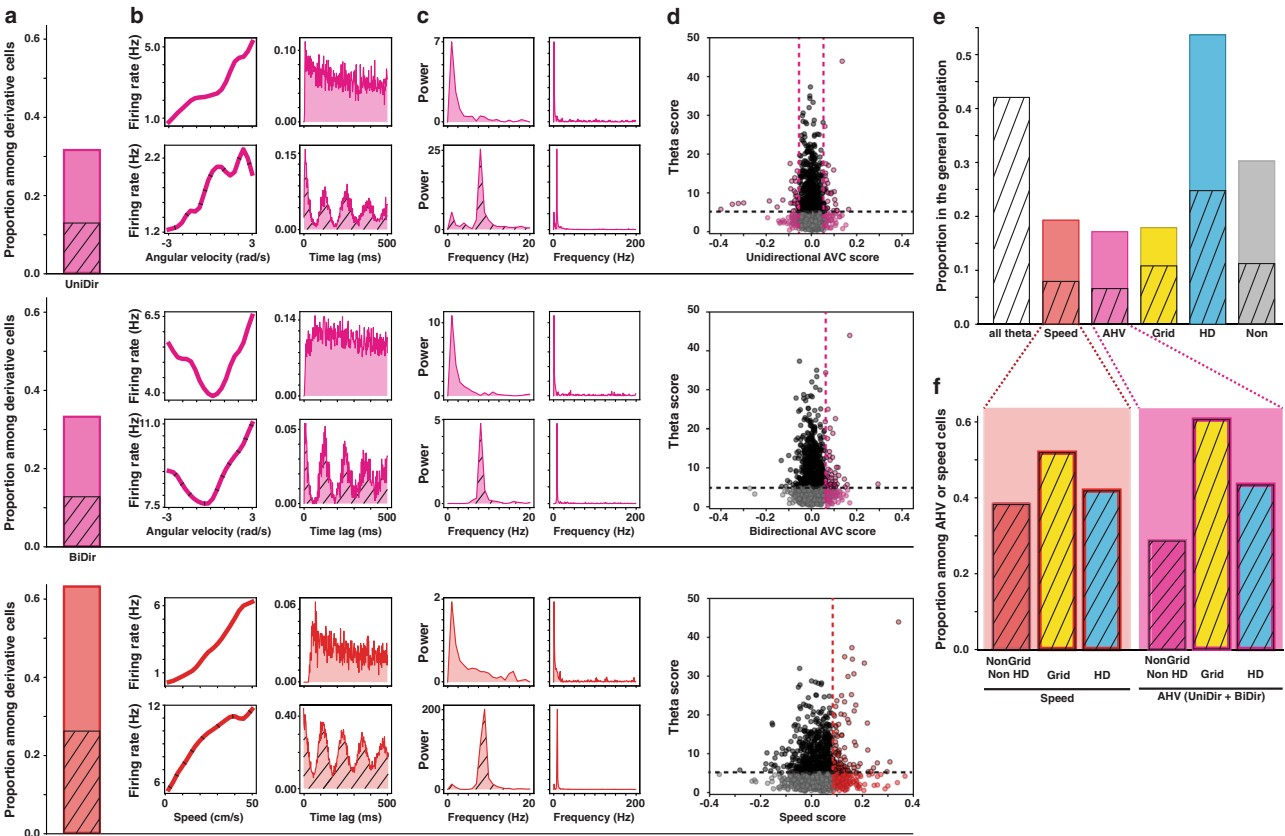

**Fig. 6 Speed and angular velocity coding are independent from theta modulation. a** Proportions of unidirectional AHV cells (first row, 31.1%, $n = 137$), bidirectional AHV cells (second row, 32.9%, $n = 145$) and speed cells (third row, 62.9%, $n = 277$) within the population of derivative cells. Dashed bars represent the fraction of theta modulated cells. UniDir-AHV: 12.5%, $n = 55$; BiDir-AHV: 12.2%, $n = 54$; speed: 25.9%, $n = 114$. **b** Examples of theta modulated (bottom row of each section) and non-modulated (top row each section) derivative cells. The first column shows the firing rate as a function of the angular velocity or speed, the second column shows the time autocorrelogram of the firing rate of the cell. **c** Power spectrum of cells in (**b**). First column: power in the range 0–20 Hz; second column: full spectrum (0–200 Hz) (**d**) Scatterplots of the distribution of theta scores and unidirectional AHV scores (first row), bidirectional AHV scores (second row) and speed scores (third row) in the population. Full black dots represent cell that are theta-modulated but not modulated by AHV or speed; Full coloured dots represent cells modulated by AHV (pink) or speed (red), but not by theta; coloured dots with black contour are cells modulated by both theta and AHV (pink) or speed (red); grey dots represent unclassified cells. **e** Distribution of speed cells (red, 19.3%), AHV cells (pink, 17.1%), grid cells (yellow, 17.8%), HD cells (blue, 53.3%) and unclassified cells (grey, 30.0%, $n = 432$) in the whole population. Dashed bars represent the proportions of theta modulated cells in the whole population (white shaded, 42.1%), and in conjunction with speed cells (8.0%), AHV cells (6.6%), grid cells (10.1%), HD cells (24.6%) and unclassified cells (11.1%). **f** Proportions of theta modulated cells among speed cells (left, red background) and AHV cells (right, pink background). Cells are divided by type: 'pure speed' (red, 38.2%), 'pure AHV' (pink, 28.6%), conjunctive grid x speed (yellow left, 51.9%), conjunctive grid x AHV (yellow right, 60.6%), conjunctive HD x speed (blue left, 41.8%) and conjunctive HD x AHV (blue right, 43.4%).

remaining cells did not show any modulation by theta (Fig. 6b–c). This agrees with previous reports of non-theta modulated speed cells in the MEC[37]. The theta modulation of AHV/speed cells was observed in comparable proportions to those observed in the general population (Fig. 6e). There was no significant correlation between AHV score and theta score, or between speed score and theta score (Pearson correlation: $p$ value > 0.05). Conjunctive coding for grid or head direction did not influence the proportions of velocity cells that were theta modulated (Fig. 6f, proportion $z$-test: $p$ value < 0.05). Theta modulation was uniformly distributed across all layers of each area except for MEC LII, which showed more theta modulation and MEC LVI, which showed less (Supplementary Fig. 6a, proportion $z$-test: $p$ value < 0.01). The proportion of velocity cells theta modulated in each layer did not differ from what was expected based on the theta modulation observed in the general population, except for MEC LVI speed cells, which showed less theta modulation than expected (Supplementary Fig. 6b–c, proportion $z$-test: $p$ <

0.05). Neither single cell shuffle thresholding methods nor GLM classification significantly impacted these results. Together, these results suggest that the code for self-motion in the para-hippocampal region seems largely independent of theta modulation at the single cell level.

## Discussion

Here, we reveal the existence of a network of parahippocampal principal neurons whose activity is modulated by angular and linear self-motion signals. Extensive mapping of the rat para-hippocampal region showed that this network was spread homogenously across all layers of several interconnected areas upstream of the hippocampal formation: the medial entorhinal cortex (MEC), the presubiculum and the parasubiculum. We observed that some self-motion neurons seemed to only respond to either angular head velocity or linear speed (i.e., an apparent non-conjunctive code), while a larger proportion was also responding to spatial and/or directional information (i.e., a

conjunctive code). Such integration at the unit level may be a crucial mechanism underlying the representation and the updating of position (place and grid cells) and direction (head direction cells) in the hippocampal/parahippocampal circuits.

Our results contribute to a fertile line of research devoted to understanding how space and movement in space are encoded in the hippocampal region. Speed had so far been investigated as a modulating factor of firing rate within grid and place cells[38], or as a non-conjunctive code in the superficial layers of the MEC[17,37]. Here, we systematically investigated movement, spatial, and direction correlates in all layers of multiple parahippocampal regions, without pre-selecting specific cell types. Running a comprehensive unbiased analysis with complementary methodological approaches allowed us to investigate features previously inaccessible – see Supplementary Fig. 3 for extended discussion and comparison between methods. Specifically, we discovered an almost complete lack of differentiation across brain regions or selectivity types. Contrary to previous studies describing an overabundance of conjunctive code[16,21,38], we observed that many self-motion cells did not respond to either spatial or directional signals. We hypothesise that such studies may have depicted an incomplete picture by limiting their investigations to grid cells or to the superficial layers of the MEC. Our analyses revealed that, in fact, MEC layer II computations are very much at odds with those of all the other parahippocampal sub-areas.

Given that linear and angular speed are the derivative in time of position and direction, respectively, we propose that our results may uncover a general algorithm for the updating of any type of information. In support of the idea of a general parahippocampal mechanism that could compute the derivative in time of other correlates, we demonstrated that angular and linear self-motion (derivative) signals were encoded in a different manner with respect to positional and directional (primary) signals, this regardless of the scoring method used. It is well documented that head direction, grid, and place cells tend to be active only when an individual is either in a given position or with its head in a given direction. In contrast, we showed that only a very small proportion of self-motion neurons responded preferentially for a given speed (8%). Most presented a relatively high baseline activity that was linearly (or quasi-linearly) ramped up in response to increasing speed. This could be the hallmark of a general strategy in the parahippocampal region for the neural coding of scalar quantities whose magnitude has a well-defined meaning – speed and angular velocity are, for instance, set by physical constraints on how fast the animal can move – as opposed to neural activity manifolds used to encode position and direction, where coordinates are relative to a reference frame. Since position and orientation are far from the only 'primary' signals encoded in these regions, we wonder: could we find a similar phenomenon for the encoding – or updating – of other signals such as the position in a cognitive space[39] or the representation of a tune frequency[40]?

Previously, angular head velocity (AHV) cells had been mainly characterised upstream of the hippocampal circuits, in less-integrated subcortical structures linked to the processing of vestibular information (e.g., lateral mammillary nuclei, dorsal tegmental nucleus, thalamic nuclei and striatum)[22–25,41,42] and in the retrosplenial cortex[26,27]. Besides, a few hippocampal neurons modulated by whole body motion had been reported in the primate[43]. AHV has been shown to influence the preferred orientation, pitch or azimuth of some presubicular head direction cells, as well as to modulate the firing rate of a few presubicular interneurons[44]. A very recent report showed coding for head pitch and roll in the superficial layers of the MEC, as well as a sparse conjunctive AHV code. Conversely, we report here a large proportion of neurons modulated by both bilateral and unilateral (i.e., CW and CCW) AHV and distributed over the presubiculum, the parasubiculum and the MEC. We hypothesise that such widespread signals were not previously reported either because of a restricted scope in analyses or because of recordings clustered in the most dorsolateral presubiculum or in the superficial MEC. To avoid such bias, we recorded from a much larger population spread across the medio-lateral and antero-posterior axis. Nevertheless, we found no topographic organisation, except for MEC layer II. Such lack of topography near the top of the cortical hierarchy is consistent with AHV being one of potentially several high-level signals, which could be extracted by derivation with respect to time[45]. In line with this idea, we observed lower correlation scores than those reported for subcortical AHV cells[22]. We hypothesise that – contrary to subcortical areas, which could be viewed as specialised to signal movement – parahippocampal areas are high-level cortical structures, integrating self-motion information with other factors, only a few of which we can test for. As such, we expect that each correlate only explains a fraction of the variance of the firing rate, leading to smaller scores.

Since the discovery of grid cells, many have attempted to understand how such a strikingly regular signal could be generated by individual neurons[13,28,46–49]. A speed code is central to much of this theoretical work, and a break-through was the characterisation of speed cells and speed modulation in the MEC[16,17,37,38,50,51]. Here, we report for the first-time speed cells in the pre- and parasubiculum. Interestingly, recent experimental work has demonstrated that grid cell activity is dependent on the integrity of the speed signal[52], which itself seems driven by the brainstem locomotor circuit[53]. Likewise, the stability in head direction coding seems dependent on AHV[54] and vestibular inputs[23,55]. Therefore, self-motion signals could be similarly involved in the generation and the maintenance of both position and direction signals.

Our findings offer robust experimental evidence particularly relevant for the reappraisal of theories describing navigation based on grid and head direction cells interacting in continuous attractor neural networks (CAN). CAN are popular theoretical models of how reciprocally interconnected neurons may extract, refine and sustain stable representations of continuous behavioural variables. This function is particularly crucial when afferent signals are weak, noisy and/or intermittent. How can such representations be updated? A common view is that neurons coding for 'primary' navigational variables – such as position and direction – are connected by a so-called 'hidden layer' of cells conjunctively coding for position, direction and self-motion. This conjunctive layer is postulated to drive the activity of the output layer to remain congruent, at any time, with the external world. Although several variants of these mechanisms have been proposed, in general they posit a categorical division of labour between the output units representing the instantaneous variables and the hidden units updating them[28,56]. Recent studies in drosophila melanogaster have shown that a ring attractor network with local excitation and global inhibition underlies the representation of head direction[34,35]. It is possible that similar mechanisms operate in mammals as hinted by a recent report of toroidal topology of population activity in grid cells[57]. Yet, it is unclear whether a mammalian brain, highly plastic and possibly lacking topography, could accommodate a neatly laid out and hard-wired ring attractor such as the one in the fly brain.

Here, we report the existence of cells conjunctively coding for position, direction and self-motion not just in MEC[16,21,38], but also in pre- and parasubiculum. Our findings can be interpreted as clear evidence of the existence of hidden layers in the parahippocampal region, yet they also challenge some of the postulates currently assumed by CAN models, which favour specific connectivity rules and segregated coding[13]. First, the continuous

– rather than bimodal – distribution of degrees of selectivity to any correlate is incongruous with a structured connectivity. Indeed, if we consider a simple HD ring attractor, it relies on three cell population: a principal attractor of HD cells and two 'steering' populations – or hidden layers – of conjunctive HD-by-AHV cells, in the CW and CCW directions, respectively. While anatomical intermingling is not completely proscribed here, structure maintenance would require delicate fine tuning, such as a systematic offset between input and output connections to each of the steering populations. Furthermore, it is problematic that we found a large majority of self-motion cells that are not silent in the absence of movement, especially given that two steering populations would have to constantly level their inputs for the activity bump to remain still. Such interactions have – so far – not been successfully included in functioning CAN models. We also observed a considerable amount of non-conjunctive, bidirectional and anticorrelated self-motion cells whose role is unaccounted for. Finally, our data cast doubts on the notion of distinct attractors, one for position and one for direction. Under which conditions CANs may operate when a given conjunctive cell belong to two attractors is an interesting open question.

We emphasise that distinct types of selectivity, not only appeared randomly admixed, but also failed to present clear-cut categorical distinctions between conjunctive and non-conjunctive cells. We observed an absence of correlation between scores (i.e., grid, HD, AHV and speed score) compatible with a scenario of independent assignment of coding properties. As such, the probability to observe, for example, a cell conjunctively coding for HD and CCW-AHV in region X is given by the product of the probabilities of coding for HD and for AHV, irrespective of X. These results are more consistent with un-organised or perhaps self-organising models than with precisely engineered ones[49,58] and agrees with recent evidence of mixed selectivity in the hippocampal formation[59]. One exception to this independence was the absence of AHV and HD cells in MEC LII, a layer that also shows a higher proportion of spatially modulated cells. This could suggest that the AHV signal is needed locally, among the same cells coding for the primary signal it serves to update. However, one should note that HD cells are present in large number in the mouse MEC LII[60]. Given the unlikelihood that spatial coding would follow different computational principles in related species like the mouse and the rat, such differences may be linked to divergences in connectivity. Moreover, areas known to be involved in the coding for movement – such as the peri- and postrhinal cortices, the retrosplenial cortex, the cerebellum, the thalamus and the septum[61–63] – all project similarly, either directly (or indirectly in the case of the cerebellum) to the MEC, the PrS and the PaS[5,64–68]. Further connectivity analyses are needed to specifically assess the differences in connectivity strength and relative influence of these inputs on hippocampal computations.

Historically, running speed has been reported to show a strong correlation with the amplitude of theta oscillations recorded in the local field potential (LFP) of freely behaving rodents[18–20]. Likewise, many place and grid cells exhibit a strong modulation of their firing rate following those theta oscillations, either in a phase-locked or in a phase-precessing manner[69–71]. In line with these observations, many models point to theta oscillations as inherent to the generation of the grid signal[72]. Among them, the oscillatory interference-based models assume a velocity input to the grid network composed of translational speed and movement direction[46,48]. Here, we show that only 40% of self-motion cells show a strong theta-modulation, a percentage compatible with independent assignment of theta- and self-motion modulation. This apparent decoupling may seem surprising, since it suggests a scaled back role of the theta frequency in locomotion. However,

the role of theta in spatial coding had already been recently challenged by several lines of evidence. One is that modulation of the septal oscillatory activity had no apparent consequence on grid signal maintenance[73], therefore suggesting that non-theta septum correlates – such as attention – were involved in the grid cell signal disruption observed after septal inactivation[74,75]. Another is that no stable theta oscillation has been recorded in the hippocampal LFP of bats, who do exhibit both place and grid coding[76]. Even though recent results showed that bats seem to exhibit theta-band modulation of grid firing and matching phase precession when considering a non-stable theta frequency[77], it is clear that a robust spatial code can be built without regular oscillatory activity. It was recently proposed that theta may have a relationship to self-motion different in the LFP from the individual cell level[37]. Here, our results suggest a weak relationship at the individual cell level, but do not conclude as to the LFP. Of specific interest is a recent study showing that theta modulation – both at the LFP and individual cell level – is more closely associated with acceleration than speed[78]. Further targeted interventional studies would be essential to dissect the ambiguous role of theta in the generation and maintenance of positional, directional and self-motion signals[79]. Discriminating between the theta and non-theta modulated self-motion neurons, revealed by our work, would be a chief target for this type of studies.

In conclusion, we provide clear evidence of a widespread parahippocampal network involved in linear and angular speed coding that could have a crucial role in the updating of the cognitive map, or perhaps be part of the map itself. The existence in the hippocampal region of neurons conjunctively coding for self-motion, position and direction would *prima facie* appear to fill a gap in the framework of continuous attractor network models. Yet, our comprehensive analysis reveals an apparent lack of organisation, calling for the revision of such models so that they can (i) express dynamical intertwined continuous attractors and (ii) account for the apparent random nature of the spatial code, as well as its peculiar lack of a clear organisation. We hypothesise that derivative algorithms may have a generalised role in the updating of continuously varying information, not just of a spatial nature. Further studies, with either targeted inactivation of neurons or testing of non-spatial correlates will be necessary to establish whether one of the main roles of the parahippocampal region is to ensure the accurate updating of the hippocampal representation.

## Methods

**Subject and surgeries**. All the data presented here have been previously published[36] but was re-analysed for this manuscript. The neuronal activity was recorded from 28 male Long-Evans rats (3–5 months old, 350–450 g at implantation, housed and food deprived as in described previously). All experiments were approved by the National Animal Research Authority of Norway. Tetrode configuration and surgical implantations are described in previously published work[30].

**Data acquisition and training procedures**. General data acquisition procedures have been described previously[36]. In brief, rats were trained to collect food crumbs thrown randomly into a 50-cm-high square or circular box with black floor and black walls surrounded by black curtains. Each trial lasted 10 min. Behaviour was relatively uniform between sessions and animals[36].

**Spike sorting**. The spike detection in the local field potential and sorting were performed as previously described[36]. Spike sorting was performed offline using graphical cluster-cutting software.

**Estimation of the behavioural correlates**

*Position*. The position of the animal was estimated from the coordinates of two light-emitting diodes (LEDs) on the head of the animal. The X and Y coordinates of both the LEDs where smoothed with a gaussian filter with a 250 ms standard deviation, chosen to match the smoothing performed on the firing rate (see below), and the average between the two LED positions was used as the position of the animal. The data previously published[36] included some sessions with only one

LED. These sessions were excluded from the current dataset – hence a discrepancy of neurons reported between the studies.

*Head Direction.* The head direction (HD) was calculated as the angle between the line connecting the small LED to the big LED and the $x$ axis. HD is expressed in radians, 0 meaning that the rat head is aligned with the $x$-axis, facing right.

*Linear speed and angular head velocity.* Speed was calculated as the modulus of the vector difference between the smoothed position at time t and the position at time $t_{+1}$. Angular head velocity was calculated as the signed difference between the head direction at time t and the head direction at time $t_{+1}$.

The absolute value of the angular head velocity was used for the scoring of bidirectional angular head velocity cells (see below). No further smoothing was applied.

**Firing rate calculation.** Instantaneous firing rate was obtained dividing the whole session in bins of 20 ms, coinciding with the frames of the tracking cameras. The spike count in each time bin was then calculated and divided by the temporal width to obtain the rate. The rate profile was smoothed with a 250 ms wide Gaussian filter.

**Speed filtering.** The analysis on speed and angular velocity was performed on movement periods, defined as the ones in which the animal speed was >2 cm/s. A speed filter was applied on the timeseries of each correlate, discarding the time points for which the instantaneous speed was below 2 cm/s, that were excluded in the subsequent analysis.

**Rate maps and tuning curves**
*Spatial rate maps.* The histograms for spike count and time spent in each location were constructed using equally spaced bins of 2 cm linear size. Each bin of the rate map was obtained as the ratio between spike count and time spent, smoothed with a Gaussian filter with standard deviation of 4 cm.

*Directional rate maps.* The histograms for spike count and time spent facing each direction were constructed using equally spaced bins of size 6 degrees. Each bin of the rate map was obtained as the ratio between spike count and time spent, smoothed with a Gaussian filter with standard deviation of 6 degrees.

*Speed and angular velocity tuning curves.* For tuning curve construction, the correlate was divided in equally spaced bins. For speed, 20 bins spanned the range between 2- and 50 cm/s (bin width: 2.4 cm/s), for angular velocity the range −3-, +3-rad/s was again divided into 20 bins (bin width: 0.15 rad/s). The firing rate in each bin was calculated as the average of the instantaneous firing rate values falling in the each given bin. A gaussian smoothing window with standard deviation 0.15 rad/s for angular velocity and 2.4 cm/s for speed was applied.

**Shuffling.** Chance-level statistics was calculated for a given variable W through a shuffling procedure. For each repetition, the firing rate time series was time shifted of a random interval of at least 30 s, with the end of the trial wrapped to the beginning.

In the region-wise shuffling, this procedure was repeated 100 times for each cell, and the shuffled score for variable W was calculated for each instance to compose the chance-level statistics. Cells from the same regions where then pooled together to obtain the null distribution for the score values. In the within-cell shuffling, the procedure was instead repeated 1000 times for each cell, and no pooling was performed: each cells had its own associated null distribution of score values.

For cell classification, the 99th percentile of the shuffled distribution was used as a classification threshold in both procedures.

**Measure used for cell type classification**
*Speed Score.* Following Kropff et al.[17], the speed score was defined as the Pearson product-moment correlation between the cell's instantaneous firing rate and the instantaneous speed of the animal, across the whole recording session. This yields a score ranging from −1 to +1.

*Unidirectional angular velocity score.* The unidirectional angular velocity score was defined as the Pearson product-moment correlation between the cell's instantaneous firing rate and the instantaneous angular velocity of the animal. Positive values of angular velocity correspond to clockwise head movement. Cells that had a score greater than the 99th percentile of the shuffled distribution were classified as clockwise modulated (CW), while cells whose score was lower than the 1st percentile were classified as counterclockwise modulated (CCW): they significantly code for head movement in the counterclockwise direction. CW and CCW populations are mutually exclusive by construction.

*Bidirectional angular velocity score.* The bidirectional angular velocity score was defined as the Pearson product-moment correlation between the cell's

instantaneous firing rate and the absolute value of the instantaneous angular velocity of the animal. Cells in this population increase their firing rate in response to head movement regardless of the direction. The unidirectional and bidirectional angular velocity scores are not mutually exclusive by construction. A cell with a strong ramping up of the activity for positive value of AHV, and a mild sensitivity to negative values, for example, could be selected as both a bidirectional and a unidirectional CW cell. A strong modulation from both negative and positive angular velocities would be picked up only by the bidirectional score, while a strictly monotonic increase (or decrease) along the whole range of velocities would give a high unidirectional score.

*Mean vector length (head-direction score).* The mean vector length score is calculated from the head-direction tuning map of a given cell as the sum: $\left|\frac{S\lambda_i e^{i\theta_i}}{S\lambda_i}\right|$, where $\theta_i$ is the orientation in radians associated with bin $i$ and $\lambda_i$ is the firing rate in the bin. The sums run over all N directional bins, and the modulus of the resulting complex number is taken. Head direction was binned in bins of 6 degrees and smoothed with a gaussian filter with a standard deviation of 6 degrees.

*Grid score.* The grid score was calculated from the spatial autocorrelogram of a given cell with a procedure similar to[80]. After exclusion of the centre of the autocorrelogram, the Pearson correlation of the autocorrelogram rotated by 30, 60, 90, 120 and 15 degrees (+− 3 degrees offsets) was considered. Only bins closer to the centre than an outer radius s were included in the calculation of the correlation. Given s, the grid score was defined as the difference between the average of the maximum correlations around 60 and 120 degrees (+− 3 degrees offsets) and the average of the minimum correlations around 30, 90 and 150 degrees (+− 3 degrees offsets). The final grid score of the cell was then defined as the maximum grid score over values of s ranging from twenty to forty bins, computed at intervals of one bin.

*Theta index.* Theta modulation of individual cells was estimated from the frequency power spectrum of the spike-train autocorrelation histogram of the cell.

A cell was defined to be theta modulated if the mean power in a 2 Hz window centred in the peak in the 5- to 11 Hz frequency range was at least fivefold greater than the mean spectral power in the 0- to 125 Hz range.

**Estimation of the significance of overlaps between cell populations.** The observed overlaps between cell populations were compared to the ones that would result from the statistical null hypothesis of independent random assignment with a two-sided binomial test. The probability of observing an overlap of size k between two populations of sizes $N_a$ and $N_b$, independently drawn from a total number of cell N is given by

$$p(k) = \frac{N!}{k!(N-k)!}p_{ab}^k(1-p_{ab})^{N-k}$$

Where $p_{ab} = p_a x p_b$ and $p_a = N_a/N$, $p_b = N_b/N$.

**Information analysis.** The information per spike conveyed by each cell about the correlate of interest (speed or angular head velocity) was calculated using the formula:

$$I = \frac{1}{\lambda}\Sigma_i\lambda_i p_i log_2\left(\frac{\lambda_i}{\lambda}\right)$$

Where $i$ is the index of the correlate bin, $p_i$ is the probability of observing the correlate in bin $i$ (i.e., the normalised occupancy), $\lambda_i$ is the average firing rate of the cell in bin $i$, and $\lambda$ is the average firing rate of the cell.

Speed was divided in 2 cm/s bins in the range 2–50 cm/s (as in all analysis, stillness periods were excluded), while angular head velocity was divided in 0.5 rad/s bins, in the range −5–5 rad/s.

Cells were considered to carry significant information about the correlate if the observed information rate exceeded the 99th percentile threshold of the null distribution obtained by shuffling the cell firing rate values (1000 shuffles per cell).

**Generalised linear model (GLM) analysis.** We analysed the effect of each correlate (speed and angular head velocity) with a linear-nonlinear Poisson spiking GLM model.

This model assumes that the firing rate of the cell depends on the value of the correlate as

$$r(t) = exp(SiX(t)_i^T w_i)/dt$$

Where $X_i(t)$ is a one-hot vector (i.e., a vector with only one non-null element) indexing which value the correlate is taking at time $t$, and $w_i$ are the coefficients of a linear filter quantifying the contribution of each value of the correlate to the firing rate of the cells.

The model is fitted using the python module statsmodel.api, which finds the set of parameters $w_i$ maximising the log-likelihood of the observed spikes, subject to an elastic-net regularisation constraint.

To perform the fitting procedure, the speed values have been binned in 10 bins in the range 2–50 cm/s, and the angular velocity values divided in 10 bins in the range −3–3 rad/s.

Cells were considered significantly modulated by a correlate if the log-likelihood of the best fit was significantly larger than the value obtained with only the average firing rate as a predictor. Significance was estimated with a 10–fold bootstrapping procedure to extract the confidence interval of the observed log-likelihood.

**Tuning curve fitting**. Two different functional forms were fitted and compared to the tuning curve of modulated cells.

A linear model:

$$r = ax + b$$

And a sigmoid model:

$$r = \frac{1}{1 + exp(-a(x - b))}$$

Here, $x$ is the value of the correlate (speed or angular head velocity) and $r$ is the average firing rate of the cell at that value of $x$. Tuning curves were rescaled by their maximum value, in order to match the two model by number of parameters.

The R2 fitting scores were then compared for each cell. Cells with a linear R2 greater than the sigmoid R2 were classified as linear, and vice versa.

To quantify the number of cells that showed modular coding (i.e., an increased rate in a particular speed or angular velocity band), we fitted their rescaled tuning curves with a gaussian profile:

$$r = exp\left(\frac{-(x - a)2}{2b}\right)$$

Cells were labelled as gaussian if the fit yielded a R2 score >0.5 and the average a of the gaussian laid within the tuning curve interval (2–50 cm/s for speed, −3–3 rad/s for AHV).

**Sparsity calculation**. To quantify the sparsity of the firing of primary and derivative cells, we calculated their tuning curves (as described above) as a function of speed (for speed cells), of AHV (for AHV cells), head direction (for HD cells) and position (for grid cells).

We then calculated as the sparsity the percentage of the bins of the tuning curve that had a firing rate value larger than the average overall firing rate of the cell.

**Reporting summary**. Further information on research design is available in the Nature Research Reporting Summary linked to this article.

## Data availability
Data is available upon request, given that there is shared ownership of this data distributed over several laboratories. Please contact charlotte.boccara@medisin.uio.no for any requests. An answer will be given within 2 weeks of email reception. Source data are provided with this paper.

## Code availability
The code can be found at https://github.com/davidespalla/code_ahv_speed_cells, and cited as: Davide Spalla, 'Angular and linear speed cells in the parahippocampal circuits', code_ahv_speed_cells, https://doi.org/10.5281/zenodo.5834018, 2022.

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

## Acknowledgements

We thank Edvard and May-Britt Moser for consent to use these data[36]. We thank René Skukies and Arnfinn Aamodt, students in the laboratory of Johan Frederik Storm, for preliminary analyses of head direction signal. We thank Silvia Girardi for her help with the schematics illustrating rats and ring attractors in Figs. 1 and 2, as well Supplementary Figs. 1, 3 and 4. We thank Dori Derdikman, Gily Ginosar, Eleonore Duvelle and Anna Chambers for providing comments on an earlier version of the manuscript. Charlotte Boccara is supported by a MSCA Fellowship #799749 and an NFR Grant #288478.

## Author contributions

C.N.B. conceptualised the experiments. C.N.B. and D.S. conceptualised the analyses. D.S. perform the analyses with support by C.N.B. and A.T., C.N.B. and D.S. wrote the paper with feedback from A.T.

## Competing interests

The authors declare no competing interests.
