## [Peer Review File · Nature Communications]

Angular and linear speed cells in the parahippocampal circuitsEditorial Note: This manuscript has been previously reviewed at another journal that is not operating a transparent peer review scheme. This document only contains reviewer comments and rebuttal letters for versions considered at Nature Communications.

REVIEWER COMMENTS

Reviewer #1 (Remarks to the Author):

General summary

Spalla et al. analysed a dataset from a previous study (Boccaro et al. <https://doi.org/10.1038/nn.2602>) that includes single units recorded in the medial entorhinal cortex (mEC), parasubiculum (PaS) and presubiculum (PrS). They report grid and head direction (HD) cells as well as cells encoding self-motion parameters such as angular head velocity (AHV) and speed. Importantly, they also found conjunctive representations of these different parameters (i.e. AHV x HD or grid x HD) which are thought to play an important role in a popular class of spatial model known as a continuous attractor network (CAN). All cell types seem to be equally likely to be conjunctive for the different self-motion parameters and the brain regions did not differ so much in this regard (except for mEC LII).

The paper is generally well written and the data are beautifully presented. The findings are informative in advancing our understanding of CANs and are likely to be of interest to experimental, theoretical and computational fields.

The original Boccaro et al. study reported conjunctive grid x HD cells in mEC, pre- and parasubiculum and this study extends that analysis to include self-motion parameters. Previous research has already reported conjunctive grid x HD x speed <https://doi.org/10.1126/science.1125572> and conjunctive position x HD x speed <https://doi.org/10.1038/s41467-021-20936-8> cells in the mEC, so these findings are not extremely novel. However, the authors extend these results to the pre- and parasubiculum where these cells have not been previously reported - speed cells have also not been previously reported in these regions.

I found the analysis quite restrictive, to the point of unfortunately undermining the main conclusions (see major comments) and I found the discussion confusing at times. The main conclusions are that conjunctive cells necessary for CANs can be found in parahippocampal regions but these cells display features that may be incompatible with CANs. The data fully support the former claim but I don't understand the latter argument and I'm not sure it is supported by the analyses. The authors want to make strong claims regarding CANs but these are often made without enough explanation or reference to specific models. I think many of the claims would require additional modelling effort to support.

Major points

1. Within-cell shuffles more accurately reflect chance level than population-wide ones (<http://dx.doi.org/10.7554/eLife.21354.001>). For instance, chance directional tuning will depend on firing rates (which vary between cells) and behavioural sampling (which varies between sessions) which the current approach cannot account for. The authors do not report behavioural statistics for rats or sessions so we do not know if it is appropriate to pool shuffles across these. Recent studies looking at novel, conjunctive spatial and self-motion encoding have mainly shifted towards using generalized linear

models (i.e. <https://doi.org/10.1016/j.neuron.2017.03.025>, <https://doi.org/10.1038/s41467-021-20936-8>, <https://doi.org/10.1126/sciadv.aaz2322>, <https://doi.org/10.1101/2020.10.03.324848>, <https://doi.org/10.1126/science.aax4192>) that also employ within-cell shuffling or cross-validation to determine significance. These approaches can also quantify the variance explained by each parameter and correct for covariation between them. The authors either need to update their analysis (also see below) or provide arguments in support of their approach.

2. The authors utilise a linear analysis instead of a more generalized one because of the ‘dominance of linear coding’ to ‘facilitate the comparison with linear speed analyses’ and to allow ‘a more direct comparison with previous literature’. I’m not convinced by these arguments. Recent research has shown that a considerable fraction (> 50%) of speed cells have nonlinear tuning curves (Hinman et al. 2016), something the authors also find and report. Similarly, AHV, HD and speed cells have all been shown to exhibit nonlinear tuning curves (Hardcastle et al. 2017; Mallory et al. 2021). Hardcastle et al. (2017) compared their linear-nonlinear analysis to the approach used by the authors and found that it missed ~60% of informative cells – especially in superficial layers of the mEC, which the authors indeed found lacking in self-motion coding. The restrictive nature of the analysis continues because the authors do not include all types of linear responses such as negatively sloped bidirectional AHV cells (~9% of AHV cells; Mallory et al. 2021) and negatively sloped speed cells (~16% of speed cells; Hinman et al. 2016). As far as I can tell linearly asymmetric and symmetric cells are not differentiated in the analysis. This approach leaves large gaps in the authors data that need to be addressed with different analyses or a discussion of the limits of the existing analysis with adjusted claims.

3. The authors claim that they provide “the first recording in the hippocampal region of the conjunctive coding postulated to be found in the “hidden layer” of CAN models”. This doesn't seem to be true because conjunctive grid x HD x speed <https://doi.org/10.1126/science.1125572> and conjunctive position x HD x speed <https://doi.org/10.1038/s41467-021-20936-8> cells have been reported in the mEC. It is also a little misleading because conjunctive HD x speed and HD x AHV cells were found a long time ago in DTN where the HD signal is thought to originate (i.e. where we think the HD CAN is actually likely to reside) <https://doi.org/10.1523/JNEUROSCI.21-15-05740.2001>. There are other omissions such as not pointing out that Hinman et al. 2016 <http://dx.doi.org/10.1016/j.neuron.2016.06.027> also found sigmoidal speed coding in speed cells or that many cells in fact have nonlinear tuning curves (Hardcastle et al. 2017; Mallory et al. 2021). These findings are foundational to the study so I am surprised they are omitted or not discussed more fully.

4. I had problems with the discussion that are quite general, but I want to list them here because they are included in the abstract and are important to the overall paper. The authors argue that 1) they have revealed the ‘hidden layer’ necessary for CAN models but 2) their conjunctive cells lack a “structured organisation” and 3) that they have “unexpected” features which call into question CAN models. It’s not really clear to me what organisation or features the authors mean; I think it’s that the cell types and conjunctive coding combinations are mixed? Why is that a problem? It is also unclear what organisation they would expect instead. There is no real explanation of this given, or examples of CAN models specifically. Ironically, I think nonlinear self-motion signals do pose a significant problem for CANs but these are mainly excluded from the analysis. There is a suggestion that cells coding ‘derivative’ signals could be used in non-spatial representations, this is brought up in the last couple of sentences of the paper and mentioned in the abstract despite there being no evidence for this or previous discussion. What do the authors mean? This needs further explanation.

Minor points

Introduction

1. Please include line numbers in future submissions (I received a version with line numbers after completing my review). Due to multiple readings some of my comments may be out of order – I have tried to add some context using text from the manuscript.
2. Overall, I thought the introduction was very clear and well written and only had some minor remarks. I will refer to Mallory et al. 2021 (<https://doi.org/10.1038/s41467-021-20936-8>) in some places. I appreciate that the authors may have submitted their manuscript before that paper was available.
3. “...an (apparently) self-standing...”
What is the purpose of ‘apparently’ here? If the authors feel that speed cell modulation is not self-standing that might warrant further explanation.
4. “...in principal neurons of the hippocampal region.”
Hippocampal region is ambiguous – it could mean hippocampus (hippocampus proper) or the hippocampal formation, which some consider to include the Sub and EC. Thus, angular velocity coding has been observed in the hippocampal formation but not the hippocampus.
5. “Most reports come from recordings of subcortical structures (e.g., lateral mammillary nuclei, dorsal tegmental nucleus), linked to the processing of vestibular information”
Also postsubiculum and retrosplenial cortex – these are cited later but I don’t know why not here:
<https://dx.doi.org/10.1177%2F2398212817721859> and <https://doi.org/10.1101/2021.01.22.427789>
6. “Most crucially, it has remained unclear whether angular velocity coding reflects an intelligent design.”
From the encyclopaedia Britannica:
Intelligent design (ID): argument intended to demonstrate that living organisms were created in more or less their present forms by an “intelligent designer.”
Is this really what the authors mean?
7. “Exciting evidence compatible with such CAN models was recently provided by investigations of the *Drosophila melanogaster* central complex...”
Also see <https://doi.org/10.1101/2021.02.25.432776> which may have been published after this paper entered review.
8. “To understand the circuit mechanism by which spatial representations can be updated in mammal, we...”
In mammals?
9. “...it appears that direction, position and speed selectivity are randomly admixed with each other...”
Why is this a problem for CANs? There needs to be some better explanation of this because it seems to be key for understanding the discussion.

Results

10. Generally, I liked the results, the figures are clear and very well presented. Most of my issues stem from the choice of analysis rather than the proceeding tests or figures.

11. Fig 1d: Some of the bars have multiple colours (i.e. mEC CCW has two colours) what do these correspond to?

12. Fig 1e: Some of the cells have very low AHV scores (i.e. <0.10) and generally it seems all scores were low (<0.20). Could the authors comment on this? Is it because the cells are conjunctive and AHV only explains a low fraction of the cell's firing? How do these scores compare to cells in vestibular regions?

13. Fig 1f: for the plot showing the mEC unidirectional data, the dashed line showing the lower percentile (CCW cutoff) is quite different to the CW one and there seem to be hardly any shuffles lower than it. Is this line plotted in the correct place?

14. Fig 1f: are the pink data curves and grey shuffle histograms calculated in the same way? i.e. does the pink curve just trace a histogram made with the same settings as the shuffle distribution? If not it is a bit hard to compare the two distributions. Also, what does 'frequency density' mean? There must be 100x as many shuffle data points so how are these normalised? Usually the y-axis would show fraction of total.

15. Fig 1F: There are regions of negative bidirectional angular velocity score that look to significantly exceed the shuffles, at least in PaS and PrS but maybe not in mEC (which would be an interesting finding). Why are only positive BiDir correlations included? You could have a lower percentile cutoff as you do with UniDir coding.

16. Fig 1H: I would rather see just the significantly modulated AHV cell scores. Perhaps the non-significant cells could be in another colour with the significant ones plotted on top? Also calling the plot a 'correlation' might be a bit confusing here – the correlation would be calculated on the points, and could be represented by a line, but the points themselves don't show 'a correlation'. Same points also for Fig 2F.

17. Fig SF2 and other figures: you do not need to repeat numbers (i.e. proportions in legend) that are plotted in a graph. These numbers inflate the figure legends and obscure more useful information. I would remove them throughout but this is up to the authors.

18. Fig SF2: how do these t-tests work on single values for each group?

19. Fig SF4c and more generally: a "Skaggs score" is mentioned in the results and described in the methods but I'm not really sure how this was used. Was it used? As far as I understand it the 'GLM class.' bars in SF4c refer to GLM results and the 'rate score' bars refer to the correlation results? If so where are the Skaggs score results? Either way I don't think these labels are very clear. If the Skaggs score is not shown or used it should just be removed from the paper.

20. Fig SF4c: The 'rate score' bars are separated into CW, CCW and BiDir, what are the other sections in between?

21. "As per our definition, AHV cells are neurons whose firing rate is positively modulated by angular velocity"

This seems like a very restrictive definition given that nonlinear-asymmetric and negatively sloped AHV cells exist (although they are admittedly less common). <https://doi.org/10.1038/s41467-021-20936-8>
Maybe the authors want to extend this analysis to include non-linear representations.

22. “velocity values were treated as categorical variables; therefore, no linear dependence was implied”
Is this correct? I’ve never checked but it seems that AHV and running speed are likely to be correlated – i.e. if a rat is running in an arc it will also have a high AHV?

23. “Out of 182 AHV cells solely picked up by the GLM method, about half of them (54%) were anticorrelated with the absolute value of angular velocity”
Are the tuning curves of these cells convincing? This seems like a significant population of informative cells that are being excluded. Also, why does this sentence highlight the anticorrelated cells specifically? Is the point that many anticorrelated cells were missed by the correlation approach? Or that more of the cells solely identified by the GLM were anticorrelated than we would expect?

24. “In addition, we observed that rectilinear speed signals...”
As far as I know speed cells haven’t been reported in pre- or parasubiculum before, so this result is quite interesting and might warrant some more discussion (i.e. do these structures differ from mEC in terms of velocity inputs?). Is there a reason the proportions might be higher in pre/parasubiculum? Are the proportions significantly higher?

25. “Speed cells were uniformly distributed across all layers in each area...”
How was a KS test run on these data – they are just single proportions?

26. “significantly overlapped with those obtained with our linear scoring method (binomial test, pvalue < 0.001”
Is the p-value for the GLM or Skaggs score? Again, where is the Skaggs score used/reported?

27. “In order to allow for comparison with the latest reports of speed cells in the MEC, we decided to use the more conservative Pearson scoring methods for all further analyses.”
This isn’t really true as it has been recently (within last 5 years) shown that about 33% of speed cells are best fit by a saturating exponential and are poorly fit by a linear model (<https://doi.org/10.1016/j.neuron.2016.06.027>) or that they are just nonlinear (<https://doi.org/10.1016/j.neuron.2017.03.025>) which is in agreement with your GLM results (and later sigmoid results). That paper also points out that non-linear speed coding presents some problems for attractor network models, which seems directly relevant to the current manuscript.

28. Fig. 2d: as with AHV there are regions of negative speed score that look to significantly exceed the shuffles in all 3 regions. We know negative speed cells exist so why are they excluded?

29. “These percentages were similar to the percentages observed in the general population”
This is quite a nice observation, I think it would be good to have a test here (Chi square test maybe) comparing the proportions.

30. “We observed all possible types of conjunction...”
Because this paper is closely linked to CAN models it might warrant spending some more time on this. On my first readthrough I wasn’t sure if you had looked at conjunctions of more than 2 parameters. Compare this to Figure 1c&d in Mallory et al. (2021; <https://doi.org/10.1038/s41467-021-20936-8>)

where we can see the breakdown of major groups up to 4 conjunctive parameters. Looking at your Fig SF5 (much harder to find) we can see, for example, that there are very few grid x HD x speed cells, but isn't this conjunction important to CANs? Why are there so few? Why are there comparatively more grid x AHV cells, which don't really fill an important computational function? Maybe these are what you mean later in the discussion when you talk about the 'features' of conjunctive cells not being what you expect; if so that needs to be much clearer.

31. "This result contrasted significantly with previously published studies of a predominantly self-standing code in speed cells recorded in the MEC superficial layers"

Hinman et al. (2016; <http://dx.doi.org/10.1016/j.neuron.2016.06.027>) report grid x HD x speed modulation in the mEC:

"...the speed modulation of firing rates of MEC neurons was positive ... for each individual cell type ... (... conjunctive, $p < 0.0005$...)"

As do Sargolini et al. (2006; <https://doi.org/10.1126/science.1125572>). These should be cited here.

32. "The scores (grid, HD, AHV and linear speed score) were mostly independent from each other"

I don't understand this analysis. The previous paragraph argues that conjunctive coding is represented across all primary cell types. As I suggested above, I think a better way to show this would be to compare the proportions observed across all cells and those observed in AHV cells (for example) or use a test looking at the intersections like the one used for the overlap of GLM and correlation results. The reason I'm not really convinced by the correlation approach is that there are far fewer derivative (speed, AHV) coding cells than primary ones (grid, HD), so we wouldn't necessarily expect a correlation between these parameters in the overall population. The authors want to make some strong claims about CAN models based on this information, so I think it is worth spending some time to make this analysis more convincing. If I am misunderstanding then maybe a figure is necessary?

33. "That self-motion information is integrated at the unit level in all cell types and all tested layers (with the notable exception of MEC LII) is incompatible with current CAN model and calls for their revision."

In what way does this finding preclude existing models? This statement needs a citation or some more explanation.

34. "It is important to notice that, contrary to what would be expected from CAN models, unidirectional AHV cells were not silent when the rats were not turning their head...."

We can't know this from the information given in Fig SF6 or the results. The authors would need to find the y-intercept of the linear or sigmoidal fits on the speed cell tuning curves and show that these are $>0\text{Hz}$. Although they have excluded negative speed cells and BiDir AHV cells which would be the ones best supporting their argument. However, if the baseline rates are $>0\text{Hz}$, why does this matter for CAN models? It seems trivial for the network to subtract baseline rates from each cell (or just use an averaged population speed code). Can the authors explain this a bit better with citations?

35. "To test this hypothesis, we calculated the percentage of the correlate values at which each cell fired more than its average firing rate (see Methods and extended data Fig 6d)."

This was difficult to find in the methods as it is under the heading "Sparsity calculation". The heading or the main text could be clearer. More generally, I didn't have any specific issues with this paragraph, but I'm not entirely sure what the conclusion is. We don't know if the cells have the same waveforms or if they are inhibitory/excitatory, which seems more important for trying to understand their firing dynamics.

36. "...could be explained by the fact that the monotonic firing profiles used to encode motion signals is less sparse..."

Didn't the authors largely exclude non-monotonic derivative signals when they chose to rely on the linear correlational approach for classifying cells? I don't think these kinds of conclusion can be made given the current analysis.

37. Figure SF3: this figure seems to be out of place, shouldn't it be later? I really like this figure by the way, and Fig SF1, mainly because they show lots of informative examples. I would like to see the examples at the bottom of Fig 3 extended to show more cells as well. Especially because this is the only place HD and grid cells are shown.

38. "...the sigmoidal fit with low steepness was slightly more predominant among speed cells (56%). This result seemed to be due to a saturation observed at high speeds..."

Hinman et al. (2016; <http://dx.doi.org/10.1016/j.neuron.2016.06.027>) also report:

"...the majority [of speed cells] were actually better fit using a saturating exponential ... function rather than a linear function ... [145 of 260] ..."

Which is about 56% or almost identical to the results reported here. Hinman is cited later for something else but should be credited here especially because this is a remarkably close replication.

39. "We thus concluded that most speed cells also followed a quasi-linear rate function."

This feels like a circular argument because the analysis for classifying cells was linear. Again, I don't think the authors can make these conclusions.

40. Fig 5: I like the detail and clarity of this figure, but is it normal that the x-axis of the left-hand plots extend underneath everything else? I would also recommend a pattern other than diagonal hatching as it creates an optical illusion. i.e. for me the HD bar in Fig 5e looks like it 'bends' where the hatching ends. Also see Fig SF7. This isn't scientifically important, but it is a little distracting.

41. Fig 5f: my understanding is that these plots show the proportion of each cell type that is both speed (or AHV) modulated AND theta modulated? i.e. the proportion of grid cells that are speed and theta modulated is around 0.39 (yellow hatched bar on red background). But the legend says: "conjunctive grid x speed (yellow left, 51.9%)" which does not correspond to the bar. None of the legend values seem to match the bars, so, what do the graphs mean?

42. In some graphs (i.e. Fig 3a&c, Fig 4a, Fig 5 and Fig SF7) the y-axes are labelled as 'frequency' while it is supposed to say 'proportion'. These words have different meanings. In some cases the main text says proportion while the figure says frequency: i.e. "the proportions of velocity cells that were theta modulated (Fig. 5f ...)" but the y-axis of 5f and its legend say frequency.

43. "...the proportions of velocity cells that were theta modulated (Fig. 5f, t-test: pvalue < 0.05)"

"...except for MEC LII which showed more theta modulation and MEC LVI which showed less (extended data Fig. 7a, t-test: pvalue < 0.01)"

How are these t-tests conducted? There is a single value representing a proportion and a chance value, I don't see how you can run a t-test?

What chance level was used for fig. 5f? the proportions add up to more than 1 (because cells can be in more than one group).

What chance level was used for Fig. 7? The legend says it was the value "expected given the average theta modulation in that specific layer" but I'm not sure how that would be calculated.

These test results need to include degrees of freedom and test-statistics and ideally a measure of effect size (same for all other stats like the KS test reported elsewhere).

44. "Velocity coding is independent of theta modulation"

Why is there no mention/citation of Hinman et al. (2016; <http://dx.doi.org/10.1016/j.neuron.2016.06.027>)? It seems really odd that this is omitted considering the whole paper is about the topic of this paragraph.

Discussion

46. "...the first recording in the hippocampal region of the conjunctive coding postulated to be found in the "hidden layer" of CAN models". Conjunctive grid x HD x speed

<https://doi.org/10.1126/science.1125572> and conjunctive position x HD x speed

<https://doi.org/10.1038/s41467-021-20936-8> cells have been reported in the mEC previously. Although for the latter conjunctive HD x speed and HD x AHV cells in DTN are more likely linked to the generation of the HD signal <https://doi.org/10.1523/JNEUROSCI.21-15-05740.2001>. Why are these citations omitted?

47. Are there any models that require all speed and AHV cells to conjunctively code for position and head direction? Could you add a citation? This seems like a straw man argument.

48. "Ultimately, they challenge the very concept of a well-defined representation of the instantaneous variables."

I'm not sure what this sentence means.

49. "In terms of attractor structures, they support extending CAN models to incorporate dynamical variables in what their output "represents", for example including units firing at a baseline rate for zero AHV, and quiescent for high either CW or CCW AHV."

What does 'dynamical variables' mean? I also don't understand the last argument; the example described either sounds like a negatively bidirectional AHV cell or a linearly asymmetrical one. But the authors' analyses excluded all of the former and as far as I can tell they never distinguish between linearly asymmetrical and unidirectional tuning. So it seems strange that the authors single these out as important for future models.

50. "... only a negligible proportion of self-motion neurons responded preferentially for a given speed. The vast majority [...] linearly (or quasi-linearly) ramped up in response to increasing speed."

Is 8% negligible? A lot of cells were also detected by the GLM that seemed to have non-linear tuning curves, which is in agreement with current reports that as many as 50% of speed cells may have a nonlinear tuning curve (Hardcastle et al. <https://doi.org/10.1016/j.neuron.2017.03.025>). Taking this into account the conclusion that derivative cells are therefore distinct from non-derivative ones is less convincing.

51. "...we demonstrated that both angular and linear self-motion (derivative) signals were encoded in a different manner..."

'Both' is redundant in this sentence. The fact that many cells are conjunctive seems to undermine this argument: how differently can the encoding be for derivative and primary signals when they exist in the same cells? I think the main argument has to do with baseline firing rates – the fact that derivative cells tend to have higher baseline rates? As I said in an earlier comment I don't think the authors actually

show this. But now I'm wondering how this works in conjunctive cells, Fig SF6 doesn't seem to differentiate conjunctive cell types, i.e. do grid x speed cells have a higher baseline firing rate than pure grid cells? If that was the case the authors' argument would be better founded. More generally, are grid x speed cells just grid cells with added speed coding (i.e. in-field firing is speed modulated) or are they speed cells with added grid modulation (i.e. firing everywhere in a speed modulated manner but with spatial peaks in a grid)?

52. "It is possible that similar mechanisms operate in mammals."

Would be good to cite Gardner et al. <https://doi.org/10.1101/2021.02.25.432776> here.

53. "...but also did not present evidence of clear-cut categorical distinctions between conjunctive and non-conjunctive cells."

This was also explored in detail by Hardcastle et al. <https://doi.org/10.1016/j.neuron.2017.03.025> and Mallory et al. <https://doi.org/10.1038/s41467-021-20936-8>

54. "One exception to this observation was the absence of AHV and HD cells in MEC LII..."

Mallory et al. also looked at this closely and suggest that this coding is underestimated in superficial layers due to nonlinear tuning curves (their section: Mixed Selectivity in Superficial Medial Entorhinal Neurons). Is it possible that a similar explanation might apply here? Could AHV and HD both be represented nonlinearly in superficial layers for some reason?

55. "Historically, running speed has been reported..."

I found this paragraph quite confusing. Why is it surprising that only a minority of AHV and speed cells were theta modulated? I'm not sure this matters at all for the models you describe. As far as I can see the Zutshi et al. citation doesn't say anything about medial septum and grid cells, maybe you meant something like Carpenter et al. <https://doi.org/10.1038/s41598-017-15100-6>? Even so, there are other reasons to think grid cells require the linear velocity modulation of theta (i.e. Winter et al. <http://dx.doi.org/10.1016/j.cub.2015.08.034>).

I'm not sure what point you are making with the bat results, do they undermine the importance of theta or not? I think you have to at least cite Barry et al. <https://doi.org/10.1038/nature11276> when discussing this. Why does investigating the theta modulation of self-motion neurons help with any of these questions? Hinman et al. <http://dx.doi.org/10.1016/j.neuron.2016.06.027> have even shown that the speed-theta frequency relationship (which may or may not affect grid cells) and speed cell signal (which I guess hasn't been tested) are dissociable, so I'm not really sure what the conclusion is here.

56. "In conclusion, we provide clear evidence ..."

See earlier comment about previous studies showing conjunctive cells in mEC and DTN.

57. "Yet, the analysis of these neurons reveals features divergent from those expected of units serving solely to update the instantaneous representation of static variables."

I'm really not sure what features this sentence refers to. I think this is a general issue with the clarity of the discussion though.

58. "Our work urges for the revision of such models so that they can (i) express dynamical continuous attractors and (ii) account for the apparent random nature of the spatial code, as well as its peculiar lack of a clear organization."

What does 'dynamical continuous attractor' mean? Why is the spatial code 'random'? What is the issue with the organisation – is it that cells were not very organised in layers or that features were not

organised between cells? We would expect both pure and conjunctive cells, both of which were observed and other models predict each type in all layers, so where is the issue?

59. “We hypothesize that derivative algorithms may have a generalized role in the updating of continuously varying information, not just of a spatial nature.”

What evidence is this based on? This is completely out of the paper’s scope.

Methods

60. Section: “Bidirectional angular velocity score”

The authors could instead calculate the CW correlation and CCW correlations separately and then take the maximum value as the AHV score. This would detect uni- and bi-directional cells as well as linearly-asymmetrical ones.

i.e. <https://doi.org/10.1523/JNEUROSCI.21-15-05740.2001>, and <https://dx.doi.org/10.1177%2F2398212817721859>

61. Sections: “Shuffling” and “Speed Score”

Should Kropff et al. (2015; <https://doi.org/10.1038/nature14622>) be cited here? The text is identical to that paper. If other methods/analyses are taken from previous papers it would be beneficial to cite them. Not just for recognition but it also strengthens the current manuscript if those methods have already been published.

62. Spike sorting: somewhere around here it would be good to know what is included in the dataset, i.e. are all clusters putative cells? Only excitatory or pyramidal? Biased towards HD and grid cells? The bars in Fig 3 seem to suggest a strong bias towards spatial cell types. I can see the citation to the previous paper but some info is important enough to warrant mentioning in the manuscript directly. Some basic information about the recording protocol would also be good to know as well as some information about HD and AHV behaviour – i.e. did all the animals show the same distribution of AHV values? Or speed? This is especially important since you want to use a population-wide shuffle; if one rat had much slower running speed than the rest it would not be correct to combine its shuffles with the others (for example). See Figure 1b in Mallory et al. (2021; <https://doi.org/10.1038/s41467-021-20936-8>)(sorry to keep referring to that paper but it is very relevant to the current manuscript and it is still fresh in my mind from recent reading).

Reviewer #2 (Remarks to the Author):

This paper reports an analysis of a previously collected data set of 1436 neurons recorded from medial entorhinal cortex (MEC), presubiculum and parasubiculum as rats freely explored a variety of environments. Head position and direction were measured using two LEDs mounted on the head, and used to reconstruct position and linear and angular head velocity as well as place and head direction correlates and grid patterning. It was found that around 17% neurons showed firing that correlated with angular head velocity (AHV), a similar proportion with linear velocity, and there was a high degree of conjunctive encoding between the “primary” variables (position and head direction) and the

“derivative” ones (linear speed and angular head velocity). It is concluded that the results “offer insights as to how linear/angular speed ... may allow the updating of spatial representations.”

I think these are interesting observations speaking to an important issue and they are beautifully presented. However I have concerns about the analyses, and about some apparent contradictions with previous recent findings in the literature. It also isn't clear to me that these findings add new insights about the computation of position, as conjunctive encoding has been reported in numerous areas and much discussed over many years.

The biggest issue is that despite claims that “angular velocity coding has not been yet established in principal neurons of the hippocampal region” and “conjunctive coding had not been found in the hippocampal region”, in fact a recent paper from the Giacomo lab (Mallory et al <https://doi.org/10.1038/s41467-021-20936-8>) reports both of these phenomena in MEC. Conjunctive encoding of position and linear velocity has also been reported some time ago in MEC (Sargolini et al). It's possible the authors were unaware of this quite recent study, but it does contain some notable differences from the present findings that it would be good to understand. For one thing, Mallory et al used a linear-nonlinear method to characterize the tuning to angular head velocity and found that indeed many neurons were tuned to specific speed bands, whereas the current study found linear tuning. In the present study the authors tried methods other than Pearson's correlation “to reduce the dependence of our results on a unique scoring method,” found that the other methods disagreed somewhat and so reverted to the Pearson's correlation on the grounds that it was the most conservative measure. This logic did not make sense to me (why even try the other analyses in this case?) and raises the possibility that their analysis missed non-linear tuning. I think we need a better understanding of the degree to which the present results disagree with those of Mallory et al., as well as of the similarities. It is suggested to consider applying the methods of Mallory et al to these data, although this is a big task. Indeed, a thorough analysis/comparison of the different methods employed and a discussion of their strengths and weaknesses would be useful, given that these types of conjunctive analyses, and how to choose among them, are an issue for many in the field.

Another issue is that I don't fully understand where the data came from and how they were selected. The data were recorded in a study that has been previously published, but in that study there were a total of 654 cells from seven rats in the presubiculum, 528 cells from a different 7 rats in parasubiculum and 630 cells from 15 rats from yet another previous study, in MEC. It wasn't clear that the recording environments and rat behaviors were equal across these different experiments. Also, the numbers do not match those in the present study so some selection has occurred – what was the basis for this selection? Was it purely on cluster quality or were firing patterns also considered (in particular, was there a focus on grid cells, which was the interest of the previous study). I'm not opposed in principle to analysing data from previous experiments but since they were originally collected, selected and analysed with a different purpose in mind, one needs to be especially careful to avoid accidental biases, and to be very clear about the provenance of the data.

On the plus side, the observations from pre- and parasubiculum *are* novel, and the absence of angular head velocity tuning in layer II MEC is intriguing. The analysis of the primary vs. derivative signals is also interesting and novel, and indeed could usefully be moved from the supplementary section to the main paper.

To sum up: I think more work needs to be done to highlight what is novel, relate the findings here to previous in the literature, and to develop a little more (including in the abstract) the theoretical insights these data afford.

Minor comments

Abstract: “These self-motion neurons often conjunctively encoded position and/or direction” – the conjunction with the velocity signals is the more interesting thing here

P2 “Intelligent design” is conventionally used to mean “God” – I think a preferable term is “functional” or perhaps “optimal”

P3 “general self-organizing derivative algorithm” is too vague at this early stage and needs explanation

The absence of AHV encoding in MEC layer II is intriguing – any speculations as to what this could mean, or is it just sampling variability?

Why was the analysis restricted to positively modulated firing rates – might there not have been cells inhibited by AHV?

P4 “with naturally the exception...” needs explaining as readers won’t necessarily remember that there weren’t any AHV-modulated neurons there.

Concerning the two additional methods of analysis – the results from the information measure (Skaggs) weren’t reported that I could see. As mentioned, these analyses are critical and should be a major part of a revised manuscript. I found it odd that the GLM found 54% of cells anti-correlated with AHV – I feel this shouldn’t be brushed off but rather investigated and explained. What happens when Mallory et al’s methods are applied to these data?

P6 t-test needs degrees of freedom and the t statistic (here and elsewhere). What was being compared in this primary/derivative conjunction? I was expecting just a chi-square test.

In what way is the integration of self-motion information at the unit level incompatible with CAN models?

“Self-standing” is a new and unintuitive terminology.

I found the discussion interesting but a little rambling and could be tightened a little, and explain a little more clearly why the findings constrain some of the CAN models.

Fig. 2e I’m not clear on the meaning of “instantaneous linear speed” for this AHV cell.

Fig. SF6 is missing part (d) in the legend

Reviewer #3 (Remarks to the Author):

Understanding how the brain encodes information about experience in the activity of distributed populations of neurons is arguably one of the most challenging goals of modern neuroscience. Many insights about high-level neural coding have come from studying the brain's representation of space and its neural correlates in the entorhinal-hippocampal network. Here, the activity of individual neurons is modulated by variables related to the animal's movement in space, such as the animal's location, the direction of its head, the speed of its movements, or the presence of object or boundaries on its path. Computational models predict that spatial tuning might emerge from the integration of information about the environment and the animal's movement in it, and the concerted activity of neurons that are locally connected according to rigid connectivity rules within continuous attractor networks (CAN). Specifically, CAN models hypothesize that populations of neurons whose activity is exclusively modulated by position or direction might receive inputs from cells exhibiting a mixed selectivity for a repertoire of variables associated with the position, direction, and self-motion.

In this study, Spalla, Treves, and Boccara investigate the neuronal basis of self-motion coding in the entorhinal cortex, presubiculum, and parasubiculum – three areas of the entorhinal-hippocampal network where the senior author of the current study has previously recorded neurons exhibiting spatially-modulated firing (Boccara et al., *Nat Neurosci* 2010). Through a thorough re-analysis of a previously published dataset, the authors report (i) the identification of neurons whose firing is modulated by the linear and angular speed at which the animal is moving; (ii) the mixed-selectivity exhibited by the neurons that encode such signals; and (iii) the anatomically-distributed nature of the neuronal network processing linear and angular velocity.

While the presence of signals related to self-motion has been predicted by CAN models since their inception, proofs of their existence and their relation with the other spatial variables encoded within the entorhinal cortex and hippocampus had, until recently, been lacking. As such, this study represents an advancement in the current knowledge of the repertoire of spatially-modulated signals that can be recorded in the extended hippocampal network. Its main strength lies in the identification of the distributed nature of the self-motion signals, which seem to spread across multiple layers and multiple areas of the entorhinal-hippocampal network. Moreover, since such signals had previously been hypothesized to be essential for the computations of CANs and the emergence of spatially-modulated firing patterns, the present study confirms a long-standing prediction within the field and proposes a very interesting idea (i.e., the distinction between coding of direct variables and their derivatives) that calls for further investigation. The fact that the same dataset has now been used for multiple studies is a testament to the excellence and rigorousness of the experiments analyzed. The statistical methods deployed are appropriate and adequately described.

I have only minor comments regard the formulation of the manuscript: once these will be addressed I will fully support it for publication in *Nature Communications*.

1. In a recent study, it has been reported that entorhinal cortex neurons are variably tuned to self-motion signals, including linear and angular velocity (Mallory et al., *Nat Commun.* 2021 Jan 28;12(1):671.). The existence of such study clearly affects the formulations of the present manuscript, and forces the authors to tone-down some of their statements, like “angular velocity has not yet been established in principal neurons of the hippocampal region”, or “... providing the first recording in the hippocampal region of the conjunctive coding...”. Also, not citing the Mallory study in the current manuscript would be an unforgivable oversight: the authors should go further and put some thoughts into how the two studies fit together and report this in the main text.

2. On page 2, the authors write: “it has remained unclear whether angular velocity coding reflects an intelligent design”. This sentence should be re-written taking into consideration that “intelligent design” is an expression that is charged with meaning that most likely goes beyond what the authors meant to express. If instead, their use of the expression is to be framed in the context of a theological discussion about the origin of life, then this point should definitely be explored further in the text.
3. On page 6, the authors state that “the fact that self-motion information is integrated at the unit level in all cell types and tested layers is incompatible with current CAN models and calls for revision”. Why such finding is incompatible with current CAN models is not clear to me. First of all, not all neurons recorded exhibit conjunctive coding (see fig 3), so this dataset is still being compatible with the idea that only the hidden layer should exhibit conjunctive coding to support the “pure” tuning of the other units of the model. Second, I am not aware that the CAN model is strictly associated with an exclusive correspondence between the layers of the attractor network and the layers of the MEC. Why would the current model not work if units with conjunctive coding and units with pure coding were to be anatomically intermingled, provided that their connectivity would still match the one predicted in CAN models?
4. The authors state that velocity coding is independent of theta modulation. How do the authors reconcile this finding with the recent finding that theta rhythm is not linearly modulated by speed as it was previously thought, and that its frequency is modulated by acceleration (E Kropff et al, Neuron, 2021)? Is it possible that such modulation would lead to an overestimation of AHV tuning in the theta-modulated neurons?
5. Last, the authors seem to draw some conclusions from the absence of AHV and HD cells in MEC LII, which, in their view, would suggest that “AHV signal is needed locally among the same cells coding for the primary signal it serves to update”. It is now a consolidated notion that the anatomical distribution of spatially modulated cell types is species-specific: even though HD cells seem not to be present in MEC LII in the rat (see the present study, but also Sargolini et al., Science 2006), they are abundant in the mouse MEC LII (see Rowland et al., 2018). Given that it would not be reasonable to hypothesize that spatial coding follows different computational principles in related species like the mouse and the rat, it remains to be determined if the anatomical distribution of functional cell types bears any information at all about the computational function that these cell types subserve.

**REVIEWER COMMENTS**
**Reviewer #1 (Remarks to the Author):**
General summary (R1)
Spalla et al. analysed a dataset from a previous study (Boccaro et al. <https://doi.org/10.1038/nn.2602>)
that includes single units recorded in the medial entorhinal cortex (mEC), parasubiculum (PaS) and
presubiculum (PrS). They report grid and head direction (HD) cells as well as cells encoding self-motion
parameters such as angular head velocity (AHV) and speed. Importantly, they also found conjunctive
representations of these different parameters (i.e. AHV x HD or grid x HD) which are thought to play
an important role in a popular class of spatial model known as a continuous attractor network (CAN).
All cell types seem to be equally likely to be conjunctive for the different self-motion parameters and
the brain regions did not differ so much in this regard (except for mEC LII).
The paper is generally well written and the data are beautifully presented. The findings are informative
in advancing our understanding of CANs and are likely to be of interest to experimental, theoretical and
computational fields.
*We thank the reviewer for this positive opinion of our work.*
The original Boccaro et al. study reported conjunctive grid x HD cells in mEC, pre- and parasubiculum
and this study extends that analysis to include self-motion parameters. Previous research has already
reported conjunctive grid x HD x speed <https://doi.org/10.1126/science.1125572> and conjunctive
position x HD x speed <https://doi.org/10.1038/s41467-021-20936-8> cells in the mEC, so these findings
are not extremely novel.
However, the authors extend these results to the pre- and parasubiculum where these cells have not been
previously reported - speed cells have also not been previously reported in these regions.
*As the reviewer hypothesised later, our manuscript was indeed submitted before the publication of*
*Mallory et al. We have of course adjusted both the text and the analysis to acknowledge this study. Had*
*we been aware of the Giacomo lab decision to add angular head velocity analyses to their study, we*
*would have coordinated with them to submit this article as a proper back-to-back. However, we trust*
*that our findings, contrasting with studies such as theirs or Sargolini et al., limited to MEC superficial*
*layers, will provide the readers with the necessary broader understanding of how primary (position,*
*direction) and derivative (self-motion) codes can be differently integrated at the unit level in different*
*sub-circuits of the parahippocampal network. As it is highlighted by all the reviewers, such results had*
*not been previously reported. We think this novel evidence is relevant for the understanding of the*
*function of the circuit responsible for the representation of speed, especially because it highlights its*
*widespread and distributed nature.*
*Specifically, our careful mapping of linear and angular signals in both the deep and superficial layers*
*of the presubicular, parasubicular and medial entorhinal cortices has shown that previous studies*
*provide an incomplete picture of self-motion coding by mostly limiting their analyses to the superficial*
*layers of the MEC. In this manuscript, we demonstrated that, in fact, MEC layer II computations are*
*very much at odds with those of all other parahippocampal sub-areas, accounting for the small number*
*of angular head velocity cells reported in Mallory et al. By providing more in-depth analyses beside*
*the GLM and a much wider dataset, our results complement and contrast this interesting report.*
*With respect to Sargolini 2006, we obviously referred to their seminal work. We however would like to*
*draw the attention of the reviewers to the fact that our study contrary to Sargolini et al (i) considers*
*both angular and rectilinear velocity (while their study was limited to linear speed), (ii) is not limited*
*to MEC and (iii) our approach to detection of speed (and AHV) modulation is different: we look at the*
*correlation between the time series of the firing rate of each cell and the instantaneous speed (AHV) so*
*that we can build a robust statistical null model with the shuffling procedure, and use it to quantify*
*significance. In Sargolini 2006, the approach was not concerned with finding significantly modulated*
*cells, but by the modulation of the firing rate of grid/HD cells. As such they only looks at the distribution*
*of correlation values between speed and firing rate of grid/HD cells. We believe that our approach*
*affords a better understanding of self-motion coding per se, enriching the current knowledge of the*
*phenomenon. In fact, the picture that emerges from our study on the relationship between grid/HD*
*coding and self-motion coding is very different: only a fraction of cells code conjunctively for any given*
*collection of correlates, while in Sargolini 2006 141/150 grid cells and 153/220 HD cells are shown to*
*have a positive (though not necessarily larger than a null model) correlation.*
I found the analysis quite restrictive, to the point of unfortunately undermining the main conclusions
(see major comments) and I found the discussion confusing at times. The main conclusions are that
conjunctive cells necessary for CANs can be found in parahippocampal regions but these cells display
features that may be incompatible with CANs. The data fully support the former claim but I don't
understand the latter argument and I'm not sure it is supported by the analyses. The authors want to
make strong claims regarding CANs but these are often made without enough explanation or reference
to specific models. I think many of the claims would require additional modelling effort to support.
*Based on the reviewer suggestions, we have proceeded to more extensive analyses, as well as point of*
*comparison with previously published studies. In addition, we have now both mitigated and clarify our*
*claims related to the incompatibility of our findings with CANs throughout the main text (introduction,*
*result and discussion)– see detailed answer in major comment.*
Major points (R1)
1. Within-cell shuffles more accurately reflect chance level than population-wide ones
(<http://dx.doi.org/10.7554/eLife.21354.001>). For instance, chance directional tuning will depend on
firing rates (which vary between cells) and behavioural sampling (which varies between sessions) which
the current approach cannot account for. The authors do not report behavioural statistics for rats or
sessions so we do not know if it appropriate to pool shuffles across these.
*Within-cell vs. region-wise shuffle: We have now re-analyse our complete data-set using “within cell*
*shuffle” or what we called “single cell shuffle” thresholding methods. Based on this re-evaluation our*
*main conclusions remain the same - see below for specifics.*
*We originally omitted this method based on behavioural analyses that showed quite homogeneous self-*
*motion behaviour between sessions and rate between cells. Furthermore, we wanted to provide analyses*
*that could allow an easy comparison with Boccara 2010.*
*However, we agree with the reviewer that such additional analyses are providing a much more complete*
*picture of coding in our dataset. Furthermore, while there was no difference as to “derivative” signals,*
*it highlighted some differences in the percentages of “primary” signals (i.e., grid and HD). These*
*changes are now highlighted in the text. Yet, we would like to stress to the reviewer that such differences*
*in grid and HD proportions in the whole population did not affect any of our main conclusions on self-*
*motion modulation. Indeed, proportions changed in a similar fashion within the general population and*
*within the self-motion population.*
Recent studies looking at novel, conjunctive spatial and self-motion encoding have mainly shifted
towards using generalized linear models
(i.e. <https://doi.org/10.1016/j.neuron.2017.03.025>, [https://doi.org/10.1038/s41467-021-20936-](https://doi.org/10.1038/s41467-021-20936-8)
[8, https://doi.org/10.1126/sciadv.aaz2322](https://doi.org/10.1126/sciadv.aaz2322), <https://doi.org/10.1101/2020.10.03.324848>, [https://doi.org/](https://doi.org/10.1126/science.aax4192)
[10.1126/science.aax4192](https://doi.org/10.1126/science.aax4192)) that also employ within-cell shuffling or cross-validation to determine
significance.
*GLM approach: In our original text we already had provided some points of comparison with GLM*
*analyses but opted to not extend this approach to all our analyses to preserve points of comparison with*
*both Kropf et al and Boccara et al. from an analyse point of view, our rational was that cross-validation*
*and bootstrapping methods, both sound and effective methods for building statistical null models, are*
*equivalent in nature to our shuffling procedure, and equally sound. In particular our method is non-*
*parametric, low variance and does not rely in hyperparameter tuning (see below).*
*However, (i) given the recent publication from Mallory et al., (ii) to answer the reviewer legitimate*
*concerns and (iii) to offer a more extensive (and interesting) comparison between methods, we re-*
*performed all our analysis using a GLM approach to detect speed and AHV modulation. The main*
*results presented in the paper remained, regardless of the method used. The only exception is about*
*MEC LII, on which we commented in the main text.*
*Given the similarity between the results and in order not to overcomplicate the reading flow of our*
*manuscript, we chose to only present the “Pearson results” in the main figures and show the*
*(equivalent) “GLM results” in the supplementary figures (while referring to those in the main text).*
*Nevertheless, we are thankful to the reviewers for pointing at this appropriate comparison. This*
*prompted us to add a section in the discussion adjacent to supplementary fig 3 to debate pros and cons*
*of each method, as well as justify our primary use of the Pearson correlation method as follow:*
*(i) The Pearson correlation method, combined with a region-wise shuffle, affords a very large statistical*
*power (pooling the number of shuffles allows to have a statistical null distribution with hundreds of*
*thousands of data points), that can be very conservative in the selection of cell. Indeed, this is the method*
*that yields the lower number of modulated cells. We believe that the very low rate of false positives that*
*this method guarantees is a strength in the interpretation of the significance and reproducibility of the*
*results.*
*(ii) The Pearson method has a lower variance than GLM (its hypothesis class is much smaller than the*
*one of GLMs), which avoid the pitfalls of data overfitting. The method is also non-parametric: it does*
*not require a choice of bins and bin size and no choice of regularization hyperparameters and*
*procedure. This makes the results more easily interpretable, reproducible and generalizable.*
*(iii) Our analysis yields a direct measure of explained variance in the Pearson score, whose absolute*
*value is directly interpretable as the fraction of variability of the firing that is explained by the correlate.*
*We check variable covariation by looking at the correlation of different score, which we find to be*
*absent of or mild.*
*These approaches can also quantify the variance explained by each parameter and*
*correct for covariation between them. The authors either need to update their analysis (also see below)*
*or provide arguments in support of their approach.*
*As mentioned above, we checked for the interaction between different properties by looking at the*
*correlation between the different scores, of which we find none or mild. This is reported in the main*
*text page 6 and 7.*
**2.** *The authors utilise a linear analysis instead of a more generalized one because of the ‘dominance of*
*linear coding’ to ‘facilitate the comparison with linear speed analyses’ and to allow ‘a more direct*
*comparison with previous literature’. I’m not convinced by these arguments. Recent research has shown*
*that a considerable fraction (> 50%) of speed cells have nonlinear tuning curves (Hinman et al. 2016),*
*something the authors also find and report. Similarly, AHV, HD and speed cells have all been shown*
*to exhibit nonlinear tuning curves (Hardcastle et al. 2017; Mallory et al. 2021). Hardcastle et al. (2017)*
*compared their linear-nonlinear analysis to the approach used by the authors and found that it missed*
*~60% of informative cells – especially in superficial layers of the mEC, which the authors indeed found*
*lacking in self-motion coding. The restrictive nature of the analysis continues because the authors do*
*not include all types of linear responses such as negatively sloped bidirectional AHV cells (~9% of AHV*
*cells; Mallory et al. 2021) and negatively sloped speed cells (~16% of speed cells; Hinman et al. 2016).*
*As far as I can tell linearly asymmetric and symmetric cells are not differentiated in the analysis. This*
*approach leaves large gaps in the authors data that need to be addressed with different analyses or a*
*discussion of the limits of the existing analysis with adjusted claims.*
*We thank the reviewer for these comments. They made us realise that:*
*1) We were not clear in the text on the fact that the linear analysis that we are using (Pearson*
*correlation) allows quite some variation from a strictly linear response, including sigmoid curves*
*rightfully described in Hinman et al in MEC and by ourselves in MEC, PrS and PaS. We realise that*
*this omission was quite misleading. We have now cleared up this important fact within the text.*
2) It would be indeed very interesting to comment more on negatively sloped and other miscellaneous
coding. We had already added some description on what we called the GLM only population which is
encompassing such non-linear modulation. We have now (i) added a much more thorough description
of this varied population, including example of rate-profiles in supplementary figure S3. Furthermore,
given the importance of this population and in order to allow comparison with studies using GLM, we
have now (ii) re-done all our analyses with the GLM dataset and have systematically reported variation
with results obtained with the linear method. As stated in response to point #1 our main claims remained
unchanged. Finally, we have (iii) increased the extended discussion adjacent to supplementary figure
3 on the comparison between these methods, specifically regarding how the different populations are
captured by each method (see supplementary discussion after legend of supplementary fig 3).
3. The authors claim that they provide “the first recording in the hippocampal region of the conjunctive
coding postulated to be found in the “hidden layer” of CAN models”. This doesn't seem to be true
because conjunctive grid x HD x speed <https://doi.org/10.1126/science.1125572> and conjunctive
position x HD x speed <https://doi.org/10.1038/s41467-021-20936-8> cells have been reported in the
mEC.
We have now adjusted the quoted sentence, to stress what is novel in our findings and how they confirm
what was found a relatively low number of cells in Mallory et al. To be clear, our claim was specifically
referring to ring attractor models and their hidden HD x AHV layer in the parahippocampal region,
that was not observed before Mallory 2021. As stated above (line #23), our manuscript was submitted
before the publication (and without knowledge of impending publication) of this study. We have now
highlighted how our study contrasts and complements previous work, notably by providing substantial
evidence of the widespread nature of self-motion conjunctive codes, as well as some data and discussion
on what they mean for theoretical models. We already commented on the difference between our study
and Sargolini 2006 line #34 (see above).
It is also a little misleading because conjunctive HD x speed and HD x AHV cells were found a long
time ago in DTN where the HD signal is thought to originate (i.e. where we think the HD CAN is
actually likely to reside) <https://doi.org/10.1523/JNEUROSCI.21-15-05740.2001>.
We agree that the DTN is likely to be a very important subcortical structure in directional signal
processing. However, our opinion is that that the hippocampal region has a wider role in information
integration, notably through what we prove to be an intermingling between (primary and derivative)
directional and spatial signal – especially given that no spatial modulation was reported in the DTN.
As such, our study brings significant complementary and contrasting result to the important work from
Joshua Bassett and Jeffrey Taube. While we had already cited their work, we have now adjusted the
discussion to reflect this point (see manuscript p11).
There are other omissions such as not pointing out that Hinman et al.
2016 <http://dx.doi.org/10.1016/j.neuron.2016.06.027> also found sigmoidal speed coding in speed cells
or that many cells in fact have nonlinear tuning curves (Hardcastle et al. 2017; Mallory et al. 2021).
These findings are foundational to the study so I am surprised they are omitted or not discussed more
fully.
We have already discussed the absence of citation to Mallory. Hardcastle and Hinman were already
cited in the text, however we are thankful for the reviewer to highlight the need for a clearer discussion
(and adequate citation) on sigmoid coding (also see our response line #145). We have now adjusted the
text at multiple locations to include more references to these studies.
4. I had problems with the discussion that are quite general, but I want to list them here because they
are included in the abstract and are important to the overall paper. The authors argue that 1) they have
revealed the ‘hidden layer’ necessary for CAN models but 2) their conjunctive cells lack a “structured
organisation” and 3) that they have “unexpected” features which call into question CAN models. It’s
not really clear to me what organisation or features the authors mean; I think it’s that the cell types and
conjunctive coding combinations are mixed? Why is that a problem? It is also unclear what organisation
they would expect instead. There is no real explanation of this given, or examples of CAN models
specifically. Ironically, I think nonlinear self-motion signals do pose a significant problem for CANs
but these are mainly excluded from the analysis.
*We thank the reviewer for these comments. We understood from them that our argumentation was not*
*clear enough on that point. We have now both mitigated and clarify our claims related to the conflicts*
*between our findings and current CAN models.*
*Our argument can be summarized as follow:*
*(i) Ring attractor models (the simplest example of CAN), require three population of cells: a principal*
*attractor of HD cell and two “steering” populations (the hidden layers), that are conjunctive for HD*
*and angular head velocity in the clockwise and countercklockwise direction respectively. This*
*populations have to be connected to the principal population with a constant offset between their input*
*and their output, to generate the activity flow required to move the activity bump.*
*(ii) Such models have no use for pure AHV cells, as well as for bidirectional AHV cells or anticorrelated*
*AHV.*
*(iii) If steering cells are not silent in the absence of movement, then the two steering population have to*
*be active at the same level for the bump to remain still: this fine tuning requires the two steering cell*
*populations to interact. CAN models have not been, so far demonstrated to work if these interactions*
*are included.*
*(iv) The random mix of selectivity that we find suggests that population are shared between different*
*CANs: grid cells do not only belong to the position-coding CAN, but can be part of a ring attractor*
*CAN as well, if they code for HD or AHV or both. Are then different CAN able to function if they share*
*the population on which they are built? Under which conditions?*
*The text has now been modified to reflect these arguments.*
There is a suggestion that cells coding ‘derivative’ signals could be used in non-spatial
representations, this is brought up in the last couple of sentences of the paper and mentioned in the
abstract despite there being no evidence for this or previous discussion. What do the authors mean?
This needs further explanation.
*We agree with the reviewer that this speculative point deserves more space and explanation in the main*
*text. We tried to provide more context in the new version of the discussion, and provide some more*
*context here.*
*Our speculation draws from the many similarities we find between speed and AHV cells (the*
*quantitative preponderance of quasi-linear coding, the unstructured conjunctions with other cells, and*
*the similarity in the firing properties), and the observation that both speed and angular head velocity*
*signals exist only as variation of “primary” quantities (position and head direction), whose coding is*
*instead very different, with a preponderance of highly nonlinear “receptive field” coding (place/grid*
*fields, preferred head directions).*
*Speed and angular head velocity are scalar in nature, and take values in a finite range (maximal speed*
*and angular velocity values are set by physical constraints on how fast the animal can move). This*
*makes them representable with some form of monotonic dependence of the firing rate from their value,*
*something that is impossible for multidimensional and unbounded (position) or periodic (head*
*direction) quantities. Since position and orientation as far from the only “primary” signals that the*
*hippocampal region encode we wonder: could we find a similar phenomenon for the coding of the*
*variation of other “primary” signals (e.g. the position in a cognitive space ([10.1126/science.aaf0941](https://doi.org/10.1126/science.aaf0941))*
*or the representation of a tune frequency ([10.1038/nature21692](https://doi.org/10.1038/nature21692)))?*
Minor points (R1)
Introduction
**1.** Please include line numbers in future submissions (I received a version with line numbers after
completing my review). Due to multiple readings some of my comments may be out of order – I have
tried to add some context using text from the manuscript.
*We have remedied to that and apologize for the inconvenience.*
**2.** Overall, I thought the introduction was very clear and well written and only had some minor remarks.
I will refer to Mallory et al. 2021 (<https://doi.org/10.1038/s41467-021-20936-8>) in some places. I
appreciate that the authors may have submitted their manuscript before that paper was available.
*We are happy to read the reviewer’s appreciation of our work. As mentioned above, we indeed*
*submitted without any knowledge that Mallory et al had decided to add angular velocity analyses to*
*their study. The reference to this work is now added in several places in the text.*
**3.** “...an (apparently) self-standing...” What is the purpose of ‘apparently’ here? If the authors feel that
speed cell modulation is not self-standing that might warrant further explanation.
*We modulated the self-standing statement with “apparently” because the authors cannot guarantee in*
*our opinion that there is not a code for other features that they had not tested. However, we understand*
*that this modulation while true, may be misleading. We thus followed the suggestion from the reviewer*
*and decided to simplify the text by suppressing it.*
**4.** “...in principal neurons of the hippocampal region.” Hippocampal region is ambiguous – it could
mean hippocampus (hippocampus proper) or the hippocampal formation, which some consider to
include the Sub and EC. Thus, angular velocity coding has been observed in the hippocampal formation
but not the hippocampus.
*In using the term “hippocampal region”, we followed the nomenclature established among others by*
*David Amaral and Menno Witter (Paxino’s *The rat nervous system*, 3rd edition chapter 21 on*
*hippocampal formation, Elsevier). A widely accepted reference in our opinion when it comes to the*
*anatomy of the hippocampal region. The last author, Charlotte Boccara, was herself involved in an*
*anatomic work with Menno Witter that was also cited in the text.*
*According to Witter and Amaral the hippocampal region is constituted of the hippocampal formation*
*(archeocortex: DG, CA1–3 and subiculum) and the parahippocampal region (EC, PrS, PaS, POR and*
*PER). In order to clarify our statement, we have added a reference to the Paxinos chapter when we*
*introduce the term hippocampal region in reference to distinct brain regions.*
**5.** “Most reports come from recordings of subcortical structures (e.g., lateral mammillary nuclei, dorsal
tegmental nucleus), linked to the processing of vestibular information” Also postsubiculum and
retrosplenial cortex – these are cited later but I don’t know why not here:
<https://dx.doi.org/10.1177%2F2398212817721859> and <https://doi.org/10.1101/2021.01.22.427789>
*We did not cite the postsubiculum and the retrosplenial results here (but only later) because (i) they are*
*not subcortical regions and the point we wanted to stress out here was that most reports (both in term*
*of number of studies and in total number/percentages of cells) were coming from the DTN and the*
*mamillary nuclei. Furthermore, (ii) the aforementioned study focused on how head direction cells are*
*modulated by angular head velocity. While our purpose here was to investigate coding for self-motion*
*as a feature of its own. Only as a second step we turned to investigate its interaction with position and*
*direction. This resulted in our novel and robust findings that there is no special abundance of AHV x*
*HD cells, and AHV cells can be found on their own in the parahippocampal region. (iii) We were in*
*contact with Keshavarzi Sepiedeh but their study is still at the stage of pre-print as for now.*
**6.** “Most crucially, it has remained unclear whether angular velocity coding reflects an intelligent
design.” From the encyclopaedia Britannica: Intelligent design (ID): argument intended to demonstrate
that living organisms were created in more or less their present forms by an “intelligent designer.” Is
this really what the authors mean?
*This has now been corrected.*
**7.** “Exciting evidence compatible with such CAN models was recently provided by investigations of
the *Drosophila melanogaster* central complex...”
Also see <https://doi.org/10.1101/2021.02.25.432776> which may have been published after this paper
entered review.
*Thank you for pointing out this new study, however we would prefer to only cite peer reviewed articles.*
**8.** “To understand the circuit mechanism by which spatial representations can be updated in mammal,
we...” In mammals?
*This typo has now been corrected. Thank you for spotting it.*
**9.** “...it appears that direction, position and speed selectivity are randomly admixed with each other...”
Why is this a problem for CANs? There needs to be some better explanation of this because it seems to
be key for understanding the discussion.
*We agree with the reviewer that this point was not clear enough, and we have rewritten the discussion*
*to clarify this aspect. We refer to point #4 (line 200 of the present document) for a summary of our*
*arguments on how our results, and in particular the conjunction structure, are at odds with the implicit*
*assumptions of CAN models.*
Results
**10.** Generally, I liked the results, the figures are clear and very well presented. Most of my issues stem
from the choice of analysis rather than the proceeding tests or figures.
*We are happy to read the reviewer appreciates our work. We will answer to the choice of analysis when*
*specifically mentioned by the reviewer.*
**11.** Fig 1d: Some of the bars have multiple colours (i.e. mEC CCW has two colours) what do these
correspond to?
*Shaded areas represent overlapping populations. We had mentioned this in the figure legends with the*
*following sentence: “The shaded areas represent the intersection between AHV-CCW & AHV-BiDir*
*(darker shade) or between AHV-CW & AHV-BiDir (lighter shade).” However, based on the reviewer*
*comment, we highlighted that point in the main text to ease the comprehension (p4 l11).*
**12.** Fig 1e: Some of the cells have very low AHV scores (<0.10) and generally it seems all scores were
low (<0.20). Could the authors comment on this? Is it because the cells are conjunctive and AHV only
explains a low fraction of the cell’s firing? How do these scores compare to cells in vestibular
regions? AHV scores (i.e. <0.10) and generally it seems all scores were low (<0.20). Could the authors
comment on this? Is it because the cells are conjunctive and AHV only explains a low fraction of the
cell’s firing? How do these scores compare to cells in vestibular regions?
*We agree with the reviewer that the scores are low – even though significantly higher than the statistical*
*null model. We believe – and in part show – that cells in the parahippocampal region hardly serve a*
*single purpose. We thus expect – as suggested by the reviewer – that the firing of each cell is modulated*
*by a relatively large variety of factors, only a few of which we can test for.*
*In other words, we expect only a fraction of the variance of the firing rate to be explained by each*
*correlate, leading to small values of the scores. The strict criteria we impose on our main detection*
*method (a null distribution with tens of thousands of samples, and a 99th percentile threshold) make us*
*confident that the effect we detect are solid and reproducible, despite the low scores.*
*Cells in subcortical regions usually show a larger correlation (around 0.5 on average, from Basset &*
*Taube 2001, for example). This is in line with the idea that these regions are specialized in the*
*construction and representation of movement signals as their main function, while high-level cortical*
*structures integrate this information with several other factors. We have now stressed out this important*
*point in the discussion (page 11 line 26 of the manuscript), thank you for pointing it out.*
**13.** Fig 1f: for the plot showing the mEC unidirectional data, the dashed line showing the lower
percentile (CCW cutoff) is quite different to the CW one and there seem to be hardly any shuffles lower
than it. Is this line plotted in the correct place?
*We thank the reviewer for noting the problem. The threshold was indeed wrongly plotted in the figure.*
*We have now fixed this error.*
**14.** Fig 1f: are the pink data curves and grey shuffle histograms calculated in the same way? i.e. does
the pink curve just trace a histogram made with the same settings as the shuffle distribution? If not it is
a bit hard to compare the two distributions. Also, what does ‘frequency density’ mean? There must be
100x as many shuffle data points so how are these normalised? Usually the y-axis would show fraction
of total.
*The data curves and the shuffled histograms are calculated by binning the correlate in the same way,*
*just plotted differently (one as a curve, one as a histogram) to minimize superposition and facilitate*
*visualization by the reader. We chose this type of plotting for consistency with Boccara 2010 and other*
*seminal work in the field.*
*We agree with the reviewer that the term frequency density is not the clearest, and have now changed*
*it to probability density. Density here means that the y value is the raw frequency divided by the total*
*number of data points and the size of the interval on which the histogram is calculated. This ensures*
*that the area under the curve is normalized to one for both histograms, and makes it possible to compare*
*them despite the difference in sample size. Note that the term density, even though a bit confusing, is*
*needed, because to obtain probabilities one has to multiply the y-values for the bin width.*
**15.** Fig 1F: There are regions of negative bidirectional angular velocity score that look to significantly
exceed the shuffles, at least in PaS and PrS but maybe not in mEC (which would be an interesting
finding). Why are only positive BiDir correlations included? You could have a lower percentile cutoff
as you do with UniDir coding.
*As mentioned above (line 148), we had originally decided to follow the example of primary correlates*
*definition, as well as the definition of self-motion modulation outlined in Kropff et al, both of which*
*only included positive correlations. However, we agree with the reviewer that negative correlate coding*
*is indeed very interesting, and we have now adjusted the text so that we report negative correlates for*
*derivative signal in more details when discussing the GLM results (page 4, ligne18 and 32 in the*
*manuscript).*
**16.** Fig 1H: I would rather see just the significantly modulated AHV cell scores. Perhaps the non-
significant cells could be in another colour with the significant ones plotted on top? Also calling the
plot a ‘correlation’ might be a bit confusing here – the correlation would be calculated on the points,
and could be represented by a line, but the points themselves don’t show ‘a correlation’. Same points
also for Fig 2F.
*We agree with the reviewer that the plot itself is better defined by the “scatterplot” label. We changed*
*the legend to “scatterplot showing the correlation between ...”. Following the reviewer suggestion, we*
*have also modified the plot so that self-motion cells are shown in color.*
**17.** Fig SF2 and other figures: you do not need to repeat numbers (i.e. proportions in legend) that are
plotted in a graph. These numbers inflate the figure legends and obscure more useful information. I
would remove them throughout but this is up to the authors.
*Following the reviewer recommendation, we took away numbers in SF2.*
**18.** Fig SF2: how do these t-tests work on single values for each group? these t-tests work on single
values for each group?
*We thank the reviewer for spotting an error in our reporting: the test used were not t-tests, but*
*proportion z-tests. Such tests can be used to compare proportions and is, in the two-samples, two-sided*
*version we used, equivalent to a proportion chi-squared test. We corrected the name of the test in in the*
*manuscript. For further information on that test, see:*
https://www.statsmodels.org/stable/generated/statsmodels.stats.proportion.proportions_ztest.html
**19.** Fig SF4c and more generally: a “Skaggs score” is mentioned in the results and described in the
methods but I’m not really sure how this was used. Was it used? As far as I understand it the ‘GLM
class.’ bars in SF4c refer to GLM results and the ‘rate score’ bars refer to the correlation results? If so
where are the Skaggs score results? Either way I don’t think these labels are very clear. If the Skaggs
score is not shown or used it should just be removed from the paper.
*We agree that while the so-called Skaggs score is interesting in itself, it does not add much information*
*here beyond what the GLM analyses bring; and that – in this context – it is misleading. We have*
*therefore chosen to follow the reviewer’s suggestion and remove it from the paper.*
**20.** Fig SF4c: The ‘rate score’ bars are separated into CW, CCW and BiDir, what are the other sections
in between?
*See answer to minor point #11. See legend of SF4: “Shaded areas represent overlap between CW*
*(CCW) and BiDir cells.”*
**21.** “As per our definition, AHV cells are neurons whose firing rate is positively modulated by angular
velocity” This seems like an very restrictive definition given that nonlinear-asymmetric and negatively
sloped AHV cells exist (although they are admittedly less common). [https://doi.org/10.1038/s41467-](https://doi.org/10.1038/s41467-021-20936-8)
[021-20936-8](https://doi.org/10.1038/s41467-021-20936-8). Maybe the authors want to extend this analysis to include non-linear representations.
*See answer to main point #1, #2 and minor point #15.*
**22.** “velocity values were treated as categorical variables; therefore, no linear dependence was implied”
Is this correct? I’ve never checked but it seems that AHV and running speed are likely to be correlated
– i.e. if a rat is running in an arc it will also have a high AHV? no linear dependence was implied”
*With this sentence we wanted to clarify the difference between the GLM approach and the Pearson*
*scoring method, underlying how a the «Linear» in Generalized Linear Model does not refer to the*
*relation between the independent and dependent variables.*
*On the observation of arc running, we would like to highlight the fact that AHV was calculated with the*
*use of two LEDs on the animal head, and therefore refers to the relative motion of the head with respect*
*to the body, which is almost constant in a curved trajectory.*
**23.** “Out of 182 AHV cells solely picked up by the GLM method, about half of them (54%) were
anticorrelated with the absolute value of angular velocity” Are the tuning curves of these cells
convincing? This seems like a significant population of informative cells that are being excluded. Also,
why does this sentence highlight the anticorrelated cells specifically? Is the point that many
anticorrelated cells were missed by the correlation approach? Or that more of the cells solely identified
by the GLM were anticorrelated than we would expect?
*See answer to main points #1, #2 and minor points #15, #21. What we highlight here is that indeed AHV*
*cells identified only by the GLM method were for more than half of them anti-correlated. We initially*
*decided to exclude such cells from the main analyses for the reasons explained above. Not because we*
*do not think that they are interesting but to be consistent with Kropf 2025 as well as with how other*
*correlates are analysed. However, we have now added a more detailed description of this population*
*as well as elements of discussion and re-did all analyses with the GLM method to highlight a possible*
*effect of this population on our main results. As reported previously, we did not find such an effect.*
**24.** “In addition, we observed that rectilinear speed signals...” As far as I know speed cells haven’t been
reported in pre- or parasubiculum before, so this result is quite interesting and might warrant some more
discussion (i.e. do these structures differ from mEC in terms of velocity inputs?).
*We agree with the reviewer. The presence of speed cells in PrS and PaS is one of our main findings. As*
*to the questions of their inputs, we added a sentence on this in the discussion (page 13 of the*
*manuscript). In brief, velocity modulation has been hypothesized to be linked with many regions*
*including the PER/POR, the retrosplenial cortex, the cerebellum, the thalamus and the septum (Muir*
*and Bilkey 2003, Cooper and Mizumori 2001, Rochefort et al. 2011). All these regions project in a*
*similar fashion, either directly (or indirectly in the case of the cerebellum) to the MEC, the PrS and the*
*PaS (Amaral and Witter, Van Groen and Wyss 1990, Burwell and Amaral 1998, Van Strien 2009, Bohne*
*2019, Watson 2019). Further connectivity analyses would be needed to assess specifically the difference*
*in strength of connection between these areas. We agree with the reviewer that this would be very*
*interesting to get an answer on that question.*
Is there a reason the proportions might be higher in pre/parasubiculum? Are the proportions
significantly higher?
*The proportions are not significantly higher, as now reported in figure 2 legend, in which significant*
*differences are highlighted. The fact that speed coding is widely present in the hippocampal regions is*
*indeed one of the main result of this work, one we think can be of interest for the understanding of how*
*this code work at the mechanistic level.*
**25.** “Speed cells were uniformly distributed across all layers in each area...” How was a KS test run on
these data – they are just single proportions?
*We used the one sample KS test from the modelstats API – see the following reference for more*
*information: <https://docs.scipy.org/doc/scipy/reference/generated/scipy.stats.kstest.html> – in order to*
*compare the observed distribution of proportion with an uniform distribution. In this sense, each*
*percentage is treated as a data point.*
**26.** “significantly overlapped with those obtained with our linear scoring method (binomial test, pvalue
< 0.001” Is the p-value for the GLM or Skaggs score? Again, where is the Skaggs score used/reported?
*The p-value was for both the GLM and Skaggs. Now we have adjusted the text to report the p-value for*
*GLM only and have removed Skaggs. See point # 19.*
**27.** “In order to allow for comparison with the latest reports of speed cells in the MEC, we decided to
use the more conservative Pearson scoring methods for all further analyses.” This isn’t really true as it
has been recently (within last 5 years) shown that about 33% of speed cells are best fit by a saturating
exponential and are poorly fit by a linear model (<https://doi.org/10.1016/j.neuron.2016.06.027>) or that
they are just nonlinear (<https://doi.org/10.1016/j.neuron.2017.03.025>) which is in agreement with your
GLM results (and later sigmoid results). That paper also points out that non-linear speed coding presents
some problems for attractor network models, which seems directly relevant to the current manuscript.
*See answer to main points #1, #2 and minor points #15, #21, #23.*
**28.** Fig. 2d: as with AHV there are regions of negative speed score that look to significantly exceed the
shuffles in all 3 regions. We know negative speed cells exist so why are they excluded?
*See answer to main point #2 and minor point #15, #21, #23.*
**29.** “These percentages were similar to the percentages observed in the general population” This is quite
a nice observation,
*We agree with the reviewer, this is indeed one of the points that we have attempted to stress out.*
I think it would be good to have a test here (Chi square test maybe) comparing the proportions.
*Whenever comparing percentages, we used a two sample, two-sided proportion z-test, that is indeed*
*equivalent to a proportion chi-square test.*
**30.** “We observed all possible types of conjunction...” Because this paper is closely linked to CAN
models it might warrant spending some more time on this. On my first readthrough I wasn’t sure if you
had looked at conjunctions of more than 2 parameters. Compare this to Figure 1c&d in Mallory et al.
(2021; <https://doi.org/10.1038/s41467-021-20936-8>) where we can see the breakdown of major groups
up to 4 conjunctive parameters. Looking at your Fig SF5 (much harder to find) we can see, for example,
that there are very few grid x HD x speed cells, but isn’t this conjunction important to CANs? Why are
there so few? Why are there comparatively more grid x AHV cells, which don’t really fill an important
computational function? Maybe these are what you mean later in the discussion when you talk about
the ‘features’ of conjunctive cells not being what you expect; if so that needs to be much clearer.
*We reformulated the discussion to clarify this point. What the reviewer report is indeed correct, and is*
*the kind of percentages that we would expect from our null hypothesis of random independent*
*assignment: the probability of grid x HD x speed conjunction, in this null model, is the product of three*
*probabilities (p_{grid} , p_{hd} and p_{speed}), and is therefore quite low.*
*The grid x AHV conjunction is instead the product of just two probabilities, and it’s higher. This does*
*not mean that there is a preference for AHV and grid to be conjunctively encoding, but in fact that each*
*feature is independently assigned. The fact that grid x AHV has no use in CAN models and are yet found*
*in quite large proportions is precisely one of the observations that make our data at odds with the*
*standard formulation of CANs.*
**31.** “This result contrasted significantly with previously published studies of a predominantly self-
standing code in speed cells recorded in the MEC superficial layers” Hinman et al.
(2016; <http://dx.doi.org/10.1016/j.neuron.2016.06.027>) report grid x HD x speed modulation in the
mEC: “...the speed modulation of firing rates of MEC neurons was positive ... for each individual cell
type ... (... conjunctive, $p < 0.0005$...)”
As do Sargolini et al. (2006; <https://doi.org/10.1126/science.1125572>). These should be cited here.
*See answer to main point #1 and #2. We would like to stress to the reviewer our use of the word*
*predominant here. Sargolini only looked at speed modulation among grid cells (and did not report*
*significance). Their results therefore cannot address the question of the dominance of self-standing*
*code. The picture that emerges from our study is indeed in line with the results of Hinman et al., which*
*we extended to angular head velocity, and regions other than MEC, and tied together in the single*
*theoretical framework of the independent assignment hypothesis. We have modified the text to better*
*reflect both these studies.*
**32.** “The scores (grid, HD, AHV and linear speed score) were mostly independent from each other” I
don’t understand this analysis. The previous paragraph argues that conjunctive coding is represented
across all primary cell types. As I suggested above, I think a better way to show this would be to compare
the proportions observed across all cells and those observed in AHV cells (for example) or use a test
looking at the intersections like the one used for the overlap of GLM and correlation results. The reason
I’m not really convinced by the correlation approach is that there are far fewer derivative (speed, AHV)
coding cells than primary ones (grid, HD), so we wouldn’t necessarily expect a correlation between
these parameters in the overall population. The authors want to make some strong claims about CAN
models based on this information, so I think it is worth spending some time to make this analysis more
convincing. If I am misunderstanding then maybe a figure is necessary?
*We have in fact conducted the analysis of independence in two ways: by looking at correlation between*
*scores (also see manuscript page 8 line 14) and by looking at the intersection between the populations*
*of cell coding for different properties, using a binomial test in the same way done for the intersection*
*between GLM and Pearson methods (see methods and page 8 in manuscript). Both analyses yielded the*
*same results: there is no tendency for cells to segregate or cluster together on the base of what they*
*code for. This can be seen both by the lack of correlations between scores and by the fact that*
*percentages of conjunctive cells are not significantly higher or lower than what would be expected by*
*the null hypothesis of random independent assignment. Figure 3 and 4 are meant to illustrate this point*
*by providing example and a global picture of how score and percentages behave. Based on the reviewer*
*comment we have now clarified the implications of these results by restructuring the discussion.*
**33.** “That self-motion information is integrated at the unit level in all cell types and all tested layers
(with the notable exception of MEC LII) is incompatible with current CAN model and calls for their
revision.” In what way does this finding preclude existing models? This statement needs a citation or
some more explanation.
*We thank the reviewer for pointing out the lack of clarity of this sentence, that we changed in the main*
*text. What we meant was that many of the observed features of self-motion coding at the unit level (and*
*not, indeed, self-motion information at the unit level per se), are at odds with CAN models. We have*
*now clarified this point in the new discussion. See answer to main point #4 for how this is at odds with*
*existing models.*
**34.** “It is important to notice that, contrary to what would be expected from CAN models, unidirectional
AHV cells were not silent when the rats were not turning their head...” We can’t know this from the
information given in Fig SF6 or the results. The authors would need to find the y-intercept of the linear
or sigmoidal fits on the speed cell tuning curves and show that these are $>0\text{Hz}$. Although they have
excluded negative speed cells and BiDir AHV cells which would be the ones best supporting their
argument. However, if the baseline rates are $>0\text{Hz}$, why does this matter for CAN models? It seems
trivial for the network to subtract baseline rates from each cell (or just use an averaged population speed
code). Can the authors explain this a bit better with citations?
*We agree with the reviewer that the information about firing at zero AHV cannot be extracted from*
*SF6. We drew the conclusion from the fact that, in AHV cells (see example tuning curves in main and*
*supplementary figures) the average firing rate in periods of zero AHV – i.e., the point of the plotted*
*tuning curves at $x=0$ – is not zero. This can be seen directly from the tuning curve, and its true*
*regardless of the shape of the tuning curve itself. We added the reference to example tuning curve in*
*the text.*
*In ring attractor CAN models the activity bump is stable (is not pushed left or right), only when the two*
*supporting self-motion integrating rings (composed of HD x AHV cells) are both silent (at zero AHV).*
*If those populations are not both silent, they must be active with exactly the same average. How this is*
*achieved is not described in CAN models, and is not a trivial aspect, since it requires the addition of*
*interactions between the two hidden layers of the model.*
**35.** “To test this hypothesis, we calculated the percentage of the correlate values at which each cell fired
more than its average firing rate (see Methods and extended data Fig 6d).” This was difficult to find in
the methods as it is under the heading “Sparsity calculation”. The heading or the main text could be
clearer. More generally, I didn’t have any specific issues with this paragraph, but I’m not entirely sure
what the conclusion is. We don’t know if the cells have the same waveforms or if they are
inhibitory/excitatory, which seems more important for trying to understand their firing dynamics.
*We have now modified the text so that sparsity calculation appears before the reference to the methods.*
*In our opinion, the analysis of firing dynamics, and its modulation of functional properties, could*
*provide useful information for the scientific community trying to understand such a phenomenon at the*
*circuit level.*
**36.** “...could be explained by the fact that the monotonic firing profiles used to encode motion signals
is less sparse...” Didn’t the authors largely exclude non-monotonic derivative signals when they chose
to rely on the linear correlational approach for classifying cells? I don’t think these kinds of conclusion
can be made given the current analysis.
*Based on the reviewer legitimate concern, we have now re-done these analyses for with the GLM cells.*
*We observed that sparsity analyses yielded similar results with GLM modulated cells. This, we*
*hypothesize might be due to the fact that truly “receptive field” like cells are rare, and the difference*
*between primary and derivative coding is not affected by their presence. These new results have now*
*been added and strengthen our conclusions.*
**37.** Figure SF3: this figure seems to be out of place, shouldn’t it be later? I really like this figure by the
way, and Fig SF1, mainly because they show lots of informative examples. I would like to see the
examples at the bottom of Fig 3 extended to show more cells as well. Especially because this is the only
place HD and grid cells are shown.
*We thank the reviewer for their positive comments on our supplementary figures. We have now*
*exchange SF3 and SF4 and have added examples of non-linear as well as negative self-motion coding.*
**38.** “...the sigmoidal fit with low steepness was slightly more predominant among speed cells (56%).
This result seemed to be due to a saturation observed at high speeds...”
Hinman et al. (2016; <http://dx.doi.org/10.1016/j.neuron.2016.06.027>) also report: “...the majority [of
speed cells] were actually better fit using a saturating exponential ... function rather than a linear
function ... [145 of 260] ...” Which is about 56% or almost identical to the results reported here.
Hinman is cited later for something else but should be credited here especially because this is a
remarkably close replication.
*We thank the reviewer for pointing this overlook from our part and have now cited Hinman in this*
*paragraph in addition to our later citation.*
**39.** “We thus concluded that most speed cells also followed a quasi-linear rate function.”
This feels like a circular argument because the analysis for classifying cells was linear. Again, I don't
think the authors can make these conclusions.
*We understand where the concern of the reviewer comes from, but we do not agree that the argument*
*is circular. We would like to stress that the score that yielded modulation label is a correlation analysis*
*between the whole time series of the firing rate and the behavioural correlate (speed/AHV), plus the*
*comparison with the suitable statistical null model provided by the shuffling. This is different from a*
*linear fit with goodness of fit as a modulation index. Indeed, in many cases cells were better fitted by a*
*sigmoidal functional shape, highlighting the fact that the methods is more sensible to proportionality*
*between rate and correlate than perfect linear relationship. In this sense the analysis of the tuning curve*
*shape provides additional qualifying information to the simple score. Regardless, we have now adjusted*
*the text in several point to clarify the goodness of fit analyses. Also see answer to main point #2.*
**40.** Fig 5: I like the detail and clarity of this figure, but is it normal that the x-axis of the left-hand plots
extend underneath everything else? I would also recommend a pattern other than diagonal hatching as
it creates an optical illusion. i.e. for me the HD bar in Fig 5e looks like it ‘bends’ where the hatching
ends. Also see Fig SF7. This isn’t scientifically important, but it is a little distracting.
*We thank the reviewer for their positive comments on our figures. We had opted to extend the x-axis to*
*help with the clarity of the figure (in order to separate the graphs belonging to UniDir AHV, BiDir*
*AHV and speed cells). We could not find another pattern than the diagonal hatching that would still*
*result in a clear figure, coherent with the colour and pattern codes used throughout the whole*
*manuscript.*
**41.** Fig 5f: my understanding is that these plots show the proportion of each cell type that is both speed
(or AHV) modulated AND theta modulated? i.e. the proportion of grid cells that are speed and theta
modulated is around 0.39 (yellow hatched bar on red background). But the legend says: “conjunctive
grid x speed (yellow left, 51.9%)” which does not correspond to the bar. None of the legend values
seem to match the bars, so, what do the graphs mean?
*We thank the reviewer for spotting this mix up in our reports of numbers! We have now fixed this issue.*
*Just to be clear, 5f shows the proportions among each of 6 principal categories of self-motion coding*
*cells that we have defined as (i) AHV only, (ii) AHV-by-grid, (iii) AHV-by-HD, (iv) speed only, (ii)*
*speed-by-grid, (iii) speed-by-HD. Meaning that our results of 38.2 % of “pure speed” being theta-*
*modulated signify that 61.8% of “pure speed” are not passing our criteria for qualify as theta-*
*modulated Note that there might be some overlap between AHV-by-HD and AHV-by-grid. Same is true*
*with speed-by-HD and speed-by-grid. The goal of this analyse is to determine whether some*
*subcategories (such as grid-by-speed) could present a higher percentage of theta modulation. We show*
*here that this is not the case and that there seems to be no influence of assignment to a specific category*
*on propensity to be theta modulated.*
**42.** In some graphs (i.e. Fig 3a&c, Fig 4a, Fig 5 and Fig SF7) the y-axes are labelled as ‘frequency’
while it is supposed to say ‘proportion’. These words have different meanings. In some cases the main
text says proportion while the figure says frequency: i.e. “the proportions of velocity cells that were
theta modulated (Fig. 5f ...” but the y-axis of 5f and its legend say frequency while it is supposed to
say ‘proportion’.
*We thank the reviewer for highlighting the inconsistency, we changed to the uniform use of "proportion"*
*throughout the manuscript.*
**43.** “...the proportions of velocity cells that were theta modulated (Fig. 5f, t-test: pvalue <
0.05)”“...except for MEC LII which showed more theta modulation and MEC LVI which showed less
(extended data Fig. 7a, t-test: pvalue <0.01)”How are these t-tests conducted? There is a single value
representing a proportion and a chance value, I don’t see how you can run a t-test? What chance level
was used for fig. 5f? the proportions add up to more than 1 (because cells can be in more than one
group). What chance level was used for Fig. 7? The legend says it was the value “expected given the
average theta modulation in that specific layer” but I’m not sure how that would be calculated. These
test results need to include degrees of freedom and test-statistics and ideally a measure of effect size
(same for all other stats like the KS test reported elsewhere).
*We thank the reviewer for pointing to an error in the test name reporting in the manuscript. We used z-*
*test (and not t-test) to compare percentages of modulated cell in each layer to the percentage found in*
*the general population, using the proportion z-test from statsmodel python API:*
https://www.statsmodels.org/stable/generated/statsmodels.stats.proportion.proportions_ztest.html
*The test we use (the two sample, two side alternative) is equivalent to the proportion chi-square test, as*
*reported in the API references. We now report the z-statistic value and the number of degree of freedom*
*together in the pvalue in the text.*
*In figure 5f, the comparison was between the percentage of theta modulated cells in each group and*
*the percentage of theta modulated cell in the whole population. In figure 7, the legend was wrong, we*
*thank the reviewer for pointing to the error. We updated to report the correct information: the*
*comparison was between the observed percentage of theta modulated cell in each layer and the*
*percentage in the the whole population.*
**44.** “Velocity coding is independent of theta modulation” Why is there no mention/citation of Hinman
et al . (2016; <http://dx.doi.org/10.1016/j.neuron.2016.06.027>)? It seems really odd that this is omitted
considering the whole paper is about the topic of this paragraph.
*We thank the reviewer for pointing out this omission that is now corrected.*
Discussion
**46.** “...the first recording in the hippocampal region of the conjunctive coding postulated to be found
in the “hidden layer” of CAN models”.
Conjunctive grid x HD x speed <https://doi.org/10.1126/science.1125572> and conjunctive position x HD
x speed <https://doi.org/10.1038/s41467-021-20936-8> cells have been reported in the mEC previously.
Although for the latter conjunctive HD x speed and HD x AHV cells in DTN are more likely linked to
the generation of the HD signal <https://doi.org/10.1523/JNEUROSCI.21-15-05740.2001>. Why are
these citations omitted?
*See answer to main point #3, line 23, 34 and line 163. The text in the discussion has been adjusted.*
**47.** Are there any models that require all speed and AHV cells to conjunctively code for position and
head direction? Could you add a citation? This seems like a straw man argument.47. Are there any
models that require all speed and AHV cells to conjunctively code for position and head direction?
Could you add a citation? This seems like a straw man argument.
*We agree with the reviewer that, as it was pointed out already in main point #4 and minor point #9, the*
*discussion on the implication for CAN was not clear. We substantially rewrote the discussion to improve*
*this aspect. All CAN model cited in the manuscript, even though they not explicitly require all speed*
*and AHV cells to conjunctive, they have no use and no explanation for cell that do not code*
*conjunctively. For a more detailed answer, see line 200 of the present document.*
**48.** “Ultimately, they challenge the very concept of a well-defined representation of the instantaneous
variables.” I’m not sure what this sentence means.
*We agree with the reviewer that the sentence was not clear. We changed it in the context of a wide*
*restructuration of the discussion.*
**49.** “In terms of attractor structures, they support extending CAN models to incorporate dynamical
variables in what their output “represents”, for example including units firing at a baseline rate for zero
AHV, and quiescent for high either CW or CCW AHV.” What does ‘dynamical variables’ mean? I also
don’t understand the last argument; the example described either sounds like a negatively bidirectional
AHV cell or a linearly asymmetrical one. But the authors’ analyses excluded all of the former and as
far as I can tell they never distinguish between linearly asymmetrical and unidirectional tuning. So it
seems strange that the authors single these out as important for future models.
*We agree with the reviewer that this sentence was not well formulated, and we have now taken it out in*
*our broad reformulation of the discussion.*
**50.** "... only a negligible proportion of self-motion neurons responded preferentially for a given speed.
The vast majority [...] linearly (or quasi-linearly) ramped up in response to increasing speed."
Is 8% negligible? A lot of cells were also detected by the GLM that seemed to have non-linear tuning
curves, which is in agreement with current reports that as many as 50% of speed cells may have a
nonlinear tuning curve (Hardcastle et al. <https://doi.org/10.1016/j.neuron.2017.03.025>). Taking this into
account the conclusion that derivative cells are therefore distinct from non-derivative ones is less
convincing.
*We adjusted the text to replace negligible with very small and added the percentage of 8%. Note that*
*the results on difference encoding between primary and derivative holds when considering GLM*
*detected cells. This control is now added to the main text.*
**51.** "...we demonstrated that both angular and linear self-motion (derivative) signals were encoded in
a different manner..." 'Both' is redundant in this sentence. The fact that many cells are conjunctive
seems to undermine this argument: how differently can the encoding be for derivative and primary
signals when they exist in the same cells? I think the main argument has to do with baseline firing rates
– the fact that derivative cells tend to have higher baseline rates? As I said in an earlier comment I don't
think the authors actually show this. But now I'm wondering how this works in conjunctive cells, Fig
SF6 doesn't seem to differentiate conjunctive cell types, i.e. do grid x speed cells have a higher baseline
firing rate than pure grid cells? If that was the case the authors' argument would be better founded.
More generally, are grid x speed cells just grid cells with added speed coding (i.e. in-field firing is speed
modulated) or are they speed cells with added grid modulation (i.e. firing everywhere in a speed
modulated manner but with spatial peaks in a grid)?
*We have removed 'both' from the sentence. We had originally constructed the analysis illustrated in fig*
*SF6 on the "pure cell" only, this in order to have a meaningful comparison between non-overlapping*
*populations. However, we absolutely agree with the reviewer that conjunction may influence that result*
*and thus tested it. The result is maintained if we include conjunctive cells – even if the effect shrinks as*
*a consequence that conjunctive cells appear in more than one distribution. This control is reported in*
*the text.*
**52.** "It is possible that similar mechanisms operate in mammals." Would be good to cite Gardner et
al. <https://doi.org/10.1101/2021.02.25.432776> here.
*As far as we are aware, this study is not peer-reviewed yet.*
**53.** "...but also did not present evidence of clear-cut categorical distinctions between conjunctive and
non-conjunctive cells." This was also explored in detail by Hardcastle et al.
<https://doi.org/10.1016/j.neuron.2017.03.025> and Mallory et al. [https://doi.org/10.1038/s41467-021-](https://doi.org/10.1038/s41467-021-20936-8)
[20936-8](https://doi.org/10.1038/s41467-021-20936-8)
*We agree with the reviewer that this aspect of our data is in agreement with the work of Hardcastle et*
*al. And Mallory et al – in the context of MEC superficial layers.*
**54.** "One exception to this observation was the absence of AHV and HD cells in MEC LII..." Mallory
et al. also looked at this closely and suggest that this coding is underestimated in superficial layers due
to nonlinear tuning curves (their section: Mixed Selectivity in Superficial Medial Entorhinal Neurons).
Is it possible that a similar explanation might apply here? Could AHV and HD both be represented
nonlinearly in superficial layers for some reason?
*We do find a few cells (5) modulated by AHV when using the GLM method, yet this seems to be due to*
*a coarser criterion for statistical significance rather than the presence of nonlinear coding per se. Only*
*a very small percentage of GLM cell were responding to AHV bands, and many did not show any*
*recognizable pattern in their tuning curves.*
*As for HD cells, their absence in MEC LII (in rats) is a robust result from many that we replicate in*
*our data. Since the tuning profile of HD cells is not linear, it is unclear to us what feature would GLM*
*analysis pick up that escapes mean-vector-length or similar scores. We hypothesize that the larger*
*number of HD cells yielded by the GLM methods finds might be due to large hypothesis class, that is*
*difficult to constrain effectively to low level of false positives.*
**55.** “Historically, running speed has been reported...” I found this paragraph quite confusing. Why is it
surprising that only a minority of AHV and speed cells were theta modulated? I’m not sure this matters
at all for the models you describe. As far as I can see the Zutshi et al. citation doesn’t say anything about
medial septum and grid cells, maybe you meant something like Carpenter et al.
<https://doi.org/10.1038/s41598-017-15100-6?>
Even so, there are other reasons to think grid cells require the linear velocity modulation of theta (i.e.
Winter et al. <http://dx.doi.org/10.1016/j.cub.2015.08.034>). I’m not sure what point you are making with
the bat results, do they undermine the importance of theta or not? I think you have to at least cite Barry
et al. <https://doi.org/10.1038/nature11276> when discussing this. Why does investigating the theta
modulation of self-motion neurons help with any of these questions? Hinman et
al. <http://dx.doi.org/10.1016/j.neuron.2016.06.027> have even shown that the speed-theta frequency
relationship (which may or may not affect grid cells) and speed cell signal (which I guess hasn’t been
tested) are dissociable, so I’m not really sure what the conclusion is here.
*To answer the reviewer concerns, we have now revised this paragraph. Our rationale to investigate the*
*relationship between theta and self-motion coding comes from the fact that theta rhythms are known to*
*be associated, at least at the LFP level, with locomotive activity in rodents. If this relationship were to*
*translate at the level of individual cell firing, we would expect many speed cells (and possibly AHV*
*cells) to be theta-modulated. Instead, we found an independence – in the sense of the independent*
*assignment hypothesis, see answer to point #30 – between self-motion modulation and theta*
*modulation.*
*The fact that theta and self-motion coding are independent resonates with recent studies outlined in the*
*discussion and does scale back the putative role of theta in locomotion. One was just published by*
*Kropff et. Al, demonstrating that theta may be more acceleration-dependent than speed-dependent.*
*Another line of evidence resides in the work conducted on bats, suggesting that most of the hippocampal*
*phenomenology associated with spatial cognition and locomotion is present without the need of a strong*
*periodical modulation.*
*We thank the reviewer for their suggestion to cite Barry et al. A strong overlook from our part given*
*their important theoretical contribution to the understanding of the possible role of theta. Same goes*
*for the results from Hinman et. al. Their results are indeed in the same line of ours, suggesting that*
*speed-theta association at the LFP level is not necessarily translated at the single cell level. On that*
*note, we would like to stress how the independence we found suggest a reduced association between*
*theta and self-motion signals but does not imply that theta does not have a role in spatial cognition. In*
*this sense, our results are not more challenging for oscillatory interference models than they are for*
*CANs.*
**56.** “In conclusion, we provide clear evidence ...” See earlier comment about previous studies showing
conjunctive cells in mEC and DTN.
*See answer to main point #3 and #4.*
**57.** “Yet, the analysis of these neurons reveals features divergent from those expected of units serving
solely to update the instantaneous representation of static variables.” I’m really not sure what features
this sentence refers to. I think this is a general issue with the clarity of the discussion though.
*Based on the reviewer comment, we have deleted this specific sentence.*
**58.** “Our work urges for the revision of such models so that they can (i) express dynamical continuous
attractors and (ii) account for the apparent random nature of the spatial code, as well as its peculiar lack
of a clear organization.” What does ‘dynamical continuous attractor’ mean? Why is the spatial code
‘random’? What is the issue with the organisation – is it that cells were not very organised in layers or
that features were not organised between cells? We would expect both pure and conjunctive cells, both
of which were observed and other models predict each type in all layers, so where is the issue?
*We changed the main text to clarify these aspects. We refer to our answer line 200 for a more detailed*
*explanation of our arguments on the implications of our findings for CAN models.*
**59.** “We hypothesize that derivative algorithms may have a generalized role in the updating of
continuously varying information, not just of a spatial nature.” What evidence is this based on? This is
completely out of the paper’s scope.
*We agree with the reviewer that this hypothesis to be proven would require data that are not presented*
*here – as it was acknowledged by our next sentence “further studies ...”. We hoped that this sentence*
*would read as speculative in this context. The rationale behind this hypothesis is (i) our finding that*
*derivative and primary signal are encoded differently and that (ii) the hippocampal region has been*
*shown to code for primary signal that were not only spatial. Our goal with this sentence was to open*
*up a new theoretical frame that could described a generalisation of updating of information beyond the*
*spatial code – which is in our opinion very relevant.*
Methods
**60.** Section: “Bidirectional angular velocity score” The authors could instead calculate the CW
correlation and CCW correlations separately and then take the maximum value as the AHV score. This
would detect uni- and bi-directional cells as well as linearly-asymmetrical ones.
i.e. <https://doi.org/10.1523/JNEUROSCI.21-15-05740.2001>,
and <https://dx.doi.org/10.1177%2F2398212817721859>
*We thank the reviewer for the suggestion. We opted for the Pearson correlation method because it does*
*not require binning data to produce a tuning curve. Binning introduces arbitrariness that we limited to*
*only the second step of the analysis of the coding properties. With our method, it is not possible to take*
*the maximum of CW and CCW correlation: the Pearson correlation between firing rate and AHV yields*
*one single value for each cell, that is high and positive for CW cells, high and negative for CCW. These*
*two populations are mutually exclusive.*
**61.** Sections: “Shuffling” and “Speed Score”
Should Kropff et al. (2015; <https://doi.org/10.1038/nature14622>) be cited here? The text is identical to
that paper. If other methods/analyses are taken from previous papers it would be beneficial to cite them.
Not just for recognition but it also strengthens the current manuscript if those methods have already
been published.
*We have been quite explicit in the main text that we were using the same method as in Kropff et al.*
*However, we agree that if the method was read separately, a reference could be provided here as well.*
*This is now adjusted.*
**62.** Spike sorting: somewhere around here it would be good to know what is included in the dataset, i.e.
are all clusters putative cells? Only excitatory or pyramidal? Biased towards HD and grid cells? The
bars in Fig 3 seem to suggest a strong bias towards spatial cell types. I can see the citation to the previous
paper but some info is important enough to warrant mentioning in the manuscript directly. Some basic
information about the recording protocol would also be good to know as well as some information about
HD and AHV behaviour – i.e. did all the animals show the same distribution of AHV values? Or speed?
This is especially important since you want to use a population-wide shuffle; if one rat had much slower
running speed than the rest it would not be correct to combine its shuffles with the others (for example).
See Figure 1b in Mallory et al. (2021; <https://doi.org/10.1038/s41467-021-20936-8>)(sorry to keep
referring to that paper but it is very relevant to the current manuscript and it is still fresh in my mind
from recent reading).
*Most of this information is present in Boccara 2010 to which we refer to this article more clearly now*
*in the methods. Recordings come from putative principal cells. We have also added more information*
*on uniformity of behaviour and non-bias towards cell selectivity.*
**Reviewer #2 (Remarks to the Author):**
This paper reports an analysis of a previously collected data set of 1436 neurons recorded from medial
entorhinal cortex (MEC), presubiculum and parasubiculum as rats freely explored a variety of
environments. Head position and direction were measured using two LEDs mounted on the head, and
used to reconstruct position and linear and angular head velocity as well as place and head direction
correlates and grid patterning. It was found that around 17% neurons showed firing that correlated with
angular head velocity (AHV), a similar proportion with linear velocity, and there was a high degree of
conjunctive encoding between the “primary” variables (position and head direction) and the
“derivative” ones (linear speed and angular head velocity). It is concluded that the results “offer insights
as to how linear/angular speed ... may allow the updating of spatial representations.”
I think these are interesting observations speaking to an important issue and they are beautifully
presented.
*We thank the reviewer for these positive comments.*
However I have concerns about the analyses, and about some apparent contradictions with previous
recent findings in the literature. It also isn't clear to me that these findings add new insights about the
computation of position, as conjunctive encoding has been reported in numerous areas and much
discussed over many years.
**The biggest issue** is that despite claims that “angular velocity coding has not been yet established in
principal neurons of the hippocampal region” and “conjunctive coding had not been found in the
hippocampal region”, in fact a recent paper from the Giacomo lab (Mallory et al
<https://doi.org/10.1038/s41467-021-20936-8>) reports both of these phenomena in MEC. Conjunctive
encoding of position and linear velocity has also been reported some time ago in MEC (Sargolini et al).
It's possible the authors were unaware of this quite recent study, but it does contain some notable
differences from the present findings that it would be good to understand.
For one thing, Mallory et al used a linear-nonlinear method to characterize the tuning to angular head
velocity and found that indeed many neurons were tuned to specific speed bands, whereas the current
study found linear tuning. In the present study the authors tried methods other than Pearson's correlation
“to reduce the dependence of our results on a unique scoring method,” found that the other methods
disagreed somewhat and so reverted to the Pearson's correlation on the grounds that it was the most
conservative measure. This logic did not make sense to me (why even try the other analyses in this
case?) and raises the possibility that their analysis missed non-linear tuning. I think we need a better
understanding of the degree to which the present results disagree with those of Mallory et al., as well
as of the similarities. It is suggested to consider applying the methods of Mallory et al to these data,
although this is a big task.
Indeed, a thorough analysis/comparison of the different methods employed and a discussion of their
strengths and weaknesses would be useful, given that these types of conjunctive analyses, and how to
choose among them, are an issue for many in the field.
*We agree with the reviewer that, given the recent study from Mallory et al – which was indeed not
published at the time of the original submission of the present manuscript –, a thorough comparison
between the GLM method and the Pearson method is required. We have now done so and re-do all our
analyses with the GLM method. We highlight at several points in the text how our main results (and
main message) remain unchanged while using one method or the other. We also have added multiple
references to Mallory's results while underlying how our results contrast and complement theirs. For
a more developed answer (including how the data presented here differs from Sargolini 2006), see
response to reviewer #1 line 23 of the present document as well as response to main point #1 and #2,
and several minor points.*
**Another issue** is that I don't fully understand where the data came from and how they were selected.
The data were recorded in a study that has been previously published, but in that study there were a
total of 654 cells from seven rats in the presubiculum, 528 cells from a different 7 rats in parasubiculum
and 630 cells from 15 rats from yet another previous study, in MEC. It wasn't clear that the recording
environments and rat behaviors were equal across these different experiments. Also, the numbers do
not match those in the present study so some selection has occurred – what was the basis for this
selection? Was it purely on cluster quality or were firing patterns also considered (in particular, was
there a focus on grid cells, which was the interest of the previous study). I'm not opposed in principle
to analysing data from previous experiments but since they were originally collected, selected and
analysed with a different purpose in mind, one needs to be especially careful to avoid accidental biases,
and to be very clear about the provenance of the data.
*We thank the reviewer for the opportunity to be more specific here. The discrepancy in cell numbers*
*rightfully noted by the reviewer comes from the fact that some of the sessions reported in the previous*
*studies were recorded with only one LED and for the present, we selected only the sessions that were*
*recorded with two LEDs, we apologized for not stating this explicitly in the first version of this*
*manuscript, this is now corrected. Regarding recording environment and rat behaviours: they were*
*indeed comparable across experiments.*
**On the plus side**, the observations from pre- and parasubiculum **are** novel, and the absence of angular
head velocity tuning in layer II MEC is intriguing. The analysis of the primary vs. derivative signals is
also interesting and novel, and indeed could usefully be moved from the supplementary section to the
main paper.
*We thank the reviewer for that comment and will follow their advice and move the figure from*
*supplementary S6 to main figure 5.*
**To sum up:** I think more work needs to be done to highlight what is novel, relate the findings here to
previous in the literature, and to develop a little more (including in the abstract) the theoretical insights
these data afford.
*We have now revised the text and the abstract following the reviewer comments. In particular, the*
*discussion was substantially rewritten to better clarify the theoretical implications of our findings and*
*their relation with existing literature.*
Minor comments (R2)
**Abstract:** “These self-motion neurons often conjunctively encoded position and/or direction” – the
conjunction with the velocity signals is the more interesting thing here
*We agree with the reviewer than the conjunction with velocity signals (speed and AHV) is the most*
*interesting aspect. Our intent with this sentence was precisely to highlight this fact, that the velocity*
*signals we discover are often in conjunction with the “primary” signals of position and direction. If*
*our understanding of the reviewer comment is wrong, could the reviewer clarify what aspect would they*
*referring to?*
**P2** “Intelligent design” is conventionally used to mean “God” – I think a preferable term is “functional”
or perhaps “optimal”
**P2** “Intelligent design” is conventionally used to mean “God” – I think a preferable
term is “functional” or perhaps “optimal”
*This has now been modified.*
**P3** “general self-organizing derivative algorithm” is too vague at this early stage and needs explanation
*We thank the reviewer to pointing out this lack of clarity in the introduction. We completely agree and*
*therefore modified the text there to explain better the notion of “general self-organizing derivative*
*algorithm”.*
The absence of AHV encoding in MEC layer II is intriguing – any speculations as to what this could
mean, or is it just sampling variability?
*We agree with the reviewer on pointing out that MEC LII is an interesting outlier. We have now*
*modified slightly the text to even more emphasise how LII is at odd with all the other regions we*
*analysed – an important point when comparing our data with other studies. We are quite confident that*
*this is not a results of sampling variability, since the effect is very strong – basically a complete absence*
*of both HD and AHV cell in MEC LII.*
*As for speculations, we advanced a few hypotheses in the discussion linked to the absence of HD signal*
*in LII. Could that be linked to a need for a local network with HD to generate a AHV signal? Maybe,*
*but one should consider the important fact that long range connections exist within the MEC network.*
*It could also be that LII is dedicated to spatial integration above anything else given the very high*
*proportion of spatially modulated cells evidenced in that layer.*
Why was the analysis restricted to positively modulated firing rates – might there not have been cells
inhibited by AHV?
*Absolutely! We agree with the reviewer that negative modulation is interesting. We have now*
*incorporated results for negatively modulated neurons when discussion GLM results. See response to*
*reviewer 1 (major point #2 and minor point #15, #21 and #23) for a more complete answer on this point*
*as well as our original rationale to focus on positive modulation.*
**P4** “with naturally the exception...” needs explaining as readers won’t necessarily remember that there
weren’t any AHV-modulated neurons there.
*The text has now been adjusted.*
**Concerning the two additional methods of analysis** – the results from the information measure
(Skaggs) weren’t reported that I could see. As mentioned, these analyses are critical and should be a
major part of a revised manuscript. I found it odd that the GLM found 54% of cells anti-correlated with
AHV – I feel this shouldn’t be brushed off but rather investigated and explained. What happens when
Mallory et al’s methods are applied to these data?
*Following the reviewers comments we have now taken out the Skaggs methods as they did not bring*
*ample information in the context of the paper beyond what the GLM analyses brought forth.*
*We have re-done all our analyses following the GLM methods. As mentioned above, the main results*
*are not changed. This is explicitly stated in the text. We agree that 54% of anticorrelated AHV cells is*
*a high number but it does not seem incongruent with previous reports based on the GLM method. Note*
*that this 54% refer to GLM only cells (cell picked up by GLM and not by Pearson). We have verified*
*visually all these traces and included some examples in the supplementary figures. As mentioned in our*
*answer to previous comments, we have added new text to compare the GLM and Pearson results within*
*the results and the discussion.*
**P6** t-test needs degrees of freedom and the t statistic (here and elsewhere). What was being compared
in this primary/derivative conjunction? I was expecting just a chi-square test.
*We decided to change to a non-parametric Mann-Whitney U test for the comparison between primary*
*and derivative cells, since the distributions we are comparing are non-normal. The results were not*
*impacted by the test choice.*
*Proportion z-test were used to compare percentages of modulated cell in each layer to the percentage*
*found in the general population, using the proportion z-test from statsmodel python API –*
*https://www.statsmodels.org/stable/generated/statsmodels.stats.proportion.proportions_ztest.html.*
*The test we use (the two sample, two side alternative) is equivalent to the proportion chi-square test, as*
*reported in the API references. Percentages of conjunctively coding cell were compared with the null*
*hypothesis of independent assignment with a binomial test, from the same API –*
*https://www.statsmodels.org/stable/generated/statsmodels.stats.proportion.binom_test.html – that*
*compares the observed probability of conjunction to the one expected by the product of the two*
*individual probabilities of each of the feature whose conjunction is being investigated.*
In what way is the integration of self-motion information at the unit level incompatible with CAN
models?
*We thank the reviewer for pointing out the lack of clarity of this sentence, that we changed in the main*
*text. What we meant was that many of the observed features of self-motion coding at the unit level (and*
*not, indeed, self-motion information at the unit level per se), are at odds with CAN models. We tried to*
*clarify this in the new discussion and in response to reviewer one (line 200). We include here a brief*
*summary of the main points:*
*CAN models give no explanation and have no use for cell coding purely for speed and AHV, which we*
*observe in good numbers.*
*Conjunctive cells of the type predicted by CAN models are a small percentage (as compatible with the*
*independent assignment we hypothesize).*
*CAN model implicitly expect a very structured connectivity between different circuit, hardly compatible*
*with the conjunctive (lack of) structure we observe.*
*Our data hints that the same cells are part of the positional (grid x HD x speed) and directional (AHV*
*x HD) CAN. Whether these models can operate on such a shared population is unclear. Moreover, it is*
*likely, from the lack of structure in conjunctive coding, that these cells are not only shared between*
*these two CANs, but with other circuits as well. Under which conditions CANs can still operate in this*
*scenario is an interesting open question.*
“Self-standing” is a new and unintuitive terminology.
*We have been considering terminologies such as ‘pure’, ‘stand-alone’ and ‘independent’ yet ‘self-*
*standing’ was the best alternative we found. We have now modified the text so that a definition of that*
*term will be present the first time that it is written. We are very much welcoming suggestion from the*
*reviewer.*
I found the **discussion** interesting but a little rambling and could be tightened a little, nad explain a little
more clearly why the findings constrain some of the CAN models.
*We thank the reviewer for their interest in our discussion. Based on this comment and those of other*
*reviewers, we have now restructure part of the discussion, striving in particular to clarify the theoretical*
*implications of our data.*
**Fig. 2e** I’m not clear on the meaning of “instantaneous linear speed” for this AHV cell.
*We meant figure 2 to illustrate linear speed coding and the example cell showed is a speed cell, and not*
*an AHV cell (a similar example of an AHV cell is reported in figure 1).*
**Fig. SF6** is missing part (d) in the legend
*We thank the reviewer for noting this mistake. This is now corrected.*
**Reviewer #3 (Remarks to the Author):**
Understanding how the brain encodes information about experience in the activity of distributed
populations of neurons is arguably one of the most challenging goals of modern neuroscience. Many
insights about high-level neural coding have come from studying the brain’s representation of space
and its neural correlates in the entorhinal-hippocampal network. Here, the activity of individual neurons
is modulated by variables related to the animal’s movement in space, such as the animal’s location, the
direction of its head, the speed of its movements, or the presence of object or boundaries on its path.
Computational models predict that spatial tuning might emerge from the integration of information
about the environment and the animal’s movement in it, and the concerted activity of neurons that are
locally connected according to rigid connectivity rules within continuous attractor networks (CAN).
Specifically, CAN models hypothesize that populations of neurons whose activity
is exclusively modulated by position or direction might receive inputs from cells exhibiting a mixed
selectivity for a repertoire of variables associated with the position, direction, and self-motion.
In this study, Spalla, Treves, and Boccara investigate the neuronal basis of self-motion coding in the
entorhinal cortex, presubiculum, and parasubiculum – three areas of the entorhinal-hippocampal
network where the senior author of the current study has previously recorded neurons exhibiting
spatially-modulated firing (Boccara et al., Nat Neurosci 2010). Through a thorough re-analysis of a
previously published dataset, the authors report (i) the identification of neurons whose firing is
modulated by the linear and angular speed at which the animal is moving; (ii) the mixed-selectivity
exhibited by the neurons that encode such signals; and (iii) the anatomically-distributed nature of the
neuronal network processing linear and angular velocity.
While the presence of signals related to self-motion has been predicted by CAN models since their
inception, proofs of their existence and their relation with the other spatial variables encoded within the
entorhinal cortex and hippocampus had, until recently, been lacking. As such, this study represents an
advancement in the current knowledge of the repertoire of spatially-modulated signals that can be
recorded in the extended hippocampal network. Its main strength lies in the identification of the
distributed nature of the self-motion signals, which seem to spread across multiple layers and multiple
areas of the entorhinal-hippocampal network. Moreover, since such signals had previously been
hypothesized to be essential for the computations of CANs and the emergence of spatially-modulated
firing patterns, the present study confirms a long-standing prediction within the field and proposes a
very interesting idea (i.e., the distinction between coding of direct variables and their
derivatives) that calls for further investigation. The fact that the same dataset has now been used for
multiple studies is a testament to the excellence and rigorousness of the experiments analyzed. The
statistical methods deployed are appropriate and adequately described.
I have only minor comments regard the formulation of the manuscript: once these will be addressed I
will fully support it for publication in Nature Communications.
*We thank the reviewer for their very positive assessment of our work.*
1. In a recent study, it has been reported that entorhinal cortex neurons are variably tuned to self-motion
signals, including linear and angular velocity (Mallory et al., Nat Commun. 2021 Jan 28;12(1):671.).
The existence of such study clearly affects the formulations of the present manuscript, and forces the
authors to tone-down some of their statements, like “angular velocity has not yet been established in
principal neurons of the hippocampal region”, or “... providing the first recording in the hippocampal
region of the conjunctive coding...”. Also, not citing the Mallory study in the current manuscript would
be an unforgivable oversight: the authors should go further and put some thoughts into how the two
studies fit together and report this in the main text.
*As mentioned above, this manuscript was submitted prior to Mallory et al publication. We have now*
*adjusted the text to cite and discuss their work.*
2. On page 2, the authors write: “it has remained unclear whether angular velocity coding reflects an
intelligent design”. This sentence should be re-written taking into consideration that “intelligent design”
is an expression that is charged with meaning that most likely goes beyond what the authors meant to
express. If instead, their use of the expression is to be framed in the context of a theological discussion
about the origin of life, then this point should definitely be explored further in the text.
*This has now been adjusted. We agree that the theological discussion may have been interesting, yet*
*we will leave it for other journals.*
3. On page 6, the authors state that “the fact that self-motion information is integrated at the unit level
in all cell types and tested layers is incompatible with current CAN models and calls for revision”. Why
such finding is incompatible with current CAN models is not clear to me. First of all, not all neurons
recorded exhibit conjunctive coding (see fig 3), so this dataset is still being compatible with the idea
that only the hidden layer should exhibit conjunctive coding to support the “pure” tuning of the other
units of the model. Second, I am not aware that the CAN model is strictly associated with an exclusive
correspondence between the layers of the attractor network and the layers of the MEC. Why would the
current model not work if units with conjunctive coding and units with pure coding were to be
anatomically intermingled, provided that their connectivity would still match the one predicted in CAN
models?
*We absolutely agree that this sentence was misleading and did not reflect our opinion. This has now*
*been modified as part of a substantial rewriting of the discussion. Similar points were raised by the*
*other reviewers (see line 200 and 966 of the present document). We follow with a summary on what is*
*at odd between our data and CAN models:*
*CAN models give no explanation and have no use for cell coding purely for speed and AHV, which we*
*observe in good numbers.*
*Conjunctive cells of the type predicted by CAN models are a small percentage (as compatible with the*
*independent assignment we hypothesize).*
*CAN model implicitly expect a very structured connectivity between different circuit, hardly compatible*
*with the conjunctive (lack of) structure we observe.*
*Our data hints that the same cells are part of the positional (grid x HD x speed) and directional (AHV*
*x HD) CAN. Whether these models can operate on such a shared population is unclear. Moreover, it is*
*likely, from the lack of structure in conjunctive coding, that these cells are not only shared between*
*these two CANs, but with other circuits as well. Under which conditions CANs can still operate in this*
*scenario is an interesting open question.*
*Lastly, we agree with the reviewer that anatomical intermingling per se would not constitute a problem*
*for CAN models, but the connection structure they hypothesize requires quite careful fine tuning – let’s*
*think, for example, at the systematic offset between input and output connections to each of the “steering*
*population” of a ring attractor. Whether this requirement can be satisfied in the presence of extreme*
*noise in the connectivity structure is an open (and interesting, in our opinion) theoretical question.*
**4.** The authors state that velocity coding is independent of theta modulation. How do the authors
reconcile this finding with the recent finding that theta rhythm is not linearly modulated by speed as it
was previously thought, and that its frequency is modulated by acceleration (E Kropff et al, Neuron,
2021)? Is it possible that such modulation would lead to an overestimation of AHV tuning in the theta-
modulated neurons?
*The fact that theta (both at the LFP and unit level) depends more on acceleration than speed does*
*indeed resonate with our findings, that show theta and speed modulation to be independent (in the sense*
*of the independent assignment hypothesis).*
*In our opinion, a dependence of theta on acceleration would not affect our estimation of theta*
*dependence of AHV cells. Indeed, in this case we would also find that the percentage of AHV cells that*
*are theta modulated is the same as the percentage of theta modulated cells in the whole population.*
**5.** Last, the authors seem to draw some conclusions from the absence of AHV and HD cells in MEC
LII, which, in their view, would suggest that “AHV signal is needed locally among the same cells coding
for the primary signal it serves to update”. It is now a consolidated notion that the anatomical
distribution of spatially modulated cell types is species-specific: even though HD cells seem not to be
present in MEC LII in the rat (see the present study, but also Sargolini et al., Science 2006), they are
abundant in the mouse MEC LII (see Rowland et al., 2018). Given that it would not be reasonable to
hypothesize that spatial coding follows different computational principles in related species like the
mouse and the rat, it remains to be determined if the anatomical distribution of functional cell types
bears any information at all about the computational function that these cell types subserve.
*We agree that this absence is odd. See our answer to reviewer 2 and our modified discussion that now*
*includes a reference to Dave Rowland’s interesting report.*

REVIEWERS' COMMENTS

Reviewer #1 (Remarks to the Author):

Spalla et al. have greatly improved their introductory and discussion text. They have verified their analyses against a GLM method and within-cell shuffles. The authors have addressed all of my previous major comments except one where I still feel their arguments are unconvincing (see minor comment 1 below). However, this is no longer the central focus of the discussion which now much clearly covers a broader range of findings. Assuming the authors address the minor comments below I fully support the publication of their manuscript in Nature Communications.

Minor comments

1. Page 3 line 5-11 and page 12 line 30; having looked through the models the authors cite (e.g. refs 13, 15, 26-30 31) I could not find any that predict or rely on obvious structure, topographical or physiological clustering – except the head direction models that were structured to mirror the electrophysiological evidence (they can still actually operate if the cell types are admixed and allowed to self-organize). In these models the topography of the network is completely arbitrary, the important feature is the connectivity between neurons. Can the authors a) properly cite a specific model(s) that strictly depends on physiological structure, b) remove this as a central argument of the paper or c) make it clearer that topography is not an important feature of these models? Page 13 line 8; cells with unaccounted for roles doesn't really undermine a CAN, they could be incorporated into an updated CAN and for all we know might even improve the model, or they might just fulfil another purpose separate to the CAN. Page 13 line 10; this is assuming all of the cells recorded are part of the CAN, cells with mixed position and directional responses could just reflect the combination of two CAN outputs and may be used for some other purpose entirely. As far as I can see the authors have shown all of the components necessary for a CAN and nothing directly contradicts these models.

2. Nature allows citation of preprints: "Preprints may be cited in the reference list of articles under consideration at Nature Portfolio journals" (<https://www.nature.com/ncomms/editorial-policies/preprints-conference-proceedings>). Please reconsider citing the recent preprint from the Moser group showing strong evidence for a CAN in rat grid cells (Gardner et al. 2021; <https://www.biorxiv.org/content/10.1101/2021.02.25.432776v1>) as this is extremely relevant to the current manuscript.

3. Page 13 line 4: I think this has been successfully included in CAN models; Zhang (1996; <https://doi.org/10.1523/JNEUROSCI.16-06-02112.1996>) section 5.2 Biological mechanisms:

"In general, all we need for the activity shift is to induce some weak anisotropy in the effective connections, presumably by vestibular inputs". Basically, you can have cells that are not silent in the

absence of movement as long as they are countered by a cell with the opposite response. What matters is the value left after summing all of these responses. Alternatively, these cells might not be used in the CAN at all.

4. Page 41 line 21: this is a good point I had not really considered. It might be worth making this point in the main text results section. Page 42 line 6: "...than the one of, avoiding pitfalls..." this sentence may require attention.

Reviewer #2 (Remarks to the Author):

The authors have done an excellent job of responding to my comments; most importantly those relating to the Mallory et al paper. I have just a few remaining comments, all minor but which I think affect the clarity and quality of the paper.

The first is that although the paper is very well written, there are quite a few small grammatical errors and spelling mistakes which interrupt the flow although they don't affect the meaning (examples include Mann-Withney, Kroppf, "area" instead of "areas" etc). I'm never sure of the degree to which copyeditors catch these things but I recommend the authors review the paper very carefully too.

Second, I have a couple of issues with nomenclature: which again, are not issues that impede meaning, since they are explained in the text, but which – if widely adopted – could cause future confusion because they contain false implications. The first one is "self-standing" which I commented on in the first version of the manuscript: it's clear what is meant by it in the manuscript because of the context, and because we are told, but it is not a term that accurately reflects the property of interest. "Self-standing" means "existing without outside assistance or dependence on others", which clearly is not true of a neuron, or its signal. The authors are referring to a firing correlate that seems "pure" (a word they do use at one point) but again, there is no real fundamental sense in which any neuronal signal is pure. They also use the word "non-conjunctive" but we don't know for sure that these neurons (e.g., head direction neurons) are truly non-conjunctive either. What they really mean, I think, are "neurons whose strongest firing correlate is something we can describe in a computational model with a single variable" and so I wondered about univariate vs. multivariate, with the caveat that even "univariate" will probably turn out not to be true once we know more. I don't have strong answers, but I do find self-standing to be the least good alternative.

My second problem is with the primary/derivative terminology. I get that “derivative” comes from the mathematical calculus of relating, say, position to distance to velocity etc, but pairing it with “primary” invokes, in the mind, the idea that the second quantity is derived from the first in the more common sense of derive, which is “originates from”. Whereas in the brain, it is equally the case that the reverse is likely: the “primary” signal of, say, head direction “derives from” the variables of angular velocity and time (by integration). Thus, I think the primary and derivative terms are a little misleading and I would suggest something else: maybe static vs. dynamic? That accounts for the relationship to time without implying any causal direction.

Finally on terminology: “design” has persisted, with its problematic implications. Indeed that whole phrase is slightly clunky: “angular velocity coding expresses a pre-wired design” and should be reworded.

Other comments

The abstract says “yet lacked a structured organisation.” But the reader doesn’t know at this point why this is significant, so more explanation is needed.

P2 L17 “Reports of angular velocity modulation have come so far from recordings of subcortical structures” – but there have also been reports of angular velocity modulation in cortex, e.g. RSC, which is sometimes considered part of the extended hippocampal system, so this should be noted.

P2 L30 “These reports and the underlying mechanisms are likely restricted, in insects, to directional coding.” – why so?

P3 L4 “evidence for the elusive “hidden layer”, pillar of many CAN models” needs a reference and a little more explanation.

P3 L7 “It appears that direction, position and speed selectivity are randomly admixed with each other.” There are two senses in which this is true, which I think need to be untangled here. The first is that they are anatomically mingled, as in all three selectivities are present in all the structures. The second is that the nature of the admixture seems random inasmuch as there are no more conjunctive (multivariate) neurons that one would expect from a random assignment of coding variable to neuron.

P3 L27 “region-wise shuffle” could use a little explanation, here.

P4 L22 “taking into account that anticorrelated AHV cell may reflect a modulation not related to AHV, but rather a process that takes places only when the head of the animal is still.” I didn’t follow this reasoning – if it is anticorrelated then there is a negative relationship of firing rate to AHV, which pertains to the rotating (not still) head. Do the authors mean uncorrelated?

P12 L29 “yet they also challenge some of the postulates currently assumed by CAN models which favour specific connectivity rules and segregated coding.” This assertion needs some discussion and references.

P14 L1 The discussion of theta should mention that its frequency has also been linked to running speed. Also, given its recent link to acceleration, why did you not look for acceleration correlates in your data?

P16 L6 “position” is “smoothed position” I think? It would be useful to remind readers of this.

Reviewer #3 (Remarks to the Author):

Thanks to the authors for the amount of work they put into revising the manuscript. The introduction of new analyses that rely on a different approach (GLM) to quantify the influence of self-motion on individual neurons activity, is an important addition to the previous version of the paper and consolidates the authors claims. The manuscript has also gained clarity, especially regarding the implications of the findings in regard to CANs models. The findings have now been properly contextualized in light of recent work from the Giacomo group. My concerns have been successfully addressed, hence I recommend the paper for publication.

Response to reviewers

Reviewer #1 (Remarks to the Author):

Spalla et al. have greatly improved their introductory and discussion text. They have verified their analyses against a GLM method and within-cell shuffles. The authors have addressed all of my previous major comments except one where I still feel their arguments are unconvincing (see minor comment 1 below). However, this is no longer the central focus of the discussion which now much more clearly covers a broader range of findings. Assuming the authors address the minor comments below I fully support the publication of their manuscript in Nature Communications.

We thank reviewer #1 for their very positive view on our work.

Minor comments

1. Page 3 line 5-11 and page 12 line 30; having looked through the models the authors cite (e.g. refs 13, 15, 26-30 31) I could not find any that predict or rely on obvious structure, topographical or physiological clustering – except the head direction models that were structured to mirror the electrophysiological evidence (they can still actually operate if the cell types are admixed and allowed to self-organize). In these models the topography of the network is completely arbitrary, the important feature is the connectivity between neurons. Can the authors a) properly cite a specific model(s) that strictly depends on physiological structure, b) remove this as a central argument of the paper or c) make it clearer that topography is not an important feature of these models?

We agree that topography is not an important feature of CAN models. Our argument is that CAN models (refs 13, 15, 26-30 31) are based on a precise connectivity structure and that such a precise connectivity structure is at odds with our findings of random and unstructured mixing of firing correlates within units. Based on reviewer #1 and #2 comments, we now realised that this argument was not clear and that our phrasing could be interpreted as if both random mixing of selectivity and a lack of topography, that we both report, were major challenges for CAN – this is not the case. We thank the reviewers for highlighting this possible misinterpretation. We have now modified the text accordingly in the introduction & the discussion to distinguish between these two features (see also response to reviewer #2) and make clear that topography is not required by CAN.

Page 13 line 8; cells with unaccounted for roles doesn't really undermine a CAN, they could be incorporated into an updated CAN and for all we know might even improve the model, or they might just fulfil another purpose separate to the CAN. Page 13 line 10; this is assuming all of the cells recorded are part of the CAN, cells with mixed position and directional responses could just reflect the combination of two CAN outputs and may be used for some other purpose entirely. As far as I can see the authors have shown all of the components necessary for a CAN and nothing directly contradicts these models.

We agree that “self-standing cells” do not directly undermine CAN. What we stated in our discussion is that (i) there is no mention of them in current models and that (ii) it would be interesting to understand whether they have a role or not in CAN, especially given their relative abundance. What the reviewer proposes is indeed an interesting theory and deserves, in our opinion, to be developed in further theoretical publication/comment. This is precisely the type of debate that we thought our findings would open for.

2. Nature allows citation of preprints: “Preprints may be cited in the reference list of articles under consideration at Nature Portfolio journals” (<https://www.nature.com/ncomms/editorial-policies/preprints-conference-proceedings>). Please reconsider citing the recent preprint from the Moser group showing strong evidence for a CAN in rat grid cells (Gardner et al. 2021; <https://www.biorxiv.org/content/10.1101/2021.02.25.432776v1>) as this is extremely relevant to the current manuscript.

We are now citing this manuscript in the discussion (P12, L27)

3. Page 13 line 4: I think this has been successfully included in CAN models; Zhang (1996; <https://doi.org/10.1523/JNEUROSCI.16-06-02112.1996>) section 5.2 Biological mechanisms:

“In general, all we need for the activity shift is to induce some weak anisotropy in the effective connections, presumably by vestibular inputs”. Basically, you can have cells that are not silent in the absence of movement as long as they are countered by a cell with the opposite response. What matters is the value left after summing all of these responses. Alternatively, these cells might not be used in the CAN at all.

We agree that, in this configuration, “two steering populations would have to constantly level their inputs for the activity bump to remain still.” This was indeed our argument in the discussion: the sum of the left and right response should equate zero in absence of input for the population to be stable, which is a fine-tuning requirement because even a slight bias would induce the population to shift spontaneously and indefinitely in one direction.

4. Page 41 line 21: this is a good point I had not really considered. It might be worth making this point in the main text results section.

This was already referred to P4, l 24.

Page 42 line 6: “...than the one of, avoiding pitfalls...” this sentence may require attention.

The sentence is now corrected.

Reviewer #2 (Remarks to the Author):

The authors have done an excellent job of responding to my comments; most importantly those relating to the Mallory et al paper. I have just a few remaining comments, all minor but which I think affect the clarity and quality of the paper.

We thank reviewer #2 for their very positive view on our work.

The first is that although the paper is very well written, there are quite a few small grammatical errors and spelling mistakes which interrupt the flow although they don't affect the meaning (examples include Mann-Withney, Kroppf, “area” instead of “areas” etc). I'm never sure of the degree to which copyeditors catch these things but I recommend the authors review the paper very carefully too.

We thank the reviewer for spotting those two typos that have now been corrected in the revised manuscript. We could not find more. I hope that any remaining error can be caught by copyeditors.

Second, I have a couple of issues with nomenclature: which again, are not issues that impede meaning, since they are explained in the text, but which – if widely adopted – could cause future confusion because they contain false implications. The first one is “self-standing” which I commented on in the first version of the manuscript: it's clear what is meant by it in the manuscript because of the context, and because we are told, but it is not a term that accurately reflects the property of interest. “Self-standing” means “existing without outside assistance or dependence on others”, which clearly is not true of a neuron, or its signal. The authors are referring to a firing correlate that seems “pure” (a word they do use at one point) but again, there is no real fundamental sense in which any neuronal signal is pure. They also use the word “non-conjunctive” but we don't know for sure that these neurons (e.g., head direction neurons) are truly non-conjunctive either. What they really mean, I think, are “neurons whose strongest firing correlate is something we can describe in a computational model with a single variable” and so I wondered about univariate vs. multivariate, with the caveat that even “univariate” will probably turn out not to be true once we know more. I don't have strong answers, but I do find self-standing to be the least good alternative.

We agree with the reviewer that deciding on nomenclature is complex and that “self-standing” is problematic. We completely understand and agree with the argument of the reviewer. We ourselves had reservation about this appellation. What is clear for us is that we cannot change “conjunctive” to anything else (such as multivariate), given how widely that term is already used. In this context, we propose to use “non-conjunctive” as an alternative to “self-standing”. This with the understanding that “non-conjunctive” means non conjunctive with respect to the correlates we tested for. This change has now been made throughout the manuscript.

My second problem is with the primary/derivative terminology. I get that “derivative” comes from the mathematical calculus of relating, say, position to distance to velocity etc, but pairing it with “primary” invokes, in the mind, the idea that the second quantity is derived from the first in the more common sense of derive, which is “originates from”. Whereas in the brain, it is equally the case that the reverse is likely: the “primary” signal of, say, head direction “derives from” the variables of angular velocity and time (by integration). Thus, I think the primary and derivative terms are a little misleading and I would suggest something else: maybe static vs. dynamic? That accounts for the relationship to time without implying any causal direction.

We think that the static/dynamic terminology is problematic given that both codes could have a dynamic nature – e.g., the preferred position could, in principle, dynamically change over time. In our opinion, derivative has an unambiguous meaning when used in its mathematical terminology, and it is not different from “integration”, which has a precise and clear mathematical meaning and cannot be mistaken with its plain English meaning of “combination/unification/merger” in this context. We therefore decided to keep the derivative terminology for this manuscript.

Finally on terminology: “design” has persisted, with its problematic implications. Indeed that whole phrase is slightly clunky: “angular velocity coding expresses a pre-wired design” and should be re-worded.

We have now re-worded this sentence.

Other comments

The abstract says “yet lacked a structured organisation.” But the reader doesn’t know at this point why this is significant, so more explanation is needed.

Unfortunately given the restriction in number of character (150), we do not have the space to explain this in the abstract. Our opinion is that this term – while maybe ambiguous without the benefit or a lengthier explanatory context – is (i) not erroneous and (ii) intriguing enough for the reader to give them the curiosity to read the full article.

P2 L17 “Reports of angular velocity modulation have come so far from recordings of subcortical structures” – but there have also been reports of angular velocity modulation in cortex, e.g. RSC, which is sometimes considered part of the extended hippocampal system, so this should be noted.

We have modified this introductory sentence to include the presence of angular velocity signal in the RSC, presence that we had already highlighted in the discussion (P11 L16).

P2 L30 “These reports and the underlying mechanisms are likely restricted, in insects, to directional coding.” – why so?

We have modified this sentence to tuned down our statement as we have no arguments beside that such a beautiful, yet extremely rigid (up to the single synapse) toroidal structure found for directional coding in flies would require an extreme wiring precision that we think unlikely to be found unchanged in more complex species, as well as for other correlates.

P3 L4 “evidence for the elusive “hidden layer”, pillar of many CAN models” needs a reference and a little more explanation.

We have both added an explanation and a reference to that statement.

P3 L7 “It appears that direction, position and speed selectivity are randomly admixed with each other.” There are two senses in which this is true, which I think need to be untangled here. The first is that they are anatomically mingled, as in all three selectivities are present in all the structures. The second is that the nature of the admixture seems random inasmuch as there are no more conjunctive (multivariate) neurons that one would expect from a random assignment of coding variable to neuron.

As we wrote in response to reviewer #1 first comments, we agree that our phrasing was unclear. Both statements are true in our study. We have now modified the text accordingly in the introduction & discussion to distinguish between these two points.

P3 L27 “region-wise shuffle” could use a little explanation, here.

We have both added an explanation and a reference to the methods.

P4 L22 “taking into account that anticorrelated AHV cell may reflect a modulation not related to AHV, but rather a process that takes places only when the head of the animal is still.” I didn’t follow this reasoning – if it is anticorrelated then there is a negative relationship of firing rate to AHV, which pertains to the rotating (not still) head. Do the authors mean uncorrelated?

What we refer to is that cells firing in the absence of behaviour – e.g., when the animal is still – can reflect planning/replay as demonstrated during sharp wave ripples. Activity of this kind would lead to what appear as anti-correlated speed or AHV cells, making them difficult to interpret. This is why often analyses to determine place or grid code, are conducted with a speed filter, and why we treaded carefully in the interpretation of the role of anti-correlated cells.

P12 L29 “yet they also challenge some of the postulates currently assumed by CAN models which favour specific connectivity rules and segregated coding.” This assertion needs some discussion and references.

We have now added a reference. The next sentences of this paragraph provide a discussion of that statement.

P14 L1 The discussion of theta should mention that its frequency has also been linked to running speed. Also, given its recent link to acceleration, why did you not look for acceleration correlates in your data?

We already have a reference to that (see P14 line 5). Theta’s recent link to acceleration came out after submission, and although extremely interesting, will be left for future investigations.

P16 L6 “position” is “smoothed position” I think? It would be useful to remind readers of this.
We have modified accordingly the sentence.

Reviewer #3 (Remarks to the Author):

Thanks to the authors for the amount of work they put into revising the manuscript. The introduction of new analyses that rely on a different approach (GLM) to quantify the influence of self-motion on individual neurons activity, is an important addition to the previous version of the paper and consolidates the authors claims. The manuscript has also gained clarity, especially regarding the implications of the findings in regard to CANs models. The findings have now been properly contextualized in light of recent work from the Giocomo group. My concerns have been successfully addressed, hence I recommend the paper for publication.

We thank reviewer #3 for their very positive view on our work.